# TMEM135 regulates primary ciliogenesis through modulation of intracellular cholesterol distribution

Yunash Maharjan[1], Joon No Lee[1], Seong Ae Kwak[2], Raghbendra Kumar Dutta[1], Channy Park[1], Seong-Kyu Choe[2] & Raekil Park[1,*] (iD)

## Abstract

**Recent evidence has linked the lysosomal cholesterol accumulation in Niemann–Pick type C1 with anomalies associated with primary ciliogenesis. Here, we report that perturbed intracellular cholesterol distribution imposed by lysosomal cholesterol accumulation during TMEM135 depletion is closely associated with impaired ciliogenesis. TMEM135 depletion does not affect the formation of the basal body and the ciliary transition zone. TMEM135 depletion severely blunts Rab8 trafficking to the centrioles without affecting the centriolar localization of Rab11 and Rabin8, the upstream regulators of Rab8 activation. Although TMEM135 depletion prevents enhanced IFT20 localization at the centrioles, ciliary vesicle formation is not affected. Furthermore, enhanced IFT20 localization at the centrioles is dependent on Rab8 activation. Supplementation of cholesterol in complex with cyclodextrin rescues Rab8 trafficking to the centrioles and Rab8 activation, thereby recovering primary ciliogenesis in TMEM135-depleted cells. Taken together, our data suggest that TMEM135 depletion prevents ciliary vesicle elongation, a characteristic of impaired Rab8 function. Our study thus reveals a previously uncharacterized effect of erroneous intracellular cholesterol distribution on impairing Rab8 function and primary ciliogenesis.**

**Keywords** IFT20; intracellular cholesterol transport; peroxisome; primary cilia; Rab8; TMEM135

**Subject Categories** Cell Adhesion, Polarity & Cytoskeleton; Membranes & Trafficking

## Introduction

Primary ciliogenesis is a complex process regulated by polarized vesicle trafficking and intraflagellar transport (IFT) to induce ciliary membrane biogenesis coupled to microtubule axoneme extension from the mother centriole (M-centriole) [1–3]. The early steps of ciliogenesis are characterized by extensive vesicular trafficking to enable the formation of the distal appendage vesicle (DAV) and its fusion into ciliary vesicles (CVs) followed by CV extension [4]. Moreover, recent work has pointed to the role of EHD1 (a member of the Eps15 homology domain protein family) in early ciliogenesis, demonstrating its essential function in loss of CP110 from the M-centriole; capping of CP110 to the M-centriole acts as a suppressor to control the timing of ciliogenesis, and thus, its loss promotes the formation of CVs [4]. EHD1 was also reported to play a role in cholesterol homeostasis through regulation of low-density lipoprotein (LDL) cholesterol uptake [5]. In addition, a recent report indicating alteration of the primary cilium in Niemann–Pick type C1 (NPC1) disease, characterized by lysosomal cholesterol accumulation, and in fibroblasts treated with U18666A (a drug mimicking the NPC1 disease) suggests that the intracellular cholesterol transport pathway or cellular cholesterol distribution could be tightly coupled with ciliogenesis [6].

The distribution of cholesterol among cellular membranes is critical for biological functions such as signal transduction and membrane trafficking, as well as vesicular fusion events. Cholesterol is unevenly distributed within cells and is most abundant in the plasma membrane, with intermediate levels in the Golgi and endocytic recycling compartment (ERC) and low levels in the endoplasmic reticulum (ER) [7]. The accumulation of cholesterol in the lysosomal compartment is known to disturb overall cellular cholesterol distribution, which interferes with the cholesterol-dependent functions of endosomes, thereby directly affecting vesicle trafficking events [8–11]. The search for critical determinants involved in intracellular cholesterol trafficking through genome-wide small hairpin RNA screening in combination with amphotericin B selection revealed the peroxisomal proteins, ABCD1, PEX1, PEX3, PEX10, PEX26, ACOT8, BAAT, and TMEM135, as potential candidates regulating intracellular cholesterol transport through lysosome–peroxisome membrane contact (LPMC) [12]. Deficiency in any of these proteins decreases the possibility of LPMC (due to reduced expression of PI(4,5)P2 in the peroxisome membrane which is required for interaction with synaptotagmin 7 lysosomal membrane) triggering robust cholesterol accumulation in lysosomes that mimic the NPC1 phenotype [12]. Therefore, it is reasonable to speculate that lysosomal cholesterol accumulation negatively affects the vesicle trafficking events associated with ciliogenesis.

1 Department of Biomedical Science & Engineering, Gwangju Institute of Science & Technology, Gwangju, Korea
2 Department of Microbiology and Center for Metabolic Function Regulation, Wonkwang University School of Medicine, Iksan, Korea
*Corresponding author. Tel: +82 62 715 5361; Fax: +82 62 715 5309; E-mail: rkpark@gist.ac.kr

To test this possibility, we investigated the functional role of impaired intracellular cholesterol transport in the early steps of primary ciliogenesis. Among the peroxisome proteins mentioned above, we focused on TMEM135 as its function is largely unknown. Intracellular TMEM135 localization is controversial, as it has been found to localize in peroxisomes [13,14] and mitochondria [15]. First, we confirmed the peroxisomal localization of TMEM135 in Huh7 and RPE1 cells. Although peroxisomal abundance, peroxisomal matrix protein import, and very-long-chain fatty acid (substrate for peroxisomal β-oxidation) levels were unaffected, TMEM135 depletion invariably reduced lysosome–peroxisome contact consequently leading to lysosomal cholesterol accumulation as previously reported [12]. TMEM135 depletion significantly reduced primary ciliogenesis in RPE1 cells. Surprisingly, TMEM135 depletion did not affect much of the early ciliogenesis steps such as CP110 loss from the M-centriole, transition zone formation, and CV formation as speculated to be defective during conditions for impaired intracellular cholesterol transport. Trafficking and activation of Rab8 was impaired suggesting CV extension was defective in TMEM135-depleted cells presenting a characteristic feature of impaired Rab8 function and not associated with the function of upstream regulators of ciliogenesis Rab11 and Rabin8. Moreover, Rab8 trafficking and activation was found to be closely associated with the distribution of cholesterol, as supplementation with the cholesterol–cyclodextrin complex rescued both Rab8 trafficking and its activation, thereby partially recovering the cilia phenotype. Taken together, our data suggest that disturbances in the intracellular cholesterol distribution are tightly associated with impaired Rab8 function and ciliogenesis.

## Results

### TMEM135 depletion results in lysosomal cholesterol accumulation and impairs SREBP2 processing

To determine the subcellular localization of TMEM135, Huh7 cells were transfected with Myc-tagged TMEM135 and subjected to immunostaining with peroxisomal marker, PMP70, or mitochondrial marker, Tomm20. Immunofluorescence observation revealed that TMEM135 was localized in peroxisomes (Appendix Fig S1A, upper panel), whereas Myc-tagged TMEM135 was not detected in the mitochondria (Appendix Fig S1A, lower panel). Co-transfection of Huh7 cells with Myc-tagged TMEM135 and RFP-SKL (peroxisomal targeting sequence-1, PTS1) confirmed peroxisomal localization of TMEM135 (Appendix Fig S1A, middle panel).

Similarly, ectopically expressed Myc-TMEM135 localized to peroxisomes and not mitochondria in RPE1 cells (Appendix Fig S1B, upper and lower panels). Moreover, endogenous TMEM135 completely localized to peroxisomes (Appendix Fig S1B, middle panel). Taken together, these data suggest that TMEM135 is a peroxisomal protein.

Although no functional domain has been identified for TMEM135 that would specify a relation to a known protein family, its role in peroxisome turnover has been speculated [13]. However, TMEM135 depletion in Huh7 cells did not affect the global levels of peroxisome marker protein PMP70 or catalase, suggesting that peroxisome abundance was unaffected (Appendix Fig S2A and B). Nevertheless, protein levels of Pex5 and Pex14, which play specific roles in the import of PTS1-containing matrix protein, were slightly decreased

(Appendix Fig S2B). Therefore, we hypothesized that TMEM135 might play a role in the import of matrix proteins and the metabolism of very-long-chain fatty acid (VLCFA). The colocalization between peroxisomal matrix protein, catalase, and a membrane protein, PMP70, suggested that TMEM135 depletion did not affect the import of peroxisomal matrix proteins in Huh7 cells and RPE1 cells (Appendix Fig S2C, E and F). Moreover, TMEM135 depletion did not alter the amount of VLCFA in Huh7 cells (Appendix Fig S2D).

Next, to investigate the role of TMEM135 in intracellular cholesterol transport, lysosome–peroxisome contact was analyzed in control and TMEM135-depleted RPE1 cells. TMEM135 depletion decreased the overlap between lysosomes and peroxisomes, as suggested by decreased contact between a lysosome marker LAMP1 and a peroxisome marker PMP70 in RPE1 cells (Fig 1A and B) without affecting lysosome and peroxisome numbers (Fig EV1A–D). We also confirmed that TMEM135 knockdown efficiently depletes TMEM135 fluorescent signal from peroxisomes indicating that TMEM135 depletion from peroxisomes reduces overlap between lysosomes and peroxisome without affecting peroxisome abundance (Appendix Fig S3). As expected, TMEM135 depletion resulted in intense filipin-positive vesicular structures that colocalized with lysotracker-positive structures, suggesting that TMEM135 depletion results in lysosomal cholesterol accumulation due to the physical dissociation between peroxisomes and lysosomes (Fig EV1E). The distribution of cholesterol in the control cells along the sucrose gradient was mainly elevated in the heavy fractions containing cell membrane ($Na^+/K^+$ ATPase) and Golgi (GM130) (numbers 9–12) (Fig 1C and D). In contrast, TMEM135-depleted cells had reduced cholesterol levels in the fractions of plasma membrane and Golgi, whereas the cholesterol level was mainly elevated in the lysosomal (LAMP1) fraction (numbers 4–7). However, cholesterol content in the fractions enriched with early endosomes (EEA1) and recycling endosomes (Rab11) was not altered by TMEM135 depletion (Fig 1C and D). These data suggest that TMEM135 depletion results in lysosomal cholesterol accumulation, thereby disturbing the subcellular distribution of cholesterol.

To further validate the functional impact of lysosomal cholesterol accumulation in TMEM135-depleted cells, we evaluated the cleavage of sterol response element-binding protein 2 (SREBP2), an ER-resident protein that undergoes cleavage and nuclear translocation in response to a low intracellular cholesterol level for transcription of cholesterol biosynthesis-related genes. In control cells, the presence of LDL prevented the nuclear translocation of SREBP2, suggesting that free cholesterol derived from LDL cholesterol was efficiently transported to ER and inhibited SREBP2 cleavage. However, SREBP2 processing was not blocked by LDL in TMEM135-depleted cells, suggesting that the cholesterol derived from LDL was not transported to ER (Fig 1E). Moreover, the expression levels of SREBP2 target genes, including *HMGCR, HMGCS, INSIG1*, and *LDLR*, were decreased in the presence of LDL in control cells but not in TMEM135-depleted cells (Fig 1F). Taken together, these results demonstrate that TMEM135 depletion impairs intracellular cholesterol transport by preventing lysosome–peroxisome membrane contact.

### TMEM135 depletion impairs ciliogenesis through disruption of intracellular cholesterol distribution

To examine whether intracellular cholesterol transport affects ciliogenesis, TMEM135 depletion was also performed in RPE1 cells and

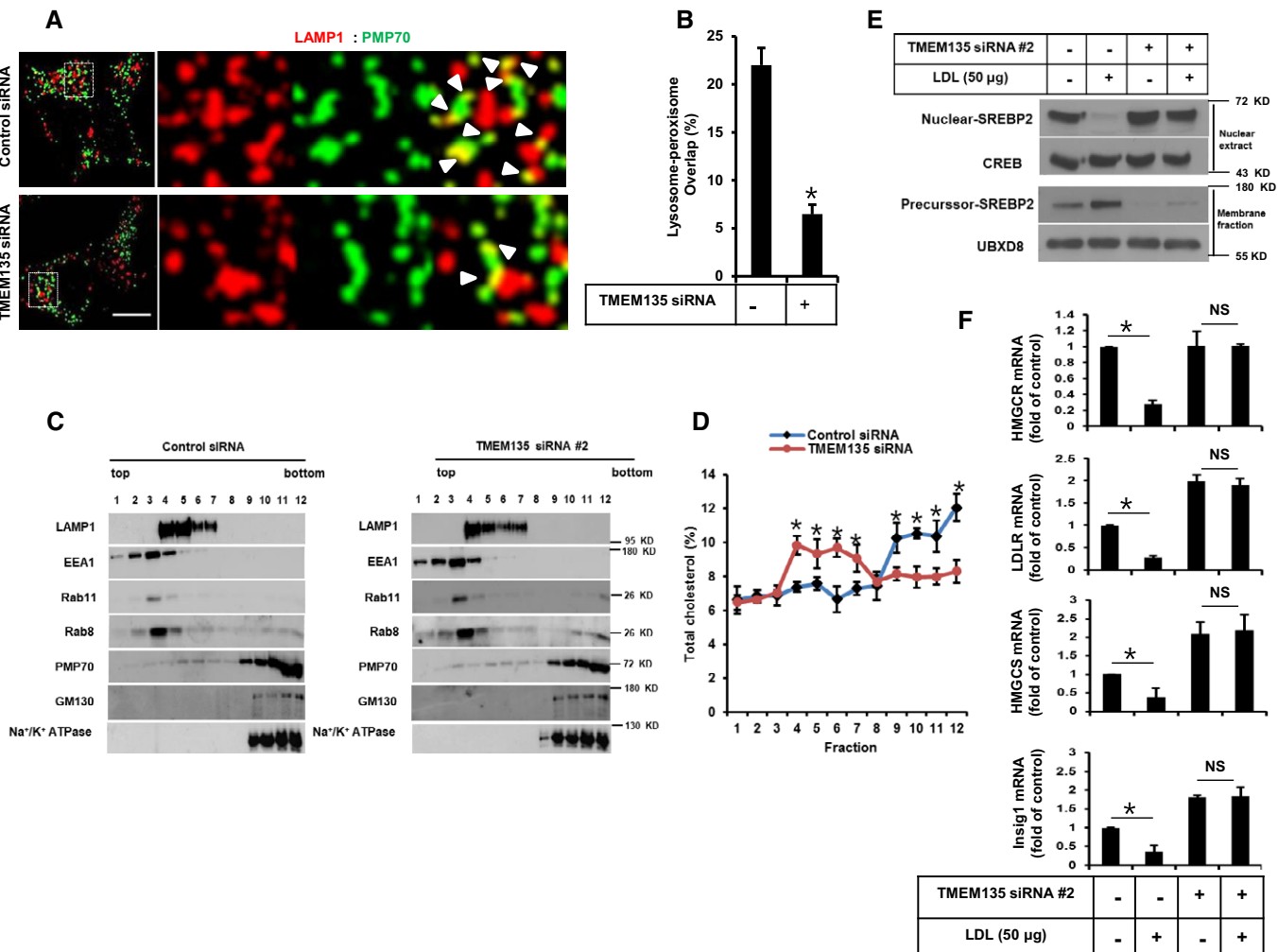

**Figure 1. Depletion of TMEM135 results in cholesterol accumulation in lysosomes.**

A   RPE1 cells were transfected with either scramble or TMEM135 siRNAs and immunostained for LAMP1 as a lysosome marker (red) and PMP70 as a peroxisome marker (green). Scale bar, 10 μm.

B   Quantification of overlap signal between lysosomes and peroxisomes shown in (A). Data represent mean ± SD (n = 3 experiments), and 50 cells were scored per condition per experiment. *P < 0.05, Student's t-test.

C   Subcellular fractions of a discontinuous sucrose gradient collected from top to bottom, separated by SDS–PAGE, and immunoblotted for LAMP1, EEA1, Rab11, Rab8, PMP70, GM130, and Na⁺/K⁺ ATPase.

D   Quantification of total cholesterol in the subcellular fractions shown in (C). Data represent mean ± SD (n = 3 experiments), *P < 0.05, Student's t-test.

E   Cells transfected with TMEM135 siRNA were subjected to a SREBP2 cleavage assay for the precursor of SREBP2 and nuclear SREBP2.

F   Cells transfected with TMEM135 siRNA were subjected to qPCR. Data represent mean ± SD (n = 3 experiments); *P < 0.05, Student's t-test.

Source data are available online for this figure.

the percentage of ciliated cells was determined using ARL13B as a cilia marker. As expected, all small interfering RNAs (siRNAs) targeting TMEM135 significantly reduced the percentage of ciliated cells, suggesting a functional coupling between lysosomal cholesterol accumulation and ciliogenesis (Fig 2A and B). Next, to examine whether removal of the accumulated cholesterol in lysosome could rescue ciliogenesis in TMEM135-depleted RPE1 cells, we performed a rescue experiment for ciliogenesis using hydroxypropyl-β-cyclodextrin (HPβCD), which is known to cause a dose-dependent reduction in cholesterol accumulation in NPC1 fibroblast cells [16,17]. As shown in Fig 2C, TMEM135 depletion was capable of accumulating cholesterol in lysosomal compartment even in

serum starvation which did not have exogenous source of LDL cholesterol, suggesting the gradual accumulation of cholesterol before subjecting the cells to serum starvation. Treatment with 0.5% HPβCD for 18 h under a serum-starvation condition cleared the accumulated cholesterol in TMEM135-depleted cells. However, the removal of accumulated cholesterol did not rescue ciliogenesis in TMEM135-depleted RPE1 cells (Fig 2D and E) as cholesterol depletion with cyclodextrin from the cell could negatively affect ciliogenesis [18].

We thus postulated that an uneven intracellular cholesterol distribution imposed by TMEM135 depletion-mediated impaired cholesterol transport might be causing the defective ciliogenesis.

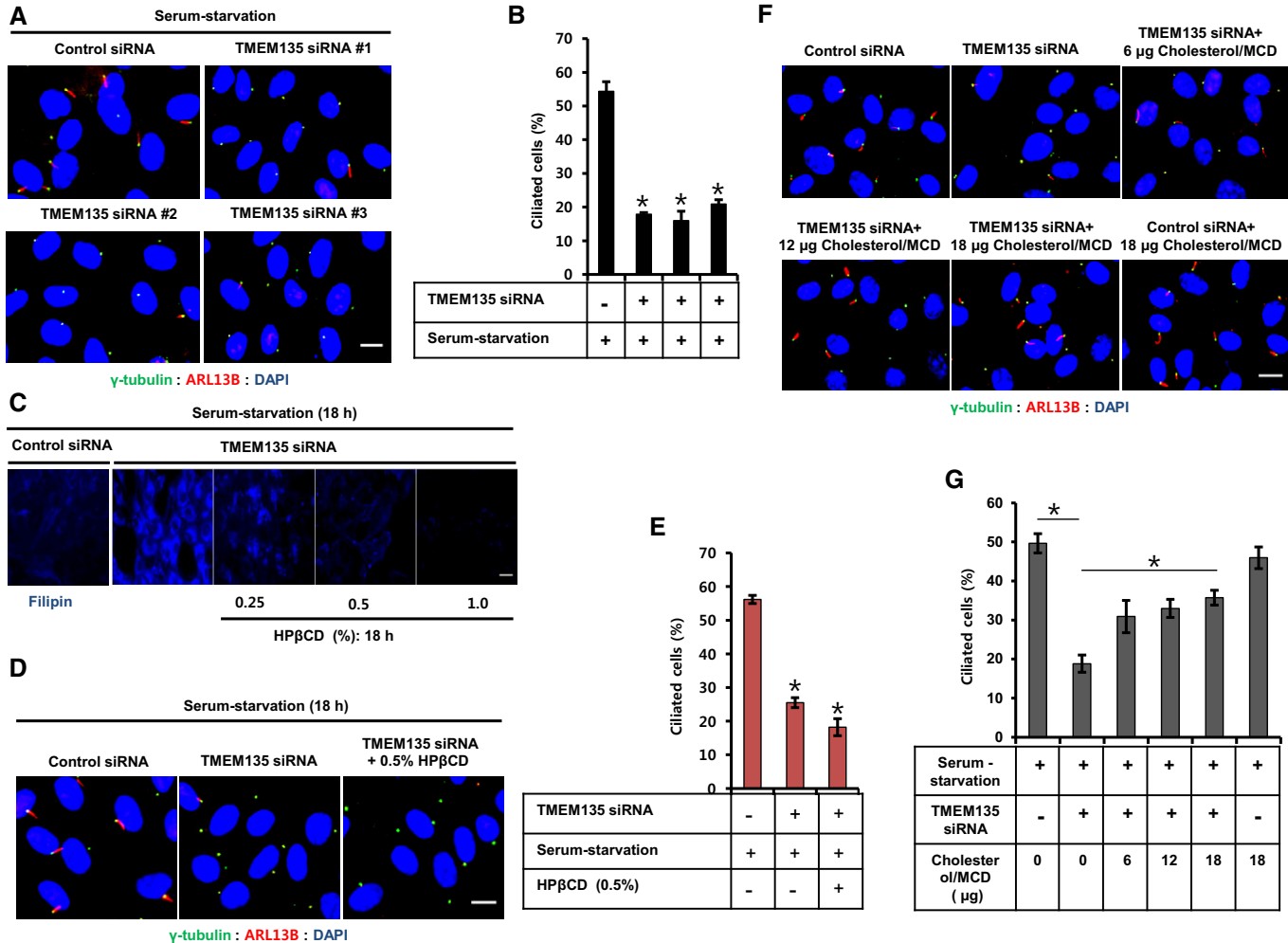

**Figure 2. Depletion of TMEM135 impairs ciliogenesis through disruption of intracellular cholesterol distribution.**

A   RPE1 cells were transfected with siRNAs as indicated, followed by serum starvation for 24 h, and immunostained for ARL13B (red) and γ-tubulin (green). Scale bar, 10 μm.

B   Quantification of the percentage of ciliated cells shown in (A). Data represent mean ± SD (n = 3 experiments), and 250 cells were scored per condition per experiment; *P < 0.05, Student's t-test.

C   Cells were transfected with siRNAs as indicated and incubated with HPβCD in serum-starvation media followed by filipin staining. Scale bar, 20 μm.

D   Cells were transfected with siRNAs shown in (C) and immunostained for ARL13B (red) and γ-tubulin (green). Scale bar, 10 μm.

E   Quantification of the percentage of ciliated cells shown in (D). Data represent mean ± SD (n = 3 experiments), and 250 cells were scored per condition per experiment; *P < 0.05, Student's t-test.

F   Cells were transfected with siRNAs as indicated, incubated with cholesterol–MβCD complex in serum-starvation media, and immunostained for ARL13B (red) and γ-tubulin (green). Scale bar, 10 μm.

G   Quantification of the percentage of ciliated cells shown in (F). Data represent mean ± SD (n = 3 experiments), and 250 cells were scored per condition per experiment, *P < 0.05, Student's t-test.

MCD/cholesterol is known to impact intracellular cholesterol and rescue ciliogenesis in cholesterol-depleted condition [18,19]. To test this possibility, cholesterol was added in a complex with methyl-β-cyclodextrin (MβCD), which likely delivers cholesterol to cells without passing through an intracellular cholesterol transport route such as lysosomes. Cholesterol–MβCD complex rescued ciliogenesis in TMEM135-depleted cells in a dose-dependent manner (Fig 2F and G). Taken together, our data suggest that proper distribution of intracellular cholesterol is an important requirement for induction of primary ciliogenesis.

**TMEM135 depletion impairs ciliogenesis at Rab8 trafficking and IFT20 recruitment to the centrioles**

To examine the role of TMEM135 on cilia formation, we analyzed the early steps of ciliogenesis in TMEM135-depleted RPE1 cells. First, we analyzed CP110 loss from the M-centriole, which is an essential early step of ciliogenesis for basal body formation [4,20]. TMEM135 depletion did not prevent CP110 loss from the M-centriole in a serum-starvation condition (Fig EV2A and B), indicating that TMEM135 depletion would not affect the basal body or

DAV [4]. Furthermore, TMEM135 depletion did not affect the formation of the ciliary transition zone, suggesting no influence on CV assembly or formation (Fig EV2C) [4].

TMEM135 depletion also did not affect Rab11 or Rabin8 localization to the centrioles (Fig EV3A–C). Serum starvation significantly enhanced the percentage of cells with Rab8 localized to the centrioles, whereas TMEM135 depletion resulted in morphologically aberrant Rab8 vesicles, which failed to localize to the centrioles in serum-starved conditions (Fig 3A and B). Although Rab8 has been reported to localize to cilia, we only detected a few cells with endogenous Rab8 on cilia (Appendix Fig S4). However, the base of cilia overlapped with the Rab8 signal, which accumulated near the centriolar region (Fig EV4A). These results suggested that TMEM135 depletion possibly impairs ciliogenesis at the CV extension step due to defective Rab8 trafficking at the centrioles.

IFT20 localization at the centrioles was also enhanced by serum starvation, which was prevented by TMEM135 depletion (Fig 3C

and D). Impairment of IFT20 recruitment to the centriole during serum starvation is suggestive of impaired ciliary vesicle formation [4,21]. As shown in Fig EV4B, IFT20 was abundant in the centrioles in serum-starved control cells and the base of cilia localizes to basal body-associated IFT20. Even though relatively less IFT20 localized to the basal body, smoothened vesicles, which are considered as CV [4,21], still localized to the basal body-associated IFT20, suggesting that ciliary vesicle formation might not be affected by TMEM135 depletion.

EHD1 is an essential protein required for fusion of preciliary vesicles into single CV, and it has been used as a CV marker along with other CV membrane proteins such as Rab8, Smo, and ARL13B [4,22]. It has been reported that single EHD1 vesicle localizes with Smo at the distal end of the basal body in Rab8-depleted cells [4]. Moreover, Wu et al [22] have successfully shown that EHD1 is localized in the CV during early hours of serum starvation, while localization of EHD1 at CV is disrupted when upstream regulator of CV

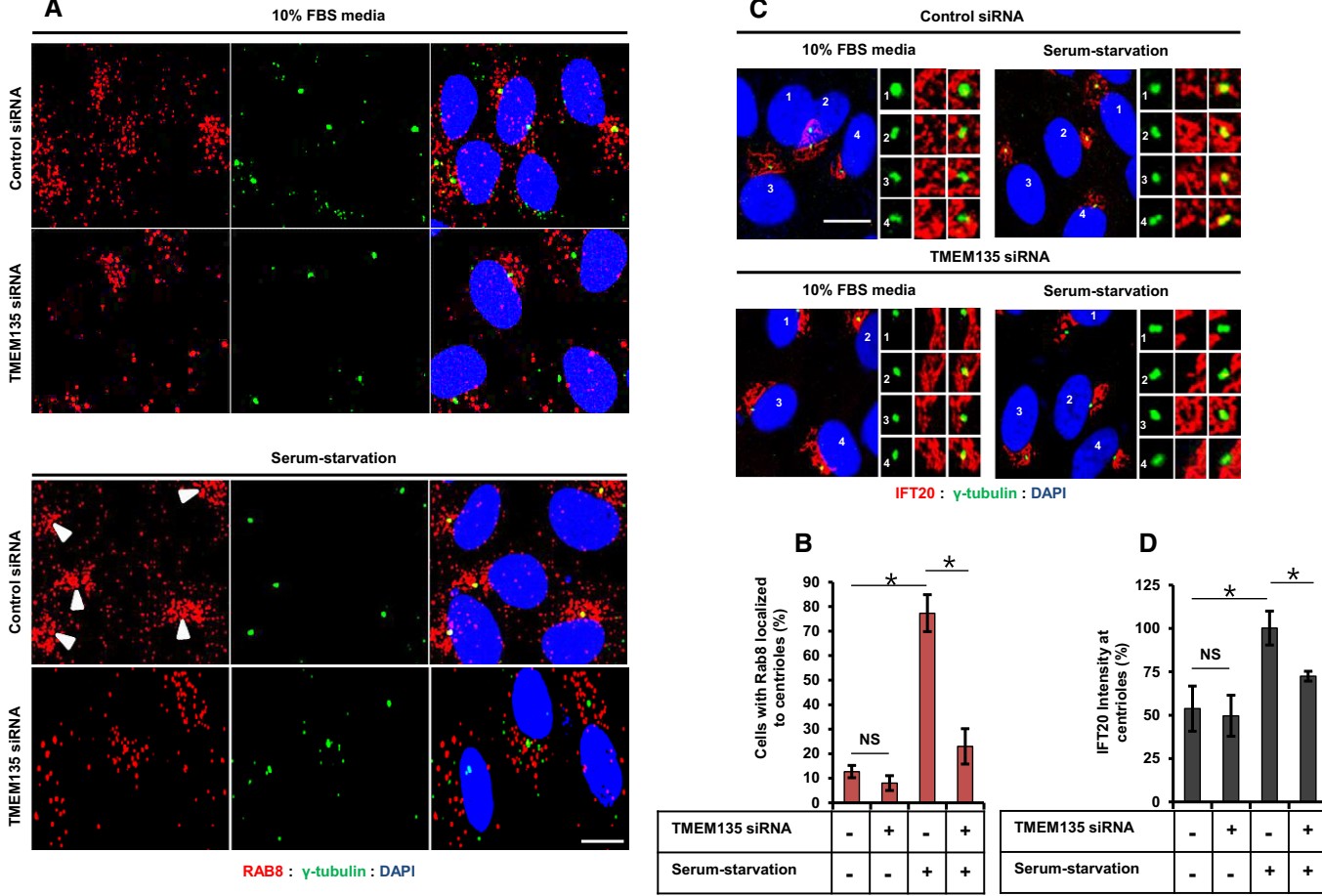

**Figure 3. Depletion of TMEM135 impairs both trafficking and enhanced IFT20 recruitment to centrioles.**

A  RPE1 cells were transfected with siRNA as indicated, and immunostained for Rab8 (red) and γ-tubulin (green) in either 10% FBS medium (upper panel) or serum-starvation medium (lower panel). Scale bar, 10 μm. Arrowhead indicates Rab8 localized to the centrioles.

B  Quantification of the percentage of cells with Rab8 localized to the centriole shown in (A). Data represent mean ± SD ($n = 3$ experiments), and 200 cells were scored per condition per experiment; *$P < 0.05$, Student's $t$-test.

C  Cells were transfected with siRNA shown in (A) and immunostained for IFT20 (red) and γ-tubulin (green). Scale bar, 10 μm.

D  Quantification of the percentage of IFT20 intensity at the centrioles in (c). Data represent mean ± SD ($n = 3$ experiments), and 150 cells were scored per condition per experiment; *$P < 0.05$, Student's $t$-test.

formation, Myo-Va, was depleted. Therefore, the presence of a single EHD1 vesicle at the distal end of the basal body in any given condition could suggest that CV formation is intact. In TMEM135-depleted cells, single EHD1 vesicle is found to localize at the distal end of the basal body, while EHD1 is localized in the cilium in the control cells suggesting that TMEM135 depletion did not prevent CV formation at the basal body but prevents extension of CV (Fig EV4C and D).

Taken together, our data suggest that impaired cholesterol transport due to TMEM135 depletion does not affect most of the early ciliogenesis steps as speculated; however, it specifically affects

ciliogenesis at the point of Rab8 trafficking to the centrioles, thereby affecting CV extension.

### Rab8 activation is compromised in TMEM135-depleted cells

Since Rab8 trafficking was identified as one of the major defects associated with ciliogenesis in TMEM135-depleted cells, we next performed a rescue experiment using transient expression of wild-type (WT) and mutant Rab8 plasmids with constitutively active (CA) Rab8 GTPase (Rab8Q67L, a GTP hydrolysis-deficient variant) and a

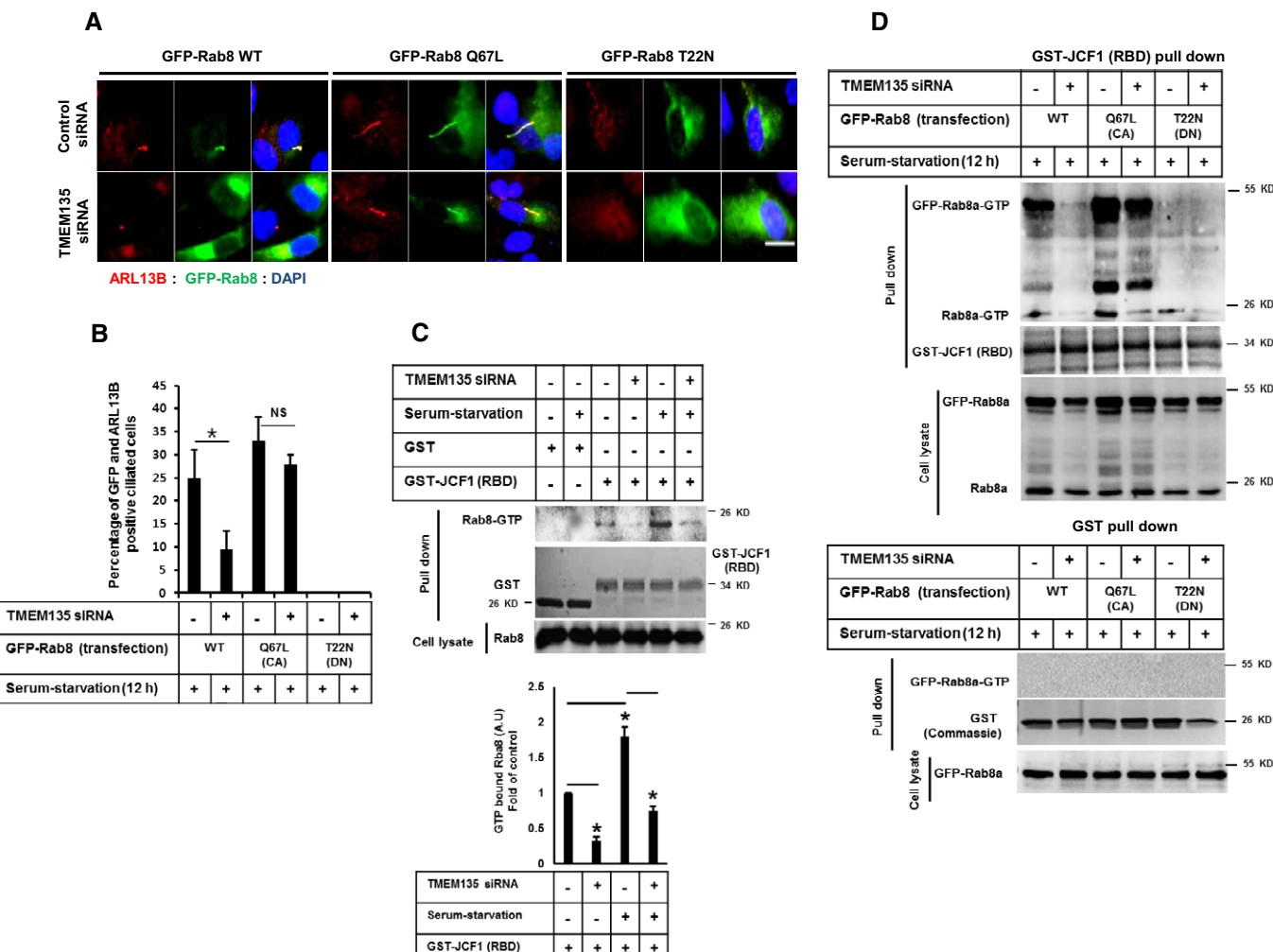

**Figure 4.  Ectopic expression of a constitutively active form of Rab8 recovers ciliogenesis in TMEM135-depleted cells.**

A   RPE1 cells were transfected with siRNAs as indicated, followed by transfection with wild-type pGFP-Rab8a (WT-Rab8), constitutively active pGFP-Rab8a (Q67L) (CA-Rab8), or DN Rab8 dominant-negative pGFP Rab8a (T22N), incubated in serum-starved media for 12 h, and immunostained for ARL13B (red), GFP-Rab8 (green), and DAPI (blue). Scale bar, 10 μm.

B   Quantification of the percentage of ciliated cells shown in (A). Data represent mean ± SD (n = 3 experiments), and 200 GFP-positive cells were scored per condition per experiment; *P < 0.05, Student's t-test.

C   (Upper panel) Cells were transfected as indicated and the cell lysates were incubated with purified proteins, including GST or GST-JCF1 (RBD). The amount of GTP-Rab8 bound to GFT-JCF1(RBD) was analyzed by Western blot with Rab8 antibody. (Lower panel) Intensity of the bands was quantified by ImageJ software. The amount of GTP-Rab8 was normalized to the control level. Bar graph represents mean ± SD (n = 3 experiments). *P < 0.05, Student's t-test.

D   Cells were transfected as shown in (A), and cell lysates were incubated with purified GST-JCF1 (RBD) fusion protein. The amount of GTP-Rab8 bound to GFT-JCF1(RBD) was analyzed by Western blot with Rab8 antibody. (Lower panel) Cells were transfected as shown in (A), and cell lysates were incubated with purified GST protein. The amount of GTP-Rab8 bound to GFT-JCF1(RBD) was analyzed by Western blot with Rab8 antibody.

Source data are available online for this figure.

dominant-negative (DN) form of Rab8 GTPase (Rab8T22N, GDP-locked variant). While the ectopic expression of WT Rab8a and DN Rab8a was not efficient to induce ciliogenesis in TMEM135-depleted cells, ectopic expression of CA Rab8 resulted in the recovery of ciliogenesis (Fig 4A and B). The inability of WT Rab8 overexpression to induce ciliogenesis is suggestive of impaired Rab8 activation in TMEM135-depleted cells. Synaptotagmin-like protein 1 (JCF1) is a Rab8 effector known to specifically interact with a GTP-bound form of Rab8 [23–25]. Therefore, we used GST-JCF1 (containing the Rab8-binding domain of JCF1) purified from bacteria to pull down GTP-bound Rab8 from RPE1 cell lysates. A higher level of Rab8 was pulled down by GST-JCF1 from serum-starved control cells compared with that obtained from serum-fed control cells, suggesting that Rab8 activation was required for ciliogenesis (Fig 4C). In contrast, TMEM135 depletion decreased Rab8-GTP level bound with GST-JCF1 in both serum-fed and serum-starved conditions. Next, we analyzed the activation state of both WT Rab8 and mutant Rab8 in TMEM135-depleted cells (Fig 4D). In agreement with the recovery of cilia by CA Rab8, the amount of Rab8-GTP pull-down by GST-JCF1 was similar in both control and TMEM135-depleted cells. In contrast, the activation of WT Rab8 was still impaired in TMEM135-depleted cells. DN Rab8 was not activated as it is a GDP-locked variant.

Since CA GFP-Rab8 could be activated and rescued cilia formation while WT GFP-Rab8 could not be activated in TMEM135-depleted cells, we hypothesized that impaired ciliary vesicle extension observed in TMEM135-depleted cells could be associated with Rab8 rather than its upstream such as Rab11 and Rabin8. It has been reported that depletion of Rab11 and Rabin8 also prevents CV extension but does not prevent CV formation [22]. We also confirmed that depletion of Rab11 and Rabin8 did not prevent CV formation as suggested by the presence of single EHD1 vesicle localized at the distal end of mother centriole (Appendix Fig S5A–D). To exclude the possibility of involvement of Rab8 upstream regulators, we performed the double knockdown of TMEM135 and Rabin8 carried out GST-JCF1 pull-down experiment for analyzing the activation state of CA GFP-Rab8. It is well known that Rabin8 is a guanine nucleotide exchange factor (GEF) for Rab8 which is required GDP release and GTP loading on Rab8. As shown in Appendix Fig S5E, the activation of CA GFP-Rab8 was prevented in Rabin8-depleted cells as well as in the cells depleted of both TMEM135 and Rabin8. These data indicate that Rabin8 was required for GTP loading on CA GFP-Rab8 and the function of Rabin 8 is intact in TMEM135-depleted cells which allow the activation of CA GFP-Rab8 (Fig 4D and Appendix Fig S5E). Thus, it is interesting that activation of endogenous Rab8 was impaired in TMEM135-depleted cells as compared to that of control cells under all conditions examined. Taken together, our data suggest that TMEM135 depletion impairs the activation of Rab8.

**Impaired ciliogenesis under TMEM135 depletion is not associated with IFT20**

It has been reported that enhanced IFT20 localization at the M-centriole is essential for CV formation [4]. Since IFT20 abundance was found to be decreased at the centrioles without affecting CV formation, we investigated whether decreased IFT20 recruitment at the centrioles affects both ciliogenesis and Rab8 trafficking in TMEM135-depleted cells. Cytosolic IFT20 has been documented during the transport of ciliary proteins from Golgi to the base of the primary cilium [26,27]. TMEM135 depletion did not alter the cytosolic abundance of IFT20 (Fig 5A). Although IFT20 depletion is well known to reduce primary ciliogenesis, it did not affect Rab8 trafficking, suggesting that Rab8 trafficking to the centrioles is independent of IFT20 (Fig 5B–D). Next, to investigate whether overexpression of IFT20 restores the impaired ciliogenesis by TMEM135 depletion, a Flag-IFT20 construct was ectopically expressed in the cells, and the percentage of ciliated cells with both ARL13B and IFT20 localized to the cilia was determined. As shown in Fig 5E and F, whereas ectopically expressed IFT20 could be visualized in the cilium of control cells, the percentage of such ciliated cells was significantly decreased in TMEM135-depleted cells despite abundant expression of Flag-IFT20. Moreover, CA Rab8 could not recover ciliogenesis in IFT20-depleted cells, suggesting that IFT20 function is mandatory for cilia formation (Fig 5G and H). Interestingly, double knockdown of both IFT20 and TMEM135 also prevented cilia recovery in cells transiently expressing CA Rab8 (Fig 5G and H). These data suggest that impaired ciliogenesis associated with TMEM135 depletion is not related to function or abundance of IFT20, rather very specific to Rab8 trafficking and activation.

**Enhanced IFT20 localization to the centrioles depends on Rab8 activation**

Since our results indicated that impaired Rab8 function might directly affect the enhanced recruitment of IFT20 to the centrioles in TMEM135-depleted cells, we next examined the ciliogenesis and IFT20 localization to the centrioles in Rab8-depleted cells. In humans, there are two isoforms of Rab8, Rab8a and its paralogue Rab8b, with 80% sequence homology. Rab8a is ubiquitously expressed, whereas Rab8b is mostly expressed in the spleen, testis, and brain. Both isoforms are known to play similar roles in vesicular trafficking from Golgi to plasma membrane [28,29]. Double knockdown of Rab8a and Rab8b significantly decreased the percentage of ciliated cells (Fig 6A–C). We confirmed that Rab8 depletion did not affect ciliary vesicle formation as EHD1-positive single ciliary vesicle (CV) could be observed at the distal end of the basal body in most of the Rab8-depleted cells (Fig 6D and E). Rab8 depletion also decreased the enhanced IFT20 localization to the centrioles (Fig 6F and G) even though Rab8 depletion did not affect CV formation. This is contradictory to a previous report showing that Rab8 depletion did not affect IFT20 recruitment to the basal body [4].

Rab8 is a master regulator of ciliary protein trafficking [30], and its association with IFT20 has been documented in T cells and zebrafish [31,32]. However, the interaction between Rab8 and IFT20 has not been assessed in RPE1 cells to date. To address the possibility that Rab8 is involved in the targeted delivery of IFT20, immunoprecipitation was carried out following the transient expression of WT Rab8 and mutant Rab8. As shown in Fig 6H–J, both WT Rab8a and CA Rab8a were found to interact with IFT20 in control cells. However, only CA Rab8a but not WT Rab8a interacted with IFT20 in TMEM135-depleted cells. Furthermore, no significant interaction was observed between IFT20 and DN Rab8a. These data suggest that Rab8 activation is required for the targeted delivery of IFT20 to the centrioles. Moreover, impaired Rab8 activation is a key factor responsible for preventing enhanced IFT20 recruitment to the centrioles in TMEM135-depleted cells.

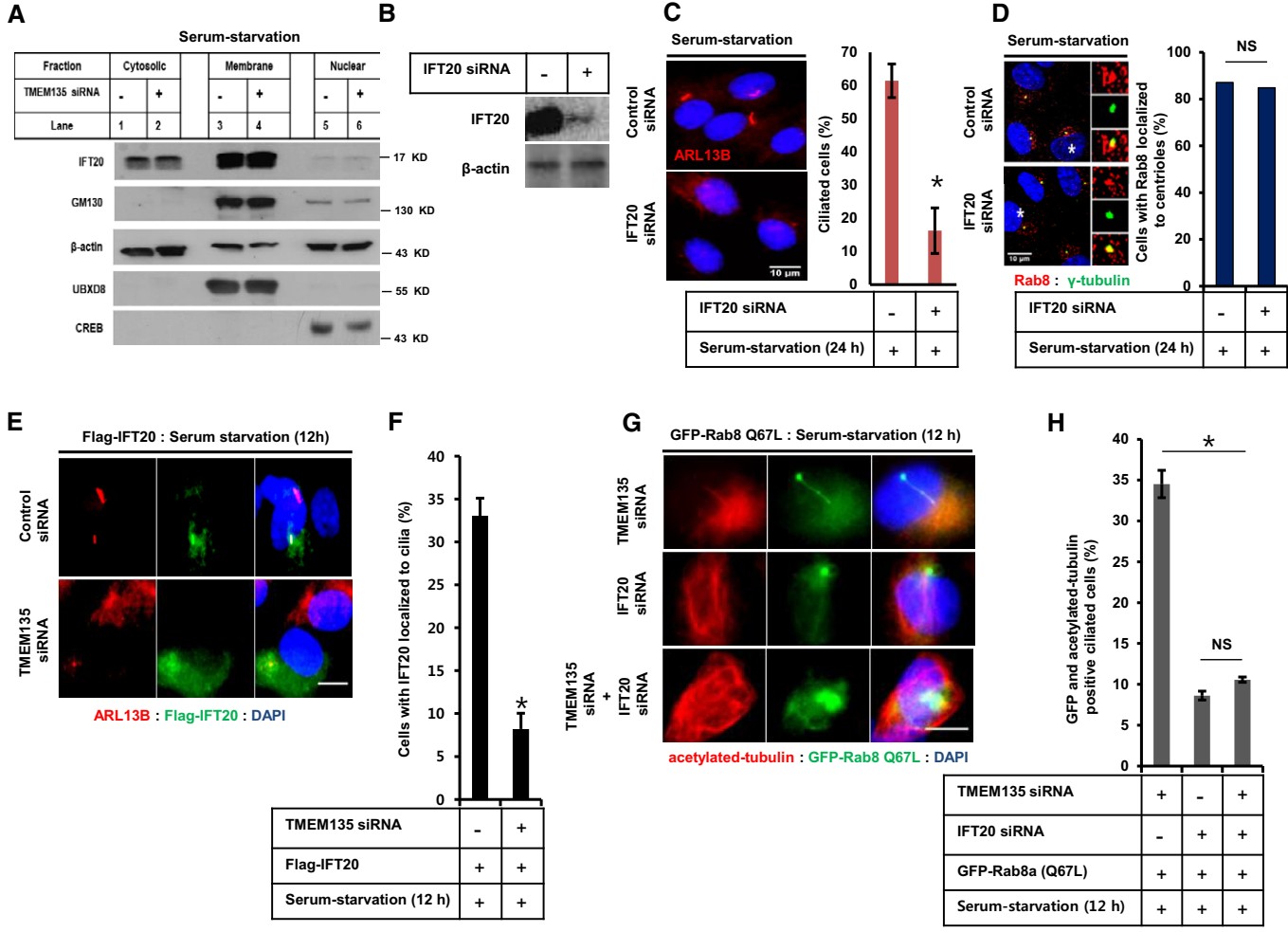

**Figure 5. Malfunctioned ciliogenesis observed during TMEM135 depletion is not associated with IFT20.**

A   RPE1 cells were transfected with siRNAs as indicated, followed by 24-h incubation in serum-starved media. Cells were then and subjected to fractionation, and Western blot for IFT20, the Golgi marker GM130, membrane marker UBXD8, and nuclear marker CREB.

B   Efficiency of IFT20 knockdown by Western blot.

C   Cells were transfected with siRNAs as indicated, followed by incubation in serum-starved media for 24 h, and immunostained for ARL13B (red). Scale bar, 10 μm. The bar graph represents the quantification of the percentage of ciliated cells. Data represent mean ± SD (*n* = 3 experiments), and 250 cells were scored per condition per experiment; \**P* < 0.05, Student's *t*-test.

D   Cells were transfected as shown in (C), and immunostained for Rab8 and γ-tubulin, followed by quantification of the percentage of cells with Rab8 localized to the centriole. Data represent average (*n* = 2 experiments).

E   Cells were transfected with siRNAs as indicated, followed by transfection with Flag-IFT20, incubated in serum-starvation media for 12 h, and immunostained for ARL13B. Representative fluorescent images of Flag-IFT20 (green), ARL13B (red), and DAPI (blue) are shown. Scale bar, 10 μm.

F   Quantification of the percentage of ciliated cells with both the Flag-IFT20 and ARL13B localized in the cilium. Data represent mean ± SD (*n* = 3 experiments), and 150 Flag-positive cells were scored per condition per experiment; \**P* < 0.05, Student's *t*-test.

G   Cells were transfected with siRNAs as indicated, followed by further transfection with CA-Rab8, incubated in serum-starvation media for 12 h, and immunostained for acetylated tubulin. Representative fluorescent images of GFP-Rab8 Q67L (green), acetylated tubulin (red), and DNA (blue) are shown. Scale bar, 10 μm.

H   Quantification of the percentage of GFP-positive ciliated cells (only those cilia having both GFP-Rab8 and acetylated tubulin on cilium were considered for quantification). Data represent mean ± SD (*n* = 3 experiments), and 200 GFP-positive cells were scored per condition per experiment, \**P* < 0.05, Student's *t*-test.

Source data are available online for this figure.

## The cholesterol–MβCD complex rescues impaired Rab8 trafficking to the centrioles and Rab8 activation, and increases IFT20 intensity at the centrioles in TMEM135-depleted cells

Finally, we evaluated whether trafficking and activation of Rab8 could be modulated by cholesterol extraction with HPβCD or through the supplementation of cholesterol with cholesterol–MβCD complex in TMEM135-depleted cells. HPβCD-mediated cholesterol extraction did not rescue trafficking, activation, and aberrant morphology of Rab8 (Fig EV5A–C). These observations are consistent with our findings that cholesterol extraction did not rescue ciliogenesis in TMEM135-depleted cells (Fig 2C–E). Interestingly, cholesterol–MβCD complex rescued trafficking and activation of Rab8 in TMEM135-depleted cells (Fig 7A–C).

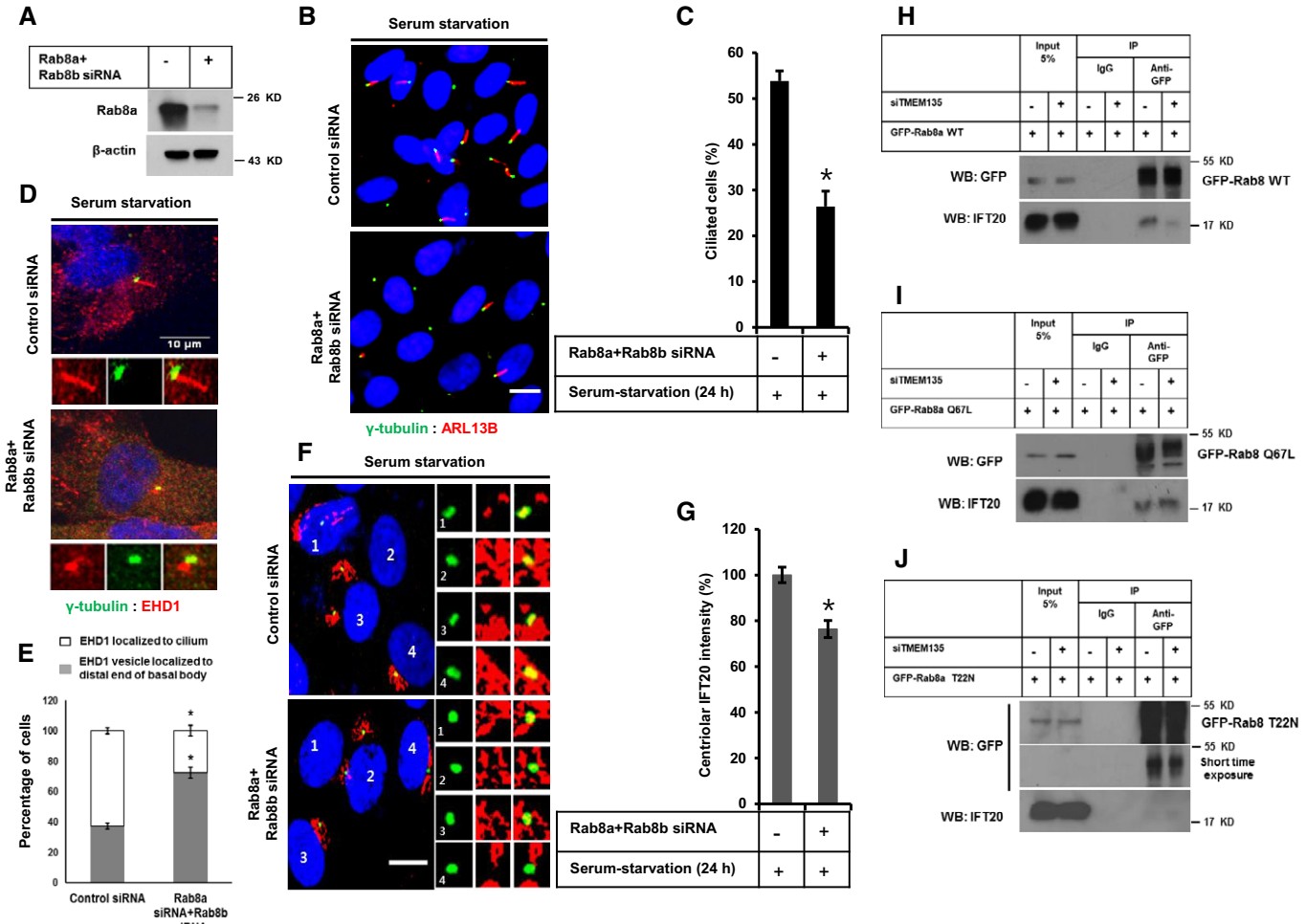

**Figure 6. Enhanced IFT20 intensity induced by serum starvation at centrioles depends on Rab8 activation.**

A   Efficiency of Rab8a depletion confirmed by Western blot in RPE1 cells.

B   RPE1 cells were transfected by siRNAs as indicated, followed by incubation in serum-starvation media for 24 h, and immunostained for ARL13B (red) and γ-tubulin (green). Scale bar, 10 μm.

C   Quantification of the percentage of ciliated cells shown in (B). Data represent mean ± SD (n = 3 experiments), and 250 GFP-positive cells were scored per condition per experiment; *P < 0.05, Student's t-test.

D   Cells were transfected by siRNA as indicated followed by incubation in serum-starvation media for 24 h, and immunostained for EHD1 (red) and γ-tubulin (green). Scale bar, 10 μm.

E   Quantification of the percentage of cells with EHD1 in cilium or in the distal end of basal body as shown in (D). Data represent mean ± SD (n = 3 experiments), and 150 cells were scored per condition per experiment; *P < 0.05, Student's t-test.

F   Cells were transfected by siRNAs as indicated, followed by incubation in serum-starvation media for 24 h, and immunostained for IFT20 (red) and γ-tubulin (green). Scale bar, 10 μm.

G   Quantification of the percentage of IFT20 fluorescent intensity at the centriole shown in (F). Data represent mean ± SD (n = 3 experiments), and 150 were scored per condition per experiment; *P < 0.05, Student's t-test.

H–J   Cells were transfected by siRNAs as indicated, followed by transfection with GFP-Rab8 WT, GFP-Rab8 Q67L, or GFP-Rab8 T22N, and further incubated in serum-starvation media for 12 h. Cell lysate was subjected to immunoprecipitation with anti-GFP antibody, followed by Western blot with antibody against GFP.

Source data are available online for this figure.

Furthermore, IFT20 localization to the centrioles was also enhanced by cholesterol–MβCD supplementation in TMEM135-depleted cells (Fig 7D and E). Taken together, our data suggest that exogenous cholesterol supplementation rescues defective ciliogenesis through ameliorating trafficking and activation of Rab8, along with Rab8-dependent IFT20 delivery to the centriole (Fig 7F).

## Discussion

In this study, we identified a previously uncharacterized functional relationship between intracellular cholesterol transport and the primary ciliogenesis. Specifically, we found that the depletion of TMEM135 results in robust lysosomal cholesterol accumulation, which could be attributed to a decrease in lysosome–peroxisome

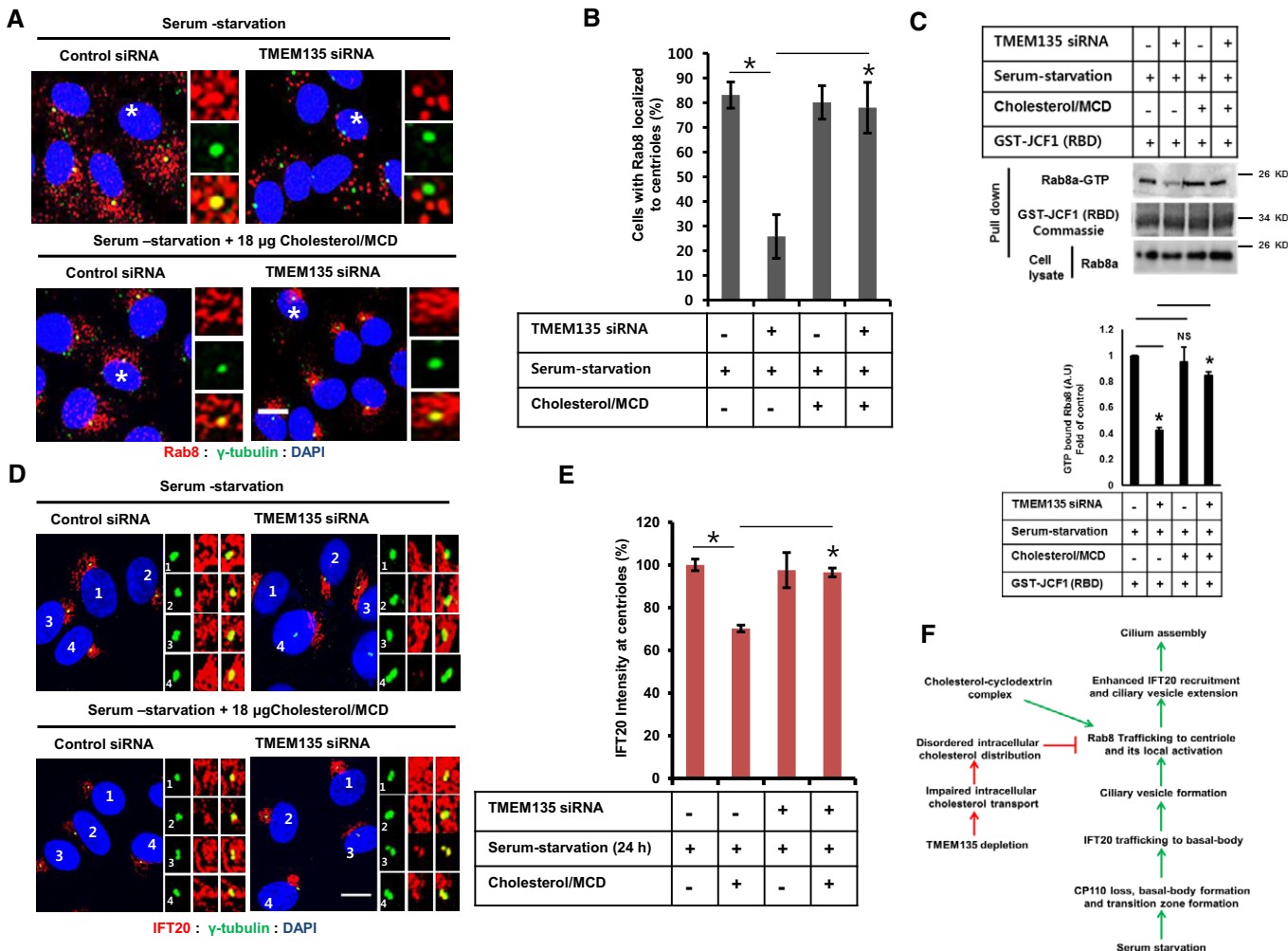

**Figure 7. Cholesterol–MβCD complex rescues impaired Rab8 trafficking to the centriole, augments Rab8-GTP binding, and increases IFT20 intensity at the centriole in TMEM135-depleted cells.**

A   RPE1 cells were transfected as indicated, followed by incubation in serum-starvation media in the presence or absence of cholesterol/MCD for 24 h, and then immunostained for Rab8 (red) and γ-tubulin (green). Scale bar, 10 μm. Representative magnified images are shown from cells labeled with white asterisks.

B   Quantification of the percentage of cells with Rab8 localized to the centriole shown in (A). Data represent mean ± SD (*n* = 3 experiments), and 200 cells were scored per condition per experiment; *P < 0.05, Student's *t*-test.

C   (Upper panel) Cells were transfected as shown in (A), and cell lysate was incubated with purified GST-JCF1 (RBD) fusion protein. The amount of GTP-Rab8 bound to GFT-JCF1(RBD) was analyzed by Western blot with Rab8 antibody. (Lower panel) The intensity of bands was quantified by ImageJ software. The amount of GTP-Rab8 was normalized to the control level. The bar graph represents mean ± SD (*n* = 3 experiments). *P < 0.05, Student's *t*-test.

D   Cells were transfected as shown in (A) and immunostained for IFT20 (red) and γ-tubulin (green). Scale bar, 10 μm.

E   Quantification of IFT20 intensity at the centriole shown in (D). Data represent mean ± SD (*n* = 3 experiments), and 150 cells were scored per condition per experiment; *P < 0.05, Student's *t*-test.

F   A proposed model showing the importance of cholesterol for primary ciliogenesis.

Source data are available online for this figure.

contact. Accordingly, TMEM135 depletion significantly suppressed ciliogenesis due to impaired trafficking and activation of Rab8. Moreover, enhanced IFT20 recruitment to the centrioles during serum starvation was found to be dependent on Rab8 activation. Since ciliogenesis was rescued by supplementation of cholesterol–MβCD complex, including trafficking and activation of Rab8 as well as enhanced IFT20 recruitment at the centrioles, our study suggests that the impaired ciliogenesis observed in TMEM135-depleted cells is related to disturbed intracellular cholesterol distribution.

Previous studies have suggested that cholesterol efflux from lysosome involves both vesicular and non-vesicular transport [33,34]. Cholesterol homeostasis was reported to be impaired in PEX2 knockout mice due to loss of peroxisomal compartmentalization, resulting in aberrant activation of SREBP2 and cholesterogenic enzymes [35]. LPMC was recently proposed as an alternative mechanism for cholesterol efflux from lysosomes and disrupting LPMC impaired LDL cholesterol transport to ER followed by aberrant SREBP2 activation [12]. Therefore, peroxisomes are critically

important for maintaining intracellular cholesterol homeostasis. Rescue of Rab8 trafficking by cholesterol–MβCD complex in TMEM135-depleted cells suggests that cholesterol content in the membranes of Rab8 vesicles might be important for the maintenance of its function and trafficking. Since inactive Rab8 was reported to preferentially localize to peroxisomes in Huh7 cells [36], it is reasonable to assume that a Rab8 vesicle might obtain cholesterol from peroxisomes to allow for its release from peroxisome and subsequent trafficking. However, we found that Rab8 activation was impaired by TMEM135 depletion, and only partial contact between endogenous Rab8 vesicles and peroxisome was increased in TMEM135-depleted cells rather than complete colocalization (Appendix Fig S6A and B). The increased overlap between Rab8 vesicles and peroxisome was not further enhanced by cholesterol extraction by HPβCD (Appendix Fig S6A and B). These data suggest that peroxisome may not directly affect the cholesterol content of Rab8 vesicles or Rab8 trafficking. Owing to technical limitations imposed by cholesterol measurement data to specifically determine the amount of cholesterol present on Rab8 vesicles (due to the presence of Rab11, lysosome, and EEA1 on the Rab8 fraction), we could not exclude other possibilities to explain the impaired Rab8 trafficking in TMEM135-depleted cells other than cholesterol content. Nevertheless, the rescue of cilia phenotype, Rab8 trafficking, and Rab8 activation by cholesterol–MβCD complex strongly suggests that TMEM135 depletion presents a situation that disturbs the intracellular cholesterol distribution. These observations indicate that recuperating the intracellular cholesterol content could reverse the aberrant phenotype of Rab8 vesicles. Although we could not exclude the possibility that the cholesterol content of Rab8 vesicles might directly affect its trafficking, precisely how cholesterol regulates Rab8 trafficking is still not clear. A previous report suggested that the accumulation of cholesterol in the lysosomal compartment disrupts the normal cholesterol distribution in other compartments, resulting in clustering of SNARE proteins that affect vesicle fusion events [37]. Furthermore, it is well known that the coordinated function of tethering factors, Rab proteins, and SNARE proteins is important for vesicle fusion events [38]. Therefore, it is likely that Rab8 trafficking along the cytoskeleton or its interaction with motor proteins and fusion events may be dependent on the intracellular cholesterol distribution.

Rab8 is the only known small Rab GTPase to localize to the basal body and cilium. It contributes to the membrane trafficking and tethering events associated with cilium biogenesis in coordination with Rabin8 and Rab11 [38]. However, the function of other Rab GTPases, especially those involved in cholesterol sorting of the endosomal compartments, might also be affected by impaired intracellular cholesterol transport; thus, the lack of influence of TMEM135 depletion on Rab11 and Rabin8 trafficking to the centrioles was somewhat surprising. The ability of CA Rab8 to recover ciliogenesis further suggests that the function of Rab11 and Rabin8 is intact in TMEM135-depleted cells, as dysfunctional Rab11 and Rabin8 are known to impair ciliary membrane biogenesis by preventing the localized activation of Rab8 at the centrioles (Appendix Fig S5E) [1]. The basal body-localizing protein Ahi1 is required for recruiting Rab8 to the basal body, and its malfunction is known to cause Joubert syndrome [30,39]. Silencing of Ahi1 could impair ciliogenesis due to defective recruitment of Rab8 to the basal body which could not be rescued by ectopic expression of CA

Rab8 [40]. Taken together, these previous reports suggest that TMEM135 depletion does not impair Ahi1 function, and thus, this possibility was not tested further in the present study.

Our results indicate that TMEM135 depletion possibly disrupts endogenous Rab8 trafficking, GTP binding capability, or GTPase activity, but not GTP loading by Rabin8 (Appendix Fig S5E). The fact that ectopic expression of CA Rab8a rescued the cilia phenotype implies that GTP loading from Rabin8 is still intact in TMEM135-depleted cells. Interestingly, results from the cholesterol–MβCD complex rescue experiment clearly suggest that TMEM135 silencing affects Rab8 trafficking to the centrioles, but not GTPase activity. If Rab8 GTPase activity had been compromised, localization of Rab8 to the basal body and partial rescue of the cilia would not be possible in TMEM135-depleted cells in response to cholesterol–MβCD complex. As cholesterol–MβCD complex rescued the ciliary phenotype, possibly by modulating Rab8 trafficking, we attempted to determine why only CA Rab8 and not WT Rab8 rescued the cilia phenotype in TMEM135-depleted cells. The ability of CA Rab8 to efficiently rescue the ciliogenesis is attributed to its inability to hydrolyze bound GTP. In contrast, WT Rab8 is not efficient at inducing ciliogenesis in TMEM135-depleted cells as it requires continuous cycles of GTP loading and hydrolysis, suggesting that TMEM135 disrupts Rab8 trafficking, thereby preventing its local activation at the centrioles.

IFT20 is a unique member of IFT family because of dual localization to Golgi and the centrioles [26]. IFT20 localization to the basal body has been reported to be associated with the early steps of ciliogenesis which is specifically required for ciliary vesicle formation [4]. We found that a considerable amount of IFT20 is localized at the centrioles, even in cells grown in the presence of serum, suggesting a possible role of IFT20 in cellular functions other than ciliogenesis. This idea is in accordance with the previous report using other cell types [41]. Serum starvation drastically increased the amount of IFT20 at the centrioles, which was prevented by TMEM135 depletion. Although enhanced IFT20 localization at the centrioles was prevented by TMEM135 depletion, ciliary vesicle formation remained unaffected. Thus, our study suggests that a defective ciliary phenotype is associated with CV extension, possibly due to defective Rab8 trafficking to the centrioles. We found that Rab8 depletion also reduced the enhanced IFT20 amount at the centrioles during serum starvation which was in contrast to the result from a previous report [4]. Since our data suggest an interaction between CA Rab8 and IFT20, it is reasonable to speculate that at least some portion of IFT20 localization at the centriole is dependent on Rab8 activation. Thus, it is likely that after ciliary vesicle formation, Rab8 activation-dependent IFT20 trafficking at the centrioles might be essential for the later steps of ciliogenesis. Rab8 is known to interact with ciliary proteins such as fibrocystin or its trafficking into cilia [42]. IFT20 has been reported to be essential for the transport of fibrocystin and polycystin from Golgi to the primary cilium [43]. Moreover, Elipsa links IFT20 to Rab8 in zebrafish [32]. However, the functions of Rab8 and IFT20 considerably overlap during early ciliogenesis; thus, further studies are required to elucidate the mechanism by which CA Rab8 interacts with IFT20 protein, IFT20 protein complex, or Golgi exocyst.

IFT20 and IFT88 belong to the IFTB complex, whose members are well known to interact with each other [23]. In

serum-supplemented cells, IFT88 was detected at the distal end of the M-centriole (Appendix Fig S7A–C) as previously reported [44]. Under serum starvation, IFT88 localized to the cilium. Although TMEM135 depletion had no effect on the IFT88 localization to the distal end of the basal body, its localization to cilia was significantly reduced, as TMEM135 depletion reduces primary cilia formation. These observations suggest the transport of IFT20 and IFT88 proteins at the site of ciliogenesis is differentially regulated although they belong to the same IFTB complex.

In summary, our study suggests a novel link between intracellular cholesterol distribution and ciliogenesis adding one more cellular physiological process that affects primary ciliogenesis. Disruption of cholesterol homeostasis due to impairment of intracellular cholesterol transport negatively affects ciliogenesis. Deficiency of the peroxisomal protein, TMEM135, prevents cholesterol-dependent Rab8 trafficking and ciliogenesis. Further, the behavior of Rab8 in response to the intracellular cholesterol distribution controls vesicle trafficking events associated with ciliogenesis. Future studies are required to evaluate the functional link between impaired cholesterol transport and ciliopathies in peroxisome dysfunction such as Zellweger syndrome.

# Materials and Methods

### Cell culture

RPE1-Smo-EGFP (a generous gift from Joon Kim, KAIST, Korea), RPE1, and Huh7 cells were cultured in high-glucose Dulbecco's modified Eagle medium (DMEM; Gibco-BRL, Grand Island, NY, USA) supplemented with 10% fetal bovine serum (FBS; Gibco-BRL), 100 IU/ml penicillin, and 100 µg/ml streptomycin at 37°C and 5% $CO_2$ in a humidified atmosphere. All cell lines were validated to be negative for mycoplasma contamination.

### siRNA and plasmid transfection

siRNA transfection was performed with Lipofectamine RNAiMAX (Invitrogen) according to the manufacturer's transfection protocol. Cells were fixed/harvested 48–72 h after transfection. The siRNA sequences used in this study are provided in Appendix Table S1. Plasmid transfection was performed with Lipofectamine 3000 reagent (Invitrogen) according to the manufacturer's transfection protocol. Cells were fixed/harvested 12–24 h after transfection.

### Plasmids

The plasmid pCMV-TMEM135-Myc encoding full-length versions of wild-type mouse TMEM135 followed by five tandem copies of a c-Myc epitope tag (EQKLISEEDL) under control of the cytomegalovirus (CMV) promoter was generated by ligating *Hind* III and *Nhe* I-cleaved vector pcDNA3.1-(Myc)5 [45] with PCR fragments containing full-length TMEM135. Amplification was performed using primers containing *Hind* III and *Nhe* I overhang with a mouse liver cDNA library as templates. The pRFP-SKL plasmid was constructed by inserting SKL, followed by a STOP codon, into the reading frame in the TagRFP vector [46]. Human wild-type pGFP-Rab8A (Plasmid #24898), human constitutively active (Q67L)

pGFP-Rab8A (Plasmid #24900), and human dominant-negative (T22N) pGFP-Rab8A (Plasmid #24899) were obtained from Addgene. The pGEX-2T-GST-JCF1 (Rab-binding domain of JCF1, RBD) plasmid was a generous gift from Dr. Wei Guo [22]. Flag-IFT20 plasmid was a generous gift from Joon Kim, KAIST, Korea.

### Reagents

The antibodies used in this study are listed in Appendix Table S2. Lysotracker (#L7528) was purchased from Thermo Fisher Scientific (Waltham, MA, USA). LDL (#437644) was purchased from EMD Millipore Corporation. Filipin (#F4767), cholesterol–methyl-beta-cyclodextrin complex (#C4951-30MG), 2-hydroxypropyl-beta-cyclo-dextrin (#332593), U18666A (#U3633), and unlabeled transferrin (#T0665) were obtained from Sigma. Transferrin Alexa Fluor 568 was purchased from Invitrogen. The Cholesterol Assay Kit (#K623-100) was obtained from BioVision.

### Filipin staining

Cells grown on a coverslip were fixed with 4% paraformaldehyde for 30 min at room temperature and rinsed three times with phosphate-buffered saline (PBS). Paraformaldehyde was quenched with 1.5 mg/ml glycine in PBS (pH 7.4) for 10 min. Subsequently, 25 µg/ml filipin in PBS was added, and incubated for 2 h at room temperature and rinsed three times with PBS, and the coverslip was mounted on slides using 90% (V/V) glycerol.

### Immunofluorescence

Cells grown on coverslips were fixed with 4% paraformaldehyde for 30 min at room temperature or with methanol at −20°C for 10 min depending on the antibodies as described in Appendix Table S2. Cells were rinsed three times with PBS, permeabilized with 0.25% Triton X-100 for 5 min, and rinsed three times with PBS, followed by blocking with 3% bovine serum albumin (BSA) for 1 h at room temperature. The cells were then incubated with primary antibodies in 3% BSA, rinsed three times with PBS, and labeled with fluorescent Alexa Fluor 488 or Alexa Fluor 568 (molecular probes)-conjugated secondary antibodies (1:500) for 30 min. To detect the nuclei, the coverslips were mounted on slides with Prolong Gold antifade reagent containing DAPI (P36931, Sigma) and examined under an Olympus FluoView 1000 confocal laser-scanning microscope or fluorescence microscope (IX71, Olympus, Tokyo, Japan).

### Quantification of fluorescent colocalization

Fluorescent intensity within co-immunostained cells was determined using ImageJ software. Colocalization was evaluated with the colocalization plugin for ImageJ software by calculating Manders' overlap coefficients (OCs), which allows for the quantification of overlapping pixels from each channel as previously reported [12]. For quantification of the percentage of lysosome–peroxisome overlap, the OC of lysosome fluorescent signal along with the peroxisome fluorescent signal was quantitatively assessed. An OC equivalent to 1 was defined as 100%, and each individual OC was expressed relative to this 100% value.

### IFT20 fluorescent intensity at centrioles

Green and red channels from IFT20 and γ-tubulin co-stained Z-stacked confocal images were separated by ImageJ software. Green channel corresponding to γ-tubulin was cropped to only include the centriole area. The same pixel dimension was applied to red channel to crop the centriole, and the IFT20 area and then intensity were measured by ImageJ software. IFT20 intensity from 100 cells was scored in each group for each experiment. IFT20 intensity from serum-starved control cells was considered as 100%.

### SREBP2 cleavage assay

Cells were treated with 1.5% HPβCD in a cholesterol-deficient medium (DMEM, 5% lipoprotein-deficient serum, 1 mM compactin, and 10 mM mevalonate) for 1 h to deplete intracellular cholesterol. The cells were then incubated with the cholesterol-deficient medium in the presence or absence of 50 μg/ml LDL for 12 h and subjected to cell fractionation followed by Western blot analysis.

### Cell fractionation

Cell pellet was suspended in Buffer A (50 mM HEPES pH 7.6, 1.5 mM MgCl$_2$, 10 mM KCl) with protease and phosphatase inhibitors (Gene Depot). Cell disruption was achieved by passing cells through a 22-G1 needle 30 times, followed by centrifugation at 1,000 $g$ for 5 min at 4°C (C1). Pellet was suspended in Buffer C (20 mM HEPES pH 7.6, 1.5 mM MgCl$_2$, 0.42 mM NaCl, 2.5% glycerol) with protease and phosphatase inhibitors and rotated for 1 h at 4°C; supernatant comprised the nuclear extract. Post-nuclear supernatant obtained from C1 was centrifuged at 18,500 $g$ (C2). Supernatant thus obtained comprised the cytosol fraction, and the pellet from C2 represented the membrane fraction. Lysis buffer [10 mM Tris–HCl pH 6.8, 100 mM NaCl, 1% sodium dodecyl sulfate (SDS)] with protease inhibitors was added and incubated for 30 min at room temperature. SDS loading buffer (60 mM Tris–HCl pH 6.8, 2% SDS, 1% β-mercaptoethanol, 10% glycerol, and 0.02% bromophenol blue) was added and denatured at 95°C for 5 min.

### Subcellular fractionation by sucrose floatation gradient

Briefly, confluent monolayers of RPE1 cells were suspended in a homogenization buffer (250 mM sucrose, 20 mM Tris–HCl pH 7.2, 1 mM EDTA with protease inhibitors) by 10 passages through a 22-G1 needle. Homogenate was centrifuged at 1,000 $g$ for 10 min at 4°C, and post-nuclear supernatant (PNS) was collected. Subcellular fractionation was performed on a discontinuous sucrose gradient as described by Walker *et al* [47], with some modifications. A discontinuous sucrose gradient was prepared by the sequential addition of three different concentrations of sucrose solution (40%, 35%, and 25%). PNS (1.5 ml) was then placed on top of the 25% gradient. Finally, homogenization buffer was added on top of the PNS, resulting in a total gradient volume of 12 ml. The gradient was centrifuged at 200,000 $g$ for 8 h at 4°C in a swingout Beckman SW41 rotor and de-accelerated without applying the brakes. After centrifugation, 1 ml of each fraction was collected from the top to bottom. Aliquots of each fraction were subjected to Western blotting for analysis of the distribution of lysosome, peroxisomes, endosome, Golgi, and cell membrane markers along the gradient.

### Total cholesterol measurement

Two rounds of cholesterol extraction were performed from each fraction of the sucrose gradient. Briefly, for each fraction, 200 μl of hexane was added followed by vigorous vortexing, and the organic phase was transferred to a new tube. The solvent was removed by nitrogen blow using dry heat at 56°C, and total cholesterol measurement was carried out according to the manufacturer's instructions.

### Whole cell lysis

Cells were lysed on ice in RIPA lysis buffer (20 mM HEPES pH 7.5, 150 mM NaCl, 1% Triton X-100, 1% sodium deoxycholate, 1 mM EDTA) supplemented with protease and phosphatase inhibitors and centrifuged at 18,500 $g$ for 10 min at 4°C. SDS loading buffer was added to the supernatant and denatured at 95°C for 5 min.

### Immunoprecipitation assay

Immunoprecipitation was carried out using the Pierce Crosslink Immunoprecipitation Kit (Thermo Scientific) according to the manufacturer's instructions. In brief, cells were rinsed once with ice-cold PBS, lysed with ice-cold lysis buffer, and centrifuged at 18,500 $g$ for 10 min at 4°C. Supernatant was added to protein A/G resins cross-linked to GFP antibody, incubated for 12 h at 4°C, and eluted. Non-reducing sample buffer was added to the eluates and input, and then denatured at 95°C for 5 min.

### Detection of GTP-bound Rab8 in cell lysates

Recombinant GST-JCF1 (RBD) was expressed in *Escherichia coli* strain BL21(DE3) and purified in glutathione Sepharose 4B according to the manufacturer's instructions (GE Healthcare). GTP-bound Rab8 in cell lysates was detected by a JCF1 (RBD) pull-down assay as reported previously [22]. Briefly, RPE1 cells were lysed in lysis buffer (20 mM Tris–HCl pH 7.4, 100 mM KCl, 5 mM MgCl$_2$, 0.5% Triton X-100, 1 mM DTT, 1 mM PMSF, protease inhibitor cocktail, and phosphatase inhibitor cocktail). Cell lysates were incubated with GST or GST-JCF1 (RBD) for 4 h at 4°C, followed by washing with lysis buffer. Bound GTP-Rab8 was analyzed by SDS–polyacrylamide gel electrophoresis (PAGE) and immunoblotted with Rab8 antibody. The intensity of bands was quantified by ImageJ software. The amount of GTP-Rab8 was normalized to the control level.

### Western blotting

Aliquots of each protein lysate were subjected to SDS–PAGE for 120 min at 100 V. After electrophoresis, proteins were transferred to nitrocellulose membranes for 120 min at 300 mA and blocked for 30 min with 5% skim milk in Tris-buffered saline with Tween buffer. Membranes were incubated with primary antibodies overnight at 4°C, followed by incubation with peroxidase-coupled secondary antibodies for 1 h at room temperature, and antibody-targeted proteins were visualized using a Western Blot Detection Kit (#LF-QC0103, Abfrontier).

## RNA extraction and reverse transcription–qPCR

Total RNA was extracted from the sample using TRIzol reagent (#15596018, Life Technologies). A Reverse Transcription Kit (Roche, Indianapolis, IN, USA) was used to transcribe cDNA, and then, qPCR was performed with cDNA as a template using a Light Cycler system with FastStart DNA Master SYBR Green I (Roche, Indianapolis, IN, USA). The human primer sequences (forward and reverse) are listed in Appendix Table S3.

## VLCFA extraction and gas chromatography–mass spectrometry (GC-MS)

Very-long-chain fatty acids were extracted using acidic methanolysis. A mixture of methanol (2 ml) and 37% hydrochloric acid solution (80:20, v/v) was added sequentially to the sample with an internal standard (5 μg/ml tricosanoic acid, C23:0 fatty acid). Collected cell pellets were fully disrupted by three freeze–thaw cycles in liquid $N_2$, placed in an ice-cold freezer, followed by 1-h incubation at 100°C. Samples were subsequently extracted twice with 4 ml of hexane by vigorously shaking for 1 min. After centrifugation (12,000 *g* for 5 min), the organic layer was transferred to a glass vial. The solvent was removed by a nitrogen purge and reconstituted into 200 μl of hexane.

Very-long-chain fatty acids were analyzed by GC-MS on a GCMSQP2010 system (Shimadzu, Tokyo, Japan) equipped with a DB-5 MS capillary column (30 m × 0.25 mm, 0.25 μm; Agilent Technologies, Palo Alto, CA, USA). High-purity helium (1 ml/min constant flow) was used as a carrier gas with the following column temperature program: initial temperature of 100°C, hold at 100°C for 2 min, increase to 190°C at 15°C/min, increase to 300°C at 5°C/min, and hold for 5 min, for a total of 35 min. The compounds were then ionized using electron impact mode with a 70-eV filament at a 200°C ion source temperature. The mass detection range was 40–500 m/z, with a scan rate of 2,500 s$^{-1}$.

## Quantification and statistical analysis

To determine the percentage of ciliated cells, at least 250 cells were scored in each experiment. For the rescue experiment using Rab8 plasmids, 200 GFP-positive cells were scored in each experiment and the percentage of ciliated cells was quantified by considering cells with both GFP-Rab8 and ARL13B signals in the primary cilium. Images were acquired with a fluorescent microscope (IX71, Olympus, Tokyo, Japan) or confocal laser-scanning microscope (Olympus FluoView 1000). To calculate the percentage of cells with IFT88 and Rab8 localized to the centrioles, 200 cells were scored in each experiment using Z-stacked confocal microscopic images (Olympus FluoView 1000). IFT20 fluorescent intensities at the centrioles were measured from the images acquired from confocal microscope, and 150 cells were scored in each experiment. For the determination of ciliated cells from Flag-IFT20-expressing cells, 150 cells were scored per independent experiment from the images acquired from fluorescent microscope (only those cilia with both the Flag-IFT20 and ARL13B were considered for quantification). To determine CP110 loss from the centrioles, 200 cells were scored per independent experiment from the images acquired from fluorescent microscope. For the

determination of the percentage of cells with EHD1 in cilium or in the distal end of basal body (ciliary vesicle), 200 cells were scored per independent experiment. The differences between experimental values were considered significant at $P < 0.05$ in a two-tailed Student's *t*-test.

**Expanded View** for this article is available online.

## Acknowledgements

This work was supported by the National Research Foundation of Korea (NRF) under grants funded by the Korean government (Grant Nos. 2018R1A5A1024340 and 2019R1A2C2086080).

## Author contributions

YM performed formal analysis and investigation, methodology development, conceptualization, writing of the original draft, review, and editing. JNL contributed to conceptualization, methodology development, project administration, and supervision. RKD performed investigations and validation of experimental results. SAK contributed to data curation, validation of experimental results, and supervision. CP contributed to project administration, supervision, and resource and software management. S-KC contributed to conceptualization and supervision. RP contributed to funding acquisition, conceptualization, visualization, and supervision.

## Conflict of interest

The authors declare that they have no conflict of interest.

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
