## [Review Process File · EMBO Reports]

TMEM135 regulates primary ciliogenesis through modulation of intracellular cholesterol distribution

Yunash Maharjan, Joon No Lee, SeongAe Kwak, Raghendra Kumar Dutta, Channy Park, Seong-Kyu Choe, and Raekil Park

Review timeline:

Submission date:	19 July 2019
Editorial Decision:	27 August 2019
Revision received:	26 November 2019
Editorial Decision:	22 January 2020
Revision received:	31 January 2020
Accepted:	14 February 2020

Editor: Deniz Senyilmaz-Tiebe

Transaction Report:

1st Editorial Decision

27 August 2019

Thank you for the submission of your research manuscript to our journal, which was now seen by three referees, whose reports are copied below.

I apologize for the delay in getting back to you, it took longer than anticipated to receive the full set of referee reports.

As you can see, the referees express interest in the proposed role of TMEM135 in ciliogenesis. However, they also raise a number of concerns that need to be addressed to consider publication here. In particular, both referees #1 (1st paragraph) and 2 (point #1 finds that as it stands, also in the context of earlier literature, the data do not conclusively support that the effect of TMEM135 depletion is solely due to disrupted intracellular cholesterol distribution. Moreover, referees require better characterization of (endogenous) TMEM135 subcellular localization, given the earlier reports demonstrating its mitochondrial localization. Furthermore, one of the referees alerted us to a potential image duplication issue in figure 2, which needs to be sorted out.

Should you be able to address all referee criticisms in full, we could consider a revised manuscript. I do realize that addressing all the referees' criticisms will require a lot of additional time and effort and be technically challenging. I would therefore understand if you wish to publish the manuscript rapidly and without any significant changes elsewhere, in which case please let us know so we can withdraw it from our system.

If you decide to thoroughly revise the manuscript for EMBO Reports, please revise your manuscript with the understanding that the referee concerns (as detailed above and in their reports) must be fully addressed and their suggestions taken on board. Please address all referee concerns in a complete point-by-point response. Acceptance of the manuscript will depend on a positive outcome of a second round of review. It is EMBO reports policy to allow a single round of revision only and acceptance or rejection of the manuscript will therefore depend on the completeness of your responses included in the next, final version of the manuscript.

REFEREE REPORTS

Referee #1:

The at hand manuscript of Maharjan et al. studies a novel function of TMEM135 in the early steps of ciliogenesis. Reduction of TMEM135 results in defective cholesterol distribution in the cell by reducing lysosome peroxisome membrane contacts. This function of TMEM135 has been previously shown (Chu et al., Cell 2015). In addition, TMEM135 deficiency results in defective Rab8 trafficking to centrioles as well as IFT20 targeting. Defective ciliogenesis in Tmem135 depleted cells is rescued by Chol-MBCD but not by removal of accumulated cholesterol from the cells. A role of cholesterol in cilia extension however has recently been suggested in zebra fish (Maerz et al, Comm Biol 2019). Here, the reduction of the transition zone component Pi(4,5)P2 has been proposed as potential mechanisms.

This is an interesting study on an interesting topic. However, I feel that some of the data is not absolutely novel (Fig 1 A,B), some is very descriptive and does not really support the conclusions drawn by the authors. In addition, the manuscript is pretty difficult to read. Figures should have a stronger focus on the novel aspects. The existence of both appendix and EV figures does not make it easier. Some editorial language polishing will be required.

One major problem is the unclear localization and function of TMEM135, which has been previously described both in peroxisomes and mitochondria. Here, solid localization data of endogenous TMEM135 is urgently required, either using antibodies or using endogenous protein tagging with Crispr/Cas9. If Tmem135 really was a major player in ciliogenesis, it is at least surprising, that a mutant mouse model displays a rather mild eye phenotype without any severe symptoms of a ciliopathy (Lee et al., elife 2016). In general, murine Tmem135 seems to be clearly linked to mitochondrial functions. In addition, beside cholesterol accumulation Tmem135 depletion might have many additional effects on peroxisome functions that could also impact cilia biology at some point. Therefore, I feel that the conclusion, that the effect of TMEM135 depletion on ciliogenesis depends on the disruption of intracellular chol. distribution, is difficult. It would be important to see to what extent the additional candidates identified in the Cell paper of Chu et al. from 2015 have similar effects on ciliogenesis, Rab8 and IFT20 as shown for Tmem135. Especially, since TMEM135 is not the candidate with the highest accumulation of cholesterol and subsequently the lowest amount of cholesterol at the PM. Knockdown of other candidate proteins has been found to have more dramatic effects (eg. ABCD1 or BAAT)

Some additional aspects that should be addressed:

- 1) As stated above, the localization of TMEM135 remains unclear. The authors claim to show endogenous protein localization, but actually they do not show this. All localization data is based on overexpressed myc.TMEM135, later stained with anti-Tmem135. Obviously, antibodies are available for both WB and IF (see staining of myc.Tmem135). Thus, untransfected cells should be stained to obtain an idea of the "real" localization of TMEM135. If expression levels are too low, perhaps increase TMEM135 levels by LXR stimulation (Renquist et al, bioRxiv 2018)?! In addition, c-term tagged Tmem135 should be investigate as well.
- 2) Does depletion of TMEM135 with siRNA really affect the staining pattern of endogenous Tmem135? Please exclude that the peroxisomal localization is the result of overexpression.
- 3) While three independent siRNAs are available, most experiments are done with only one of those. This neglects the possibility of off-target effects.
- 4) The experiment presented in Fig. 2f is unclear. What is the exact time course of this experiment? Does the chol-MBCD treatment affect the knockdown of TMEM135, does it affect transfection efficiency? TMEM135 levels should be shown for all points.
- 5) How does the inhibition of chol-synthesis by statins affect Tmem135 depletion? Statins/Cholesterol and ciliogenesis have been linked recently in zebrafish.

- 6) Does the depletion of TMEM135 affect cells that already carry cilia? Since Shh signaling depends on cholesterol: Is Shh signaling affected by Tmem135?
 7) Western blot should be shown with molecular weight standards.

Referee #2:

This report presents an interesting observation, namely that a peroxisome localized protein influences lysosomal cholesterol levels and appears to influence Rab8 activation and localization. The findings could be of broad interest. However there are a number of aspects of the study that need additional work before presentation in EMBO Reports can be recommended.

1. Previous work has shown that knockout of TMEM135 leads to a strong block in the early secretory pathway (Simpson et al. Nat Cell Bio) and also, the protein was characterized by others as causing a defect in peroxisomal long chain fatty acid oxidation, leading to an overall accumulation of triglycerides (in worm and HepG2 cells) and cell death 48-72 hours post siRNA. A recent paper on BioRxiv concluded, "Knockdown of TMEM135 in HepG2 cells caused triglyceride accumulation, indicating a potential role in [peroxisomal] β -oxidation. ...in fasted TMEM135 knockdown mice, there was a further significant increase in hepatic fatty acid concentrations and a significant decrease in NADH." If this is correct, the question is whether the block in ciliation and block in Rab8 activation reported here are specific for the ciliation pathway or are much more general. Are the cells low in ATP and GTP altogether? This would explain all of the phenotypes observed herein and needs to be examined. Aside from this significant concern, there are other issues that need to be clarified.

2. Fig. 1 A,B. What do the authors mean by lysosome-peroxisome overlap (%)? Is this percent of lysosomes showing overlap or percent of peroxisomes showing overlap? Why is there less fluorescence green and red, in the TMEM135 examples shown? Please include corresponding quantitation of total fluorescence and numbers of structures.

3. Fig 1C. The method used is not that described in ref. 45. With due respect to the authors, this reviewer believes that the bottom three rows of the blots may be flipped. Golgi (GM130) and plasma membranes (ATPase) have the same buoyant density as early endosomes (EEA1) and so it is very hard to imagine how they could have spun to the bottom of the gradient described in 90 minutes. They are usually less dense than lysosomes, or?

4. Why is there a discrepancy between JCF1 binding in Fig. 4 and 7C? The blots were done using a peroxidase method that is nonlinear and will exaggerate differences. Please add quantitation with a standard curve or use LICOR for a linear response. More importantly, ESSENTIAL: please show that the defect is specific to Rab8. In other words, does another Rab with no link to ciliation also have activation problems under these conditions?

5. The Rab8 compartment fails to coalesce perinuclearly. ESSENTIAL: Do the Golgi and lysosomes also show a different morphology? Previous work reported changes in overall mitochondrial morphology. This could be due to a general microtubule or motor defect and would be essential to address.

In summary, understanding the role of TMEM135 is interesting and important. This reviewer would prefer to see some more precise hint as to why Rabin 8 is not functioning under the experimental conditions employed. However it is likely a more general cellular defect, and more work is needed to understand the phenotypes observed before presentation in EMBO Reports can be recommended.

Referee #3:

In this manuscript, Maharjan and colleagues elucidated the function of TMEM135 during cilium-related cellular processes and cholesterol transport in mammalian cells. TMEM135 was previously identified in an shRNA screen as a potential protein that regulates cholesterol transport and was published in biorxiv as an LXR-inducible regulator of peroxisomal metabolism. In agreement with the published biorXiv work, the authors first showed that TMEM135 localizes to peroxisomes and regulates cholesterol transport. Given that they also identified ciliogenesis defects in TMEM135-

depleted cells, they investigated the molecular mechanism of this defects and showed that TMEM135 depletion results in defective Rab8 trafficking to the basal body and its activation, as well as IFT20 recruitment to the basal body. Cholesterol supplementation to TMEM135-depleted cells rescued Rab8 trafficking and activation defects, suggesting a link between intracellular cholesterol distribution and ciliogenesis.

The molecular mechanism of ciliogenesis and its regulation is general interest to the field and the novelty of this paper is that it identifies a protein at the intersection of cholesterol biology and ciliary vesicle trafficking during ciliogenesis. Although more needs to be investigated to determine how these pathways are linked, the data included in this paper is sufficient to support this link and opens new avenues for the field. However, for some of their conclusions, the authors did not include the required controls and data, which are listed as major/minor points below. Once these points are addressed by the authors, I recommend publication of this paper in EMBO Reports.

- 1- Intracellular cholesterol transportation depends on microtubules. If TMEM135 has role on intracellular cholesterol transportation via affecting microtubule organization, it would also affect Rab8 and Ift20 localization around centrosome. They should perform noc washout experiments to determine the affects of TMEM135 on microtubule nucleation and anchoring.
- 2- Based on Smo-GFP localization to basal bodies, they suggest that there are no defects in ciliary vesicle formation. Given that this is one of the major conclusions of the paper, they should show this phenotype by other markers or experiments (i.e. TEM).
- 3- They must show data to confirm efficient depletion of TMEM135 after siRNA treatment, ideally both by immunofluorescence and immunoblotting. This is important for a) confirming the specificity of the TMEM135 antibody 2) validating the knockdown of TMEM135. They used three different siRNAs in some experiments, and tow or one in other. Which one is more efficient in TMEM135 knockdown?
- 4- How did they conclude for Fig. EV1 that lysosomal accumulation is due to the physical dissociation between peroxisomes and lysosomes? This figure only demonstrates increased cholesterol accumulation upon TMEM135 depletion?
- 5- What marker did they use to quantify cholesterol levels in sucrose gradients in Fig. 1C as all markers included in this figure are markers of other organelles.
- 6- For Fig. 2A and 2B, they should perform rescue experiments with ectopic siRNA-resistant TMEM135 to confirm the specificity of the ciliogenesis defects.
- 7- They should include a reference for the data on how MbetaCD impacts intracellular cholesterol distribution (Fig. 2F/G-related text). Can they use an assay to show that TMEM135-depletion causes a defect in interacellular cholesterol depletion as this is important for their conclusions in Fig. 2F/G.
- 8- In Fig. 4A, the ability of the wild-type Rab8a and its mutants to localize to cilia varies in control and TMEM135-depleted conditions. The authors only showed that for the affect of TMEM135 depletion on Rab8a localization around the basal body. They should also quantify phenotypes associated with Rab8a ciliary recruitment to reveal the full set of phenotypes. Since Rab8a DN mutant and wt does not localize to cilia in TMEM135-depleted cells (based on their representative images in Fig. 4A), it is also likely that the phenotype is also in part due to the inability of Rab8a to localize to cilia.
- 9- For Fig. 4D, GST must be included as a control in parallel to GST-JCF1 pulldowns.
- 10- For Fig. 5F, they must quantify the percentage of IFT20-positive cilia to conclude on ciliary IFT20 recruitment defects, because the decrease in IFT20-positive cilia can just be due to the decrease in ciliogenesis upon TMEM135 depletion.
- 11- In the manuscript, for all phenotypes they report a significant change (i.e. ciliogenesis ..), they must include numbers of averages, SEM, p value, how many experiments they performed and how many samples they counted.
- 12- For some of the supplementary data (i.e. S4 ..), they only include conclusions on these data in discussion, which makes manuscript very hard to follow and does not address questions until one reads discussion. All the data in discussion must be moved to associated figures in the results section.
- 13- The figures demonstrating peroxisomal localization of myc-TMEM135 and endogenous TMEM135 (referred as Appendix FigS1) is not available as part of the manuscript files. This localization was previously published in a biorxiv paper, which must be cited.
<https://www.biorxiv.org/content/10.1101/334979v3>
- 14- The figures demonstrating consequences of TMEM135 depletion on peroxisomal protein abundance is not available (referred as Appendix FigS2).
- 15- They should include in "materials and methods" details on how they quantified the percentage of the lysosomal-peroxisomal overlap for Fig. 1A/B.

- 16- For Figure 4B, the quantification bars for GFP-Rab8T22N condition is missing from the graph.
- 17- For Fig. 5B-E, they summarize their conclusions in one sentence, which is not sufficient to support their conclusions and makes it very hard to follow the manuscript.
- 18- They should comment on why they might have observed a contradictory result from a previously published paper for the effects of Rab8a depletion on centriolar IFT20 recruitment (Fig. 6).
- In Figure 1.E, CREB is used as nucleus marker and UBXD8 is used as membrane marker however they are not indicated in text and figure legend.
 - Cilium are not visible in Figure 5C. They should show better images, maybe with another ciliary marker like Arl13b.
 - Representative images are needed for Figure 5E.
 - Figure numbers are missing.

1st Revision - authors' response

26 November 2019

Reviewer #1:

The at hand manuscript of Maharjan et al. studies a novel function of TMEM135 in the early steps of ciliogenesis. Reduction of TMEM135 results in defective cholesterol distribution in the cell by reducing lysosome peroxisome membrane contacts. This function of TMEM135 has been previously shown (Chu et al., Cell 2015). In addition, TMEM135 deficiency results in defective Rab8 trafficking to centrioles as well as IFT20 targeting. Defective ciliogenesis in Tmem135 depleted cells is rescued by Chol-MBCD but not by removal of accumulated cholesterol from the cells. A role of cholesterol in cilia extension however has recently been suggested in zebra fish (Maerz et al, Comm Biol 2019). Here, the reduction of the transition zone component Pi(4,5)P2 has been proposed as potential mechanisms. This is an interesting study on an interesting topic. However, I feel that some of the data is not absolutely novel (Fig 1 A, B), some is very descriptive and does not really support the conclusions drawn by the authors. In addition, the manuscript is pretty difficult to read. Figures should have a stronger focus on the novel aspects. The existence of both appendix and EV figures does not make it easier. Some editorial language polishing will be required.

Our Reply:

We thank the Reviewer for the evaluation of our manuscript. In response to the Reviewer's comment, we have included the results of additional experiments. We believe that these new results would effectively address the Reviewer's concerns.

We tried our best to make a coherent manuscript and logical conclusion based on different lines of evidence. We sincerely apologize for the inconvenience felt by the Reviewer and we agree that understanding this manuscript would require some effort as it encompasses several different aspects of biology like ciliogenesis, peroxisomes, lysosomes, intracellular cholesterol including different aspects of TMEM135 itself. We would be more than happy to get valuable suggestions from the Reviewer and the Editor to make this manuscript uncomplicated to read and understand. We agree with the Reviewer that data shown in Fig1A-B is not novel but this data is important to address why depletion of peroxisome membrane protein TMEM135 could trigger lysosomal cholesterol accumulation and impairs intracellular cholesterol distribution in RPE1 cells.

One major problem is the unclear localization and function of TMEM135, which has been previously described both in peroxisomes and mitochondria. Here, solid localization data of endogenous TMEM135 is urgently required, either using antibodies or using endogenous protein tagging with Crispr/Cas9. If Tmem135 really was a major player in ciliogenesis, it is at least surprising, that a mutant mouse model displays a rather mild eye phenotype without any severe symptoms of a ciliopathy (Lee et al., elife 2016). In general, murine Tmem135 seems to be clearly linked to mitochondrial functions. In addition, beside cholesterol accumulation Tmem135 depletion might have many additional effects on peroxisome functions that could also impact cilia biology at some point. Therefore, I feel that the conclusion, that the effect of TMEM135 depletion on ciliogenesis depends on the disruption of intracellular chol. distribution, is difficult. It would be important to see to what extent the additional candidates identified in the Cell paper of Chu et al. from 2015 have similar effects on ciliogenesis, Rab8 and IFT20 as shown for Tmem135. Especially, since TMEM135 is not the candidate with the highest accumulation of cholesterol and

subsequently the lowest amount of cholesterol at the PM. Knockdown of other candidate proteins has been found to have more dramatic effects (eg. ABCD1 or BAAT)

Our Reply:

We agree with the Reviewer that the localization of TMEM135 is controversial. Therefore, we take this opportunity to clarify that TMEM135 is a peroxisome localized protein. Two independent studies of rat and liver peroxisome proteomics had identified TMEM135 (originally referred to as peroxisome membrane protein 52, PMP 52) as a peroxisome localized protein (1, 2). Importantly, both the C-terminally and the N-terminally Myc-tagged version of the TMEM135 colocalized with GFP-PTS1 (1). However, as a result of the TMEM135 overexpression, the number of peroxisomes was found to be significantly decreased, and a significant portion of the GFP-PTS1 was localized in the cytoplasm, indicating that TMEM135 might play a role in maintaining peroxisome morphology, import of peroxisomal protein as well as peroxisome turn over (1). As shown in Appendix Figure S1A (Upper panel) and Appendix Fig S1B (Upper panel), C-terminally Myc-tagged TMEM135 (Myc tagged to C-terminus of TMEM135, TMEM135-Myc) colocalized with peroxisome marker PMP70 in Huh7 and RPE1 cells respectively. TMEM135-Myc also colocalized with RFP-SKL (Peroxisome targeting sequence 1, PTS1) in Huh7 cell as shown in Appendix Figure S1A, middle panel. In agreement with the peroxisomal localization of transfected TMEM135, endogenous TMEM135 (stained by TMEM135 antibody) was also found to colocalize with PMP70 in RPE1 cell (Appendix Figure 1B, middle panel). However, we did not observe the phenotype as previously described for TMEM135 overexpressed cells (1), our data strongly suggests that TMEM135 is a peroxisome localized protein. Furthermore, TMEM135-Myc was not found to colocalize with mitochondria marker Tomm20 in both the Huh7 and RPE1 cell (Appendix Fig1A, lower panel, Appendix Figure 1B, lower panel) as suggested in a recent study that both the endogenous vesicle-like structure of TMEM135 and GFP-TMEM135 (GFP fused to N-terminus of TMEM135) localizes to mitochondria (3). Additional experiments were performed to support the peroxisomal localization of TMEM135, the specificity of TMEM135 antibody and the TMEM135 siRNA which have been included in the respective comments raised by this Reviewer.

We agree with the Reviewer's comment that if TMEM135 would really play a critical role in ciliogenesis, TMEM135 mutant mouse would have shown severe features associated with ciliopathy rather than mild retinal degeneration as recently reported (3). The reason, why TMEM135 mutant mouse did not display severe ciliopathy symptoms could be summarized as follows

- 1) As mentioned in the report by Lee et al., wild-type TMEM135 contains five transmembrane helices while the 4th and the 5th transmembrane helices are lost in the mutant TMEM135 (due to early stop codon arising from frameshift mutation) which orient itself on the membrane with reversed orientation, suggesting that a shorter version of TMEM135 protein still expressed in the TMEM135 mutant mice. However, the authors have suggested that the shorter version of mutant TMEM135 proteins are likely to have abnormal function, it is hard to say with certainty about the functional domains of TMEM135 until full characterization of the TMEM135 protein (topology and biochemical properties) is achieved. Thus, it is possible that TMEM135 mutant mice did not display severe ciliopathy symptoms as mutant TMEM135 protein might exhibits its properties required for ciliogenesis.
- 2) At least in the cultured cells, Rab8 was found to be important for ciliogenesis (4) and defective Rab8 function impairs ciliogenesis at the stage of ciliary vesicle extension (5). Surprisingly, Rab8a and Rab8a/Rab8b (double) knockout mice did not display ciliopathy like features suggesting that other GTPases might be working with other Rab8 in the *in vivo* system (6, 7). In the present study, TMEM135 depletion in RPE1 was found to affect ciliogenesis through impaired Rab8 function. Therefore, even if it would be true that mutant TMEM135 protein was defunct in mutant TMEM135 mice as reported (3), the probability of observing ciliopathy in the mutant mice seems very low because the impaired ciliogenesis observed in TMEM135 depleted cells was specifically associated with Rab8 function.
- 3)

Although, it has been suggested that TMEM135 is important for controlling mitochondrial fusion and fission (3), we did not observe obvious changes in mitochondrial morphology in TMEM135 depleted cells (see FigR17, comment #5 raised by Reviewer #2). Furthermore, TMEM135 depletion did not affect expression of genes associated with beta-oxidation (mitochondrial and peroxisomes) and fatty acid synthesis, as well as total cellular ATP levels was unaffected. Moreover, TMEM135

depletion did not alter the formation of lipid droplets in basal conditions. Lipid droplets were slightly increased in TMEM135 depleted cells as compared to control cells in the presence of oleic acid (which is well known to induce lipid droplets formation). Interestingly, TMEM135-myc transfected cells tend to have a lesser number of lipid droplets as compared to untransfected cells in the presence of oleic acid, suggesting that TMEM135 might play a role in fatty acid metabolism. These observations are a part of our response to the concerns raised by to Reviewer #2 and the related data has been included in the respective Reviewer's comment section as Figure R10, Figure R11 and Figure R12. However, our observations and other reports on TMEM135 suggest that TMEM135 might have a role in lipid metabolism (mitochondrial or peroxisomal), understanding the direct or indirect role of TMEM135 on lipid metabolism might require special conditions. Importantly, TMEM135 depletion from peroxisome did not affect the abundance of peroxisome proteins like PMP70 and catalase neither the transport of peroxisomal matrix protein and very long-chain fatty acid metabolism in basal condition as shown in Appendix Figure S2. The only observable phenotype associated with TMEM135 depletion was a robust cholesterol accumulation in lysosomes and impaired ciliogenesis. Therefore, the possibility that additional peroxisomal function of TMEM135 which might impact ciliogenesis seems to be insignificant at least under the conditions employed to analyze ciliogenesis and lysosomal cholesterol accumulation.

We thank the Reviewer for raising this important issue whether depletion of other peroxisomal candidates known to impair intracellular cholesterol transport also affects ciliogenesis programming as TMEM135 depletion. We fully support the Reviewer's point that TMEM135 is not the candidate whose depletion results in highest lysosomal cholesterol accumulation and subsequently the lowest cholesterol level at the cell membrane (8). As shown in Figure R1 below, ABCD1 depletion in RPE1 cells results in cholesterol accumulation (Figure R1-B, filipin staining), decreases the percentage of ciliated cells (Figure R1-C), severely impairs Rab8 trafficking (Figure R1-D) as well as enhanced IFT20 localization at centrioles (Figure R1-E)

were negatively affected. However, more studies are needed if the ciliary phenotype observed in ABCD1 depletion is associated with intracellular cholesterol transport or very long-chain fatty acid accumulation. Albeit, these data suggest that peroxisome is an important hub for intracellular cholesterol homeostasis and defect associated with this pathway could have negative consequences on ciliogenesis.

Figure R1. ABCD1 depletion results in cholesterol accumulation, decreases percentage of ciliated cells, Rab8 and IFT20 trafficking to the centrioles. (A) ABCD1 knockdown efficiency by qPCR in RPE1 cells. Data represents mean \pm SD (n=3 experiments). *P < 0.05, Student's t-test. (B) ABCD1 knockdown results in cholesterol accumulation. Representative fluorescent images of filipin (blue) are shown. Scale bar, 20 μ m. (C) ABCD1 knockdown decreases percentage of ciliated cells. Representative images showing ciliated cells, immuno-stained by cilia marker ARL13B. The bar graph shows percentage of ciliated cells in each condition. Data represents mean \pm SD (n=3 experiments), 200 cells were scored per condition per experiment. *P < 0.05, Student's t-test. (D) ABCD1 knockdown impairs Rab8 trafficking to the centrioles. Representative fluorescent images immune-stained with Rab8 (Red) and γ -tubulin (green) are shown. Scale bar, 5 μ m. The bar graph shows percentage of cells with Rab8 localized to the centrioles in each condition. Data represents mean \pm SD (n=3 experiments), 100 cells were scored per condition per experiment. *P < 0.05, Student's t-test. (E) ABCD1 knockdown decreases IFT20 intensity at the centrioles. Representative fluorescent images immunostained with IFT20 (Red) and γ -tubulin (green) are shown. Scale bar, 10 μ m. The bar graph shows IFT20 intensity at the centrioles as described in materials and method section in the manuscript. Data represents average of two independent experiment. 100 cells were scored per condition per experiment.

Some additional aspects that should be addressed:

1) As stated above, the localization of TMEM135 remains unclear. The authors claim to show endogenous protein localization, but actually they do not show this. All localization data is based on overexpressed myc.TMEM135, later stained with anti-Tmem135. Obviously, antibodies are available for both WB and IF (see staining of myc.Tmem135). Thus, untransfected cells should be stained to obtain an idea of the "real" localization of TMEM135. If expression levels are too low, perhaps increase TMEM135 levels by LXR stimulation (Renquist et al, bioRxiv 2018)?! In addition, c-term tagged Tmem135 should be investigate as well.

Our Reply:

With due respect to the Reviewer, we believed that there is some issue of misunderstanding about the data provided in Appendix Figure S1. Localization of transfected TMEM135-Myc (Myc tagged to the C-terminus of TMEM135) to the peroxisomes was analyzed by immunostaining with Myc-antibody and PMP70 antibody (PMP70 was used as a peroxisome marker). Localization of

transfected TMEM135-Myc to the peroxisomes was not investigated using the TMEM135 antibody. Furthermore, localization of the endogenous TMEM135 to peroxisome was analyzed by immunostaining with TMEM135 and PMP70 antibodies. Although, data shown in Appendix Figure S1 would have been adequate to suggest that TMEM135 is a peroxisome localized protein, we performed another experiment in response to a Reviewer comment that untransfected cells should be stained with TMEM135 antibody to obtain an idea of the real localization of TMEM135. As shown in Figure R2, Endogenous TMEM135 shows vesicle-like structures in both the untransfected and TMEM135-Myc transfected cells while showing complete localization with transfected TMEM135-Myc. As TMEM135-Myc localizes with PMP70 (Appendix Figure S1A Upper Panel, and Appendix Figure S1B Upper Panel), endogenous TMEM135 colocalizes with PMP70 (Appendix Figure S1B, Piddle Panel) and TMEM135-Myc shows colocalization with endogenous TMEM135 (Figure R2), it could be strongly stated that TMEM135 is peroxisome localized protein. We appreciate the Reviewer for suggesting the use of LXR agonist to increase TMEM135 expression level but in our study, we found that TMEM135 protein was well expressed to be detected by Western blotting and immunofluorescence.

Figure R2. TMEM135-Myc colocalizes with Endogenous TMEM135. Representative image showing that endogenous TMEM135 shows vesicular structures in both the untransfected and TMEM135-Myc transfected and TMEM135-Myc completely colocalizes with endogenous TMEM135. Scale bar, 5 μ m.

The Reviewer has suggested that localization of C-terminally tagged TMEM135 also needed to be addressed. We did not understand if the Reviewer mentioned to analyze localization of TMEM135 with a tag on C-terminus of TMEM135 or TMEM135 attached to the C-terminus of a tag. Since we have already analyzed the TMEM135-Myc (Myc tagged on C-terminus of TMEM135), we generated another construct with EGFP fused to the N-terminus of TMEM135. Briefly, TMEM135 coding region was amplified from the cDNA library of Huh7 cells with the forward primer 5'-ATCTCGAGATGGCGGCCCTCAGCAAG-3' and the reverse primer 5'-CGCGAATTCCTAGGAAAACCTGTGGGCAAC-3' using Q5 high fidelity DNA polymerase (New England Biolabs). Amplified TMEM135 sequence was then inserted into *XhoI-EcoR* I cut of pEGFP-C3 yielding the recombinant vector pEGFP-TMEM135. As shown in Figure R3-A, EGFP-TMEM135 colocalizes with peroxisomes as reported (9). EGFP-TMEM135 also colocalizes with endogenous TMEM135 vesicular structures (Figure R3-B). In this figure, we have also included an area with untransfected cells, stained with either PMP70 antibody or TMEM135 antibody to show that TMEM135 antibody being employed is very specific and TMEM135 localization to peroxisome is genuine.

Figure R3. EGFP-TMEM135 colocalizes to peroxisome. (A) Representative image showing that EGFP-TMEM135 colocalizes with peroxisome. Scale bar, 5 μ m. (B) Representative image showing that EGFP-TMEM135 colocalizes with endogenous TMEM135. Scale bar, 5 μ m.

Although, our data strongly suggest that the TMEM135 is a peroxisome localized protein as reported (1, 2, 9), the mitochondrial localization of TMEM135 as reported by Lee *et al.*, (3) could not be ignored, as peroxisome and mitochondria are known to share proteins with each other (10, 11, 12). Especially, during *de novo* peroxisome biogenesis, it has been reported that Pex3 and Pex14 vesicles derived from mitochondria fuses with ER-derived Pex16 vesicles to form an import competent peroxisome (10). Interestingly, Pex14 localizes to the mitochondria in PEX3 and PEX16 deficient cells (10). Therefore, it could be possible that changes in the dynamics of Pex16 or Pex3 could contribute to the mislocalization of TMEM135 in the mitochondria. Furthermore, peroxisomal membrane proteins (PMPs) are classified into class 1 and class 2 membrane proteins, based on poorly understood peroxisomal membrane targeting sequences (mPTS). Pex19 is known to import class1 mPTS while class 2 mPTS is imported into the peroxisomes independent of the Pex19 function (13). In addition, Pex16 has been reported to contribute to peroxisome maintenance and is indispensable for the import of PMPs from ER to the existing peroxisomes (14).

We attempt to understand which peroxin/s (Pex16, Pex3 or Pex19) are involved in the transport of TMEM135 into existing peroxisomes. We speculated that deficiency of any of the above- mentioned peroxins could lead to mislocalization of TMEM135 to mitochondria or ER. Therefore, we designed the experiment as described in Figure R4. We found that PEX3 and PEX16 knockdown significantly decreases peroxisomes while the PEX19 knockdown did not reduce peroxisome number in TMEM135-Myc transfected cells (Figure R4-G). Pex19 depletion did not affect TMEM135-Myc localization to peroxisomes (Figure R4-H). Although PEX3 reduces peroxisome numbers, TMEM135-Myc still colocalizes to existing peroxisomes while PEX16 knockdown significantly reduces peroxisomes number similar to PEX3 depletion, TMEM135-Myc localization to existing peroxisomes was significantly reduced and transfected TMEM135-Myc tends to predominantly localize in the ER (Figure R4-H). It has been previously reported that TMEM135 is one of the proteins in the Pex16 interactome (15). Even though much work needed to be performed our data suggests that TMEM135 is a bonafide peroxisome protein and possibly require PEX16 for its localization into the peroxisome.

Figure R4: Peroxisomal localization of TMEM135 depends upon Pex16. (A) Experimental design. **(B, C, D)** siRNA knockdown efficiency of PEX3, PEX19 and PEX16 siRNA by qPCR. Data represents mean \pm SD (n=3 experiments). *P < 0.05, Student's t-test. **(E)** PEX16 siRNA knockdown efficiency by Western blot. **(F)**

Representative fluorescent images showing transfected TMEM135-Myc and peroxisomes marker, PMP70. Scale bar, 10 μ m. **(G)** Peroxisome number in the TMEM135-Myc transfected cells. The bar graph shows average number of peroxisomes per cell. Data represents mean \pm SD (n=3 experiments), 30 transfected cells were scored per condition per experiment. *P < 0.05, Student's t-test. **(H)** Colocalization of TMEM135-Myc into existing/remaining peroxisome was performed by calculating Manders' overlap coefficients (OC) using colocalization plugins in the Image J soft. An OC equivalent to 1 was defined as 100%, and each individual OC was expressed relative to this 100% value. Data represents mean \pm SD (n=3 experiments), 30 transfected cells were scored per condition per experiment. *P < 0.05, Student's t-test.

2) Does depletion of TMEM135 with siRNA really affect the staining pattern of endogenous Tmem135? Please exclude that the peroxisomal localization is the result of overexpression.

Our Reply:

We thank the Reviewer for raising this important issue related with TMEM135 siRNA efficiency and the specificity of TMEM135 antibody. All three TMEM135 siRNA used in this study significantly decreased the TMEM135 fluorescent signal from the peroxisomes (Figure R5), suggesting that TMEM135 siRNA and TMEM135 antibody employed in this study are very specific. As, endogenous TMEM135 stained with TMEM135 antibody colocalizes with peroxisomes, transfected TMEM135-Myc and EGFP-TMEM135 colocalizes with vesicular structures stained with TMEM135 antibody and peroxisome marker, and the possible involvement of Pex16 for TMEM135 localization to peroxisomes clearly indicate that TMEM135 localization to peroxisome is not an overexpression artifact.

Figure R5: TMEM135 siRNA efficiently reduces TMEM135 fluorescent signal from peroxisomes. Representative image showing that all three siRNA efficiently reduces TMEM135 protein from peroxisomes stained with PMP70. Scale bar, 10 μ m.

3) While three independent siRNAs are available, most experiments are done with only one of those. This neglects the possibility of off-target effects.

Our reply:

All three TMEM135 siRNA used in this efficiently decreases endogenous TMEM135 protein as shown in Appendix Figure S2A and Appendix Figure S2E. We have performed key experiments

like ciliogenesis assay using all three siRNA (Fig 2A-B) and cholesterol accumulation assay by filipin (Figure EV1E) by using two TMEM135 siRNA. As shown in Figure R5, all three siRNA efficiently depletes TMEM135 from peroxisomes, all the effect of TMEM135 depletion would be similar and the possibility of off-target effects seem negligible at least for the assays performed in this study.

4) The experiment presented in Fig. 2f is unclear. What is the exact time course of this experiment? Does the chol-MBCD treatment affect the knockdown of TMEM135, does it affect transfection efficiency? TMEM135 levels should be shown for all points.

Our reply:

Cholesterol-MCD complex is a water-soluble cholesterol which do not need intracellular cholesterol transport system. TMEM135 depletion impairs intracellular cholesterol transport resulting in uneven cholesterol distribution in the cell by sequestration of cholesterol in lysosomal compartment (Fig1 and EV1). The experiment presented in the Fig 2F suggested that Cholesterol-MCD rescues ciliogenesis in TMEM135 depleted cells possibly by cholesterol redistribution. TMEM135 was depleted for 48 hours followed by serum-starvation with or without different doses of cholesterol-MCD for additional 24 hours. Excluding the time from seeding of cells to siRNA treatment, the experiment shown in Fig 2F was carried out for 22 hours. As shown Figure R6, TMEM135 knockdown efficiency was not acted by Cholesterol-MCD. TMEM135 levels have been shown in all the additional experiments performed.

Figure R6: TMEM135 siRNA efficiency was unaffected by Cholesterol-MCD. (A) TMEM135 knockdown cells were treated with 18 μ g cholesterol-MCD in a serum starvation medium and TMEM135 protein levels were analyzed by Western blot. **(B)** Source data for (A).

5) How does the inhibition of chol-synthesis by statins affect Tmem135 depletion? Statins/Cholesterol and ciliogenesis have been linked recently in zebrafish.

Our Reply

We investigated the effect of inhibition of cholesterol synthesis in TMEM135 depleted cells in terms of the ciliogenesis phenotype. We seek to find out if cholesterol depletion really depletes cholesterol in basal conditions like serum-fed and serum-starved conditions. First, we analyzed the effect of compactin (statin) on the TMEM135 protein level. Compactin treatment. As shown in Figure R7-A, Compactin treatment did not affect the TMEM135 siRNA efficiency and TMEM135 protein level in Control siRNA treated cells. Next, we measured the total cellular cholesterol as shown in Figure R7-B. TMEM135 depletion does not alter cellular cholesterol in both serum-fed and serum-starved conditions. It is interesting to note that even the addition of compactin did not decrease total cholesterol in the serum-fed condition in both Control siRNA and TMEM135 siRNA treated cells, suggesting that cells depend upon serum-cholesterol for the maintenance of total cellular cholesterol levels rather than cholesterol biosynthesis. Compactin treatment slightly decreased the total cholesterol in serum-starved cells suggesting that endogenously synthesized cholesterol does contribute for the maintenance of total cellular cholesterol during serum starvation. Therefore, the accumulated cholesterol observed during serum-starvation in TMEM135 depleted cells (Fig 2C) should have been derived from serum-cholesterol before subjecting cells to serum-starvation. Importantly, even the addition of compactin during serum-starvation stills results in lysosomal cholesterol accumulation as shown in Figure R7-C, confirming that most of the accumulated

cholesterol was obtained from serum-cholesterol rather than endogenously synthesized cholesterol in TMEM135 depleted cells.

Surprisingly, compactin decreases the percentage of ciliated cells in control cells as previously reported (16), while the ciliated cells were not further decreased by compactin in TMEM135 depleted cells (Figure R7-D and E). Furthermore, Rab8 trafficking to centrioles was not affected by compactin in Control siRNA treated cells (Figure 7-F), suggesting that statin mediated defective ciliogenesis occurs via another pathway as reported (16). TMEM135 depletion shows Rab8 trafficking defect with or without compactin (Figure 7-F). As this experiment was only designed to investigate defective ciliogenesis within 24 hours serum-starvation, it is also important to know the cholesterol amount in the cells treated with compactin for longer duration like 72 hours in serum-starvation which is not available in a previous report (16). This becomes important on comparing the effect of TMEM135 depletion and statin treatment on ciliogenesis, as 24 hours statin treatment is not as potent as TMEM135 depletion in decreasing the percentage of ciliated cells.

Figure R7: Statin treatment does not affect TMEM135 depletion associated cilia phenotypes. (A) TMEM135 knockdown cells were treated with 100 μ M compactin in a serum starvation medium and TMEM135 protein levels were analyzed by Western blot. **Source data are also provided. (B)** Statin decreases total cholesterol in serum-starvation but not in serum-fed condition. Total cholesterol was measured from TMEM135 depleted cells with or without compactin. Control siRNA treated cells in serum-fed condition was taken as control for making relative total cholesterol levels. Data represents mean \pm SD (n=3 experiments)*P < 0.05, Student's t-test. **(C)** TMEM135 depletion mediated cholesterol accumulation is not affected by statin. Representative images showing filipin staining (blue). Scale Bar, 20 μ m. **(D)** TMEM135 depletion mediated decreased ciliogenesis is not further decreased by statin. Representative images showing ciliated cells, immunostained by cilia marker ARL13B. **(E)** The bar graph shows percentage of ciliated cells in each condition. Data represents mean \pm SD (n=3 experiments), 200 cells were scored per condition per experiment. *P < 0.05, Student's t-test. **(F)** Statin does not affect Rab8 trafficking to centrosomes. Representative fluorescent images immune-stained with Rab8 (Red) and γ -tubulin (green) are shown. Scale bar, 5 μ m. Arrowhead indicate the centriolar localization or Rab8.

6) Does the depletion of TMEM135 affect cells that already carry cilia? Since Shh signaling depends on cholesterol: Is Shh signaling affected by Tmem135?

Our Reply

It is well known that Sonic Hedgehog (Shh) requires cholesterol modification for Hedgehog signaling. Also, Hedgehog (HH) signaling requires functional cilium. Since TMEM135 depletion affects cellular cholesterol homeostasis it is reasonable to surmise that TMEM135 depletion might affect Hedgehog signaling as it has been shown that defective cholesterol biosynthesis suppresses HH signaling (16). We analyzed the effect of TMEM135 on the expression of HH signaling genes by qPCR. Purmorphamine (Smoothed agonist) was used to induce hedgehog signaling. As shown in Figure R2-B, Purmorphamine induces expression of Gli1, Gli2, and Gli3 in control cells but the expression of these genes was blunted by TMEM135 depletion even in the presence of Purmorphamine. This data suggests that TMEM135 suppress HH signaling, which could be due to the decreased ciliogenesis or by a direct effect of cholesterol accumulation on processing of Shh.

Figure R8:TMEM135 depletion suppresses Hedgehog signaling. (A) Experimental design. (B) qPCR analysis of expression of genes associated with Hedgehog signaling in presence or absence of 10 μ M Purmorphamine. Data represents mean \pm SD (n=3 experiments). *P < 0.05, Student's t-test. NS, not significant.

We agree with the reviewer that the effect of TMEM135 depletion on already existing cilia would be required to suggest if the effect of TMEM135 depletion on HH signaling was due to impaired ciliogenesis or cholesterol accumulation. Therefore, we designed the experiment to induce ciliogenesis in serum-fed condition using Cytochalasin D (actin-depolymerizing agent), which is known to efficiently induce ciliogenesis even in subconfluent cells (17). As shown in Figure R9-A, Cytochalasin was treated for 24 hours followed by siRNA treatment without changing medium and incubated for an additional 48 hours. TMEM135 siRNA seems to work well in this design as suggested by the robust depletion of TMEM135 protein (Figure R9-B). As Cytochalasin D results in cell cycle arrest, we can disregard the possibility of ciliogenesis induced by cell confluency during the time course of the experiment. Thus, we can speculate that Cytochalasin D induces a similar number of ciliated cells before siRNA treatment and any changes in the percentage of ciliated cells after siRNA treatment could be solely attributed to the effect of respective siRNA. Surprisingly, TMEM135 siRNA significantly reduces the percentage of ciliated cells and cilium length in presence of Cytochalasin D (Fig R9-C, D, E). These data suggest that TMEM135 depletion could negatively affect existing cilium. However, more work is needed, our data suggest that suppression of Hedgehog signaling could be associated with impaired ciliogenesis rather than direct effect of cholesterol accumulation in TMEM135 depleted cells. This experiment also indicates the possible role of Rab8 in the maintenance of existing cilium as TMEM135 depletion impairs Rab8 trafficking.

Figure R9: TMEM135 depletion negatively affects existing cilium. (A) Experimental design. (B) TMEM135 siRNA efficiency by Western blot with source data. (C) TMEM135 depletion reduces Cytochalasin D (200 nM) mediated ciliogenesis. Representative images showing ciliated cells, immunostained by cilia marker ARL13B. (C) The bar graph shows percentage of ciliated cells in each condition. * $P < 0.05$, Student's t-test. (D) Whisker and box plot of cilium length in each condition where lines represent one-way standard deviation. Plotted are a total of 150 cells from three independent experiment. * $P < 0.05$, Student's t-test.

7) Western blot should be shown with molecular weight standards.

Our reply

We have now incorporated molecular weights in all western blots of main figures, expanded view figures and appendix figures. We have also included uncropped Western blots with standard molecular weight as a separate source file.

Additional materials used for the experiments presented in this review

Pex16 antibody (14816-1-AP, proteintech), Purmorphamine (SML0868), Cytochalasin D (C8273), Cholesterol assay kit (K603, Biovision) were used.

Pre-designed human siRNA against ABCD1 (5'-AGGAGGAGCUGGUGAGCGA-3'), Pex16 (5'-AGCAGCAUCAGGAGCU-3'), Pex3 (5'-CUGUAUGCUGGUUGUUCUU-3') were used. The forward and reverse human primer sequences used for qPCR are as follows:

Gli1 (5'-TGTGGGGACAGAAGGACTGT-3' and 5'-CCGGACATGAGGTTAGCTTG-3'), Gli2 (5'-AAGGAGAGGGGACTGTTTGG-3' and 5'-ACGAGGGTCATCTGGTGGT-3'), Gli3 (5'-CTGCTGCTGCACAAACGA-3' and 5'-GGTTTGTACGCATTCTTTGGA-3'), Smoothed (5'-TACGTCATGCGTGCTTCTT-3' and 5'-GGGCACCATCCATGAACT-3'), Patched1 (5'-GGGATAAAAGCAGCGAACC-3' and 5'-ACTCGTCCTCCAACCTCCAC-3'), PEX3 (5'-TCTGGGAAATATGGACAGAA-3' and 5'-TCGTGCTTGGGCAATGTAT-3'), PEX16 (5'-CAAGGTGTGGGGTGAAGTG-3' and 5'-TCCGCAGTACAGCCTTGG-3') Pex19 (5'-

TGAGGAAGGCTGTAGTGTCG-3' and 5'-AATCATCAAGAGCACTTTCCAGA-3'), ABCD1 (5'-GTGGCTGTCCCCATCATC-3' and 5'-CCTTCTTCACGGCCTCTG-3').

References related to Referee #1

1. Islinger M, Lüers GH, Li KW, Loos M, Völkl A (2007) Rat liver peroxisomes after fibrate treatment. A survey using quantitative mass spectrometry. *J Biol Chem* **282**: 23055–23069
2. Wiese S, Gronemeyer T, Ofman R, Kunze M, Grou CP, Almeida JA, Eisenacher M, Stephan C, Hayen H, Schollenberger L, Korosec T, Waterham H.R, Schliebs W, Erdmann R, Berger J, Meyer HE, Just W, Azevedo JE, Wanders RJ, Warscheid B. (2007) Proteomics characterization of mouse kidney peroxisomes by tandem mass spectrometry and protein correlation profiling. *Mol Cell Proteome* **6**: 2045–2057
3. Lee WH, Higuchi H, Ikeda S, Macke EL, Takimoto T, Pattnaik BR, Liu C, Chu LF, Siepka SM, Krentz KJ, Rubinstein CD, Kalejta RF, Thomson JA, Mullins RF, Takahashi JS, Pinto LH, Ikeda A (2016) Mouse Tmem135 mutation reveals a mechanism involving mitochondrial dynamics that leads to age-dependent retinal pathologies. *eLife* **5**: e19264
4. Blacque OE, Scheidel N, Kuhns S (2018) Rab GTPases in cilium formation and function. *Small GTPases* **9**:76-94
5. Liu Q, Insinna C, Ott C, Stauffer J, Pintado PA, Rahajeng J, Baxa U, Walia V, Cuenca A, Hwang YS, Daar IO, Lopes S, Lippincott-Schwartz J, Jackson PK, Caplan S, Westlake CJ (2015) Early steps in primary cilium assembly require EHD1/EHD3-dependent, ciliary vesicle formation. *Nat Cell Biol* **17**: 228–240
6. Sato T, Mushiaki S, Kato Y, Sato K, Sato M, Takeda N, Ozono K, Miki K, Kubo Y, Tsuji A, *et al.*, (2007) The Rab8 GTPase regulates apical protein localization in intestinal cells. *Nature* **448**:366-369
7. Sato T, Iwano T, Kunii M, Matsuda S, Mizuguchi R, Jung Y, Hagiwara H, Yoshihara Y, Yuzaki M, Harada R, *et al.*, (2014) Rab8a and Rab8b are essential for several apical transport pathways but insufficient for ciliogenesis. *J Cell Sci* **127**:422-431.
8. Chu BB, Liao YC, Qi W, Xie C, Du X, Wang J, Yang H, Miao HH, Li LB, Song BL (2015) Cholesterol transport through lysosome peroxisome membrane contact. *Cell* **161**: 291–306
9. Benjamin JR, Thushara WM, Jon DH, *et al.*, (2018) TEM135 is an LXR-inducible regulator of Peroxisomal metabolism. *bioRxiv* doi: <https://doi.org/10.1101/334979>
10. Sugiura A, Mattie S, Prudent J, McBride HM (2017) Newly born peroxisomes are hybrid of mitochondrial and ER-derived pre-peroxisomes. *Nature* **542**: 251-254
11. Koyano F, Yamano K, Kosako H, Kimura Y, Fujiki Y, Tanaka K, Matsuda N (2019) Parkin-mediated ubiquitylation redistributes MITOL/March 5 from mitochondria to peroxisomes. *EMBO Rep* e47728
12. Dixit E, Boulant S, Zhang Y, Lee As, Odendall C, Shum B, Hacohe N, Chen ZJ, Whelan SP, Fransen M, Nibert ML, Superti-Furga G, Kagan JC (2010) Peroxisomes are signaling platforms for antiviral innate immunity. *Cell* **141**: 668-681
13. Jones JM, Morrell JC, Gould SJ (2004) PEX19 is a predominantly cytosolic chaperone and import receptor for class I peroxisomal membrane proteins. *J cell Biol* **164**: 57-67
14. Aranovich A, Hua R, Rutenberg AD, Kim PK (2014) PEX16 contributes to peroxisome maintenance by constantly trafficking PEX2 via the ER. *J Cell Sci* **127**: 3675-3686
15. Hua R, Cheng D, Coyaud E, Freeman S, Di Pietro, Wang Y, Vissa A, Yip CM, Farin GD, Braverman N, Brumell JH, Trimble WS, Raught B, Kim PK (2017) VAPs and ABCD5 tether peroxisomes to the ER for peroxisome maintenance and lipid homeostasis. *J Cell Biol* **216**:367-377
16. Maerz LD, Burkhalter MD, Schilpp C, Wittekindt OH, Frick M, Philipp M (2019) Pharmacological Cholesterol depletion disrupts ciliogenesis and ciliary function in developing zebrafish. *Commun Biol* **2**:31
17. Kim J, Jo H, Hong H, Kim MH, Kim JM, Lee JK, Heo WD, Kim J (2015) Actin remodeling factors controls ciliogenesis by regulating YAP/TAZ activity and vesicle trafficking. *Nat Commun* **6**: 6781

Referee #2:

This report presents an interesting observation, namely that a peroxisome localized protein influences lysosomal cholesterol levels and appears to influence Rab8 activation and localization. The findings could be of broad interest. However there are a number of aspects of the study that need additional work before presentation in EMBO Reports can be recommended.

Our Reply:

We thank the Reviewer for the positive assessment of our work. In response to the Reviewer's comment, we have included the results of additional experiments. We believe that these new results would effectively address the Reviewer's concerns.

1. Previous work has shown that knockout of TMEM135 leads to a strong block in the early secretory pathway (Simpson et al. Nat Cell Bio) and also, the protein was characterized by others as causing a defect in peroxisomal long chain fatty acid oxidation, leading to an overall accumulation of triglycerides (in worm and HepG2 cells) and cell death 48-72 hours post siRNA. A recent paper on BioRxiv concluded, "Knockdown of TMEM135 in HepG2 cells caused triglyceride accumulation, indicating a potential role in [peroxisomal] β -oxidation. ...in fasted TMEM135 knockdown mice, there was a further significant increase in hepatic fatty acid concentrations and a significant decrease in NADH." If this is correct, the question is whether the block in ciliation and block in Rab8 activation reported here are specific for the ciliation pathway or are much more general. Are the cells low in ATP and GTP altogether? This would explain all of the phenotypes observed herein and needs to be examined. Aside from this significant concern, there are other issues that need to be clarified.

Our reply:

We thank the reviewer for making us aware of a previous work by Simpson *et al.*, which suggests the possible localization of TMEM135 in the ER and its depletion impairs the early secretory pathway (1). Two independent studies of rat and liver peroxisome proteomics had identified TMEM135 (originally referred to as peroxisome membrane protein 52, PMP 52) as a peroxisome localized protein (2, 3). A recent study of Benjamin *et al.*, also indicates that TMEM135 is a peroxisome localized protein (4). In agreement with previous reports, our data also suggest that both the ectopic and endogenous TMEM135 localize to peroxisomes. Furthermore, TMEM135 possibly require Pex16 function for its localization into peroxisomes as in the Pex16 depleted cells, ectopic TMEM135 predominantly localized in the ER (this experiment was performed in response to the concerns of Reviewer #1 regarding TMEM135 localization and has been included in the respective Reviewer's comment section as Figure R4). Therefore, the ER localization of TMEM135, as observed by Simpson *et al.*, could be due to the overexpression artifact or associated with the dynamics of Pex16 in a condition being tested.

It is surprising that the study by Simpson *et al.*, has identified TMEM135 as one of the candidates controlling the early secretory pathway. TMEM135 has been previously implicated to play a role in lysosome-peroxisome membrane contact and depletion of TMEM135 decreases PI(4,5)P2 on peroxisome membrane thereby reducing the ability of peroxisome to interact with lysosomal Synaptotagmin VII, resulting in robust cholesterol accumulation in the lysosomes (5). One of the peculiar phenotypes associated with lysosomal cholesterol accumulation is that cholesterol cannot reach the ER and get esterified. This results in aberrant activation of SREBP2 as shown in Fig 1E-F. In response to the low ER cholesterol, SREBP2 is transported from ER to Golgi through the action of SCAP (escort protein). SCAP has binding sites for COPII proteins, which cluster the SCAP-SREBP2 complex into COPII-coated vesicles, buds from the ER membrane and finally fuses with Golgi where SREBP2 is processed into its active form for nuclear translocation. Therefore, the processing of SREBP2 is the classical example of a secretory pathway from ER to Golgi (6). Aberrant SREBP2 activation observed in TMEM135 depletion suggests that ER to Golgi secretory pathway might not be affected or blocking the early secretory pathway by TMEM135 depletion might be very specific (other than SREBP2 processing). Since Rab8 has been reported to regulate the basolateral secretory pathway (7, 8, 9) it is plausible that TMEM135 depletion might at least impairs the basolateral secretory pathway associated with dysfunctional Rab8 trafficking and activation.

TMEM135 knockout has been reported to decrease lipid droplets in *C. elegans* (10), in contrast, the recent study of Benjamin *et al.*, show that siRNA mediated TMEM135 depletion in HepG2 cells results in the accumulation of triacylglycerides (4). To investigate the possible function of TMEM135 on lipid metabolism, mRNA expression of genes involved in the mitochondrial and

peroxisomal β -oxidation as well as fatty acid synthesis were analyzed. As shown in Figure R10, TMEM135 depletion did not alter the mRNA expression of PPAR- α , PGC1- α , CPT1- α , MCAD, AOX1, HSD17B4, SREBP1c and FAS. These results are in agreement as reported by Benjamin *et al.*, (4), that TMEM135 depletion did not alter mRNA expression of the genes associated with β -oxidation as well as fatty acid synthesis.

Figure R10: TMEM135 depletion does not affect expression of gene associated with β -oxidation and fatty acid synthesis. QPCR analysis of expression of genes associated with β -oxidation and fatty acid synthesis in RPE1 cell cultured in serum-fed medium. Data represents mean \pm SD (n=3 experiments). *P < 0.05, Student's t-test. NS, not significant.

Although, TMEM135 depletion did not alter genes associated with β -oxidation and fatty acid synthesis, lipid droplet was found to increase even in the basal condition in HepG2 cells (4). Moreover, TMEM135 depletion further increased the lipid droplet accumulation in the presence of LXR agonist (T09) while suppressing the T09 mediated enhanced SREBP1c expression suggesting a possible role of TMEM135 in β -oxidation (4). Therefore, we investigated the possible involvement of TMEM135 on β -oxidation by analyzing the TG accumulation. As shown in Figure R11, TMEM135 depletion did not increase lipid droplets in the basal condition and is contradictory to those reported by Benjamin *et al.*, (4). Interestingly, Oleic acid treatment slightly increased lipid droplets in TMEM135 depleted cells as compared with the respective control group.

Figure R11: TMEM135 depletion slightly increases Oleic acid mediated lipid droplet formation. Representative fluorescent images of RPE1 cells stained with Oil-Red-O in a serum-fed condition with or

without 100 μ M Oleic acid. 50 cells were scored per condition per experiment. Data represents mean \pm SD (n=3 experiments). *P < 0.05, Student's t-test. NS, not significant. Scale bar, 10 μ m.

Considering that TMEM135 depletion suppressed T09 induced fatty acid synthesis genes but facilitates lipid droplet accumulation due to impairment in β -oxidation (4), we hypothesize that, if TMEM135 really plays a role in β -oxidation, overexpression of TMEM135 might decrease lipid droplets even in the presence of Oleic acid. As shown in Figure 12, TMEM135-Myc expressed cells show a considerably lesser number of lipid droplets as compared to untransfected cells in the presence of Oleic acid suggesting that TMEM135 might play a role in lipid metabolism. Thus, for the elucidation of TMEM135 function, special conditions would be required. Furthermore, TMEM135 depletion did not result in the accumulation of VLCFA suggesting that peroxisomal β -oxidation was not impaired. (Appendix Figure S2D). Since TMEM135 depletion did not present any reliable phenotype associated with lipid metabolism in basal condition and only observable phenotype is robust lysosomal cholesterol accumulation and defective ciliogenesis, it is unlikely that lipid metabolism associated role of TMEM135 is involved in ciliogenesis.

Figure R12: Overexpression of TMEM135-Myc reduces lipid droplet number in presence of Oleic acid. Representative fluorescent images of RPE1 cells stained with Oil-Red-O in a serum-fed condition with or without 100 μ M Oleic acid. 30 transfected and untransfected cells were scored per condition per experiment. Data represents mean \pm SD (n=3 experiments). *P < 0.05, Student's t-test. NS, not significant. Scale bar, 10 μ m.

Benjamin et al., have further suggested that, TMEM135 depletion in HepG2 decreases cell viability and ATP levels when cultured in HBSS medium (4) Therefore, we agree with the Reviewer that if ATP levels or GTP levels were altered by TMEM135 depletion, the overall ciliary phenotypes

observed would be associated with a general effect rather than the specific effect of TMEM135 depletion on ciliogenesis. As shown in Figure R13, TMEM135 did not decrease cell viability. MTT assay shows that serum-starvation slightly decreases the viable number of cells in both control and TMEM135 depleted cells which could be due to cell cycle arrest induced by serum-starvation.

Figure R13: TMEM135 depletion does not decrease cell viability in RPE1 cell. Cell viability was measured by using 3-(4,5-dimethylthiazol-2-yl)-2,5-diphenyltetrazolium bromide (MTT assay). Data represents mean \pm SD (n=3 experiments). *P < 0.05, Student's t-test. NS, not significant.

To further confirm if TMEM135 depletion was associated with cell death, Tunnel assay was carried out. As shown in Figure R14, positive control showed tunnel positive nucleus but TMEM135 depletion did not show tunnel positive nucleus in both serum-fed and serum-starved conditions. Taken together, our data suggest that TMEM135 depletion did not induce cell death.

Figure R14: TMEM135 depletion does not induce cell death in RPE1 cell. TUNEL staining was performed using In Situ cell death detection kit (Roche). Scale bar, 20 μ m.

Next, we examined the total cellular ATP level. As shown in Figure R15, TMEM135 depletion did not affect the total cellular ATP level in serum-fed or serum-starvation conditions. Since our study is mainly associated with ciliogenesis, we did not analyze the ATP level in HBSS condition as shown by Benjamin et al., (4).

Figure R15: TMEM135 depletion does not affect total cellular ATP. ATP levels were measured using ATP assay kit (biovision). Data represents mean \pm SD (n=3 experiments). *P < 0.05, Student's t-test. NS, not significant.

The possibility that TMEM135 depletion could decrease the GTP level is very minute for the following reason:

- 1) However, TMEM135 depletion decreases the level of Rab8-GTP level (Fig 4C), the amount of GTP bound Constitutively active (CA) form of Rab8 in control and TMEM135 depleted cells are similar (Fig. 4D).
- 2) TMEM135 depletion mediated impaired Rab8 activation might be very specific as Rab27A-GTP (which also binds to JCF1 as Rab8 when it is activated) level are similar in control and TMEM135 depleted cells. This experiment was part of this Reviewer 's comment and have been discussed with the data in the respective comment section.
- 3)

Thus, TMEM135 depleted cells have enough GTP to activate endogenous Rab27A or ectopic CA Rab8, suggesting that TMEM135 depletion might not affect the total cellular GTP.

2. Fig. 1 A,B. What do the authors mean by lysosome-peroxisome overlap (%)? Is this percent of lysosomes showing overlap or percent of peroxisomes showing overlap? Why is there less fluorescence green and red, in the TMEM135 examples shown? Please include corresponding quantitation of total fluorescence and numbers of structures

Our Reply:

Lysosome-Peroxisome overlap means lysosomes making a contact with peroxisomes. In the data presented in Fig 1A, B, the portion of lysosome overlapping with peroxisome was evaluated with Colocalization plugin for the Image J software by calculating Mander's overlap coefficients (OC), which allow the quantification of overlapping pixels from each channel. For the quantification of the percentage of lysosome-peroxisome overlap, the OC of lysosome fluorescent signal overlapping with peroxisome fluorescent signal was quantitatively assessed. OC equal to 1 were considered as 100% and each individual OC was expressed relative to 100%. We have now measured the fluorescent intensity and the average number of peroxisome and lysosome per cell and have been included in the Fig EV1A-D. TMEM135 did not alter either fluorescent intensity or number of lysosome and peroxisomes per cell.

3. Fig 1C. The method used is not that described in ref. 45. With due respect to the authors, this reviewer believes that the bottom three rows of the blots may be flipped. Golgi (GM130) and plasma membranes (ATPase) have the same buoyant density as early endosomes (EEA1) and so it is very hard to imagine how they could have spun to the bottom of the gradient described in 90 minutes. They are usually less dense than lysosomes, or?

Our reply:

We thank the Reviewer for pointing our mistakes related with the reference and time duration of ultracentrifugation. We sincerely apologize for our mistakes. The reference has been changed in the manuscript. The ultracentrifugation for the sucrose gradient results shown in Fig 1C was carried out for 8 hours at 4°C and de-accelerated without applying brakes. The method described for subcellular fractionation by sucrose floatation gradient has been modified in the 'materials and method' section of the revised manuscript. The blots of Golgi (GM130), Plasma membrane (Na⁺/K⁺ ATPase) and peroxisomes (PMP70) were presented in their correct orientation and they were not flipped as Reviewer have speculated. Although, it is known that the rough buoyant densities of Lysosomes, early endosomes, plasma membrane and Golgi derived vesicles (not sure about Golgi fragments) are similar, the actual buoyant density of intracellular organelles is dependent in part on the nature of the buffer composition of the density gradient and the permeability characteristics of the organelle membrane to constituents of the gradient (suggesting that density of an organelle is not constant). For instance, early endosomes (EEA1), lysosomes (LAMP2), and Na⁺/K⁺ ATPase are resolved in same fraction (usually lighter fraction) when the post nuclear supernatant is rapidly centrifuged within shorter duration and de-accelerated with maximum brakes in a Percoll gradient (11, 12). Furthermore, EEA1, LAMP1 and Na⁺/K⁺ ATPase are well separated in a continuous sucrose gradient where, EEA1 is mostly present in the lighter fraction than LAMP1 while Na⁺/K⁺ ATPase is mostly resolved in the heaviest fraction (13). Previous report also suggests that Golgi (probably intact or Golgi fraction but not Golgi derived vesicles) get resolved in heavier fraction in a discontinuous sucrose gradient (14). Therefore, we believe that distribution of organelles across the gradient could be variable depending upon the homogenization methods applied, duration of ultracentrifugation, type and concentration of gradient used, type of rotor and de-acceleration rate.

4. Why is there a discrepancy between JCF1 binding in Fig. 4 and 7C? The blots were done using a peroxidase method that is nonlinear and will exaggerate differences. Please add quantitation

with a standard curve or use LICOR for a linear response. More importantly, ESSENTIAL: please show that the defect is specific to Rab8. In other words, does another Rab with no link to ciliation also have activation problems under these conditions?

Our Reply:

We agree with the Reviewer that blots were developed using peroxidase method. Comparing the blots generated with certain exposure time with another set of data with different exposure time is not the standard practice. We acknowledge that, the data shown in Fig 4C and 7C shows different amount of endogenous Rab8-GTP being pull down by same amount of GST-JCF1. This is due to difference in exposure time applied for developing the blots between two different experiment. Therefore, we have incorporated new data for Fig 7C using similar exposure time as in Fig 4C and Fig EV5 for developing the blots. The data shows similar results as previous data that Cholesterol-MCD recovers Rab8 activation in TMEM135 depleted cells. The new data has been replaced as the same figure designation (Fig 7C). Although, the amount of Rab8-GTP pull down by same amount of GST-JCF1 might be different due to different exposure time when comparing with data sets generated at different time period, the meaning of the data remains same until the comparison is made within the same set of data as a 'fold of control or normalized with the control level. Therefore, the quantitative data representing amount of Rab8 bound to GTP has been incorporated as previously reported (15) for all the GST-JCF1 pull down of endogenous Rab8 (Fig 4C, Fig 7C and Fig EV5C). The quantitative data has been generated by calculating the intensity of Rab8-GTP (Pull down) to intensity of Rab8 (Input) by Image J software and comparing the differences in the ratio by considering ratio of the respective control group as 1 (fold of control).

We thank the Reviewer for asking this important question that if TMEM135 depletion mediated Rab8 activation defect is specific. We choose to analyze the activation state of Rab27A in TMEM135 depleted cells as Rab27A has not been implicated in ciliogenesis and binds with the common effector molecule JCF1 in its activated form as Rab8 does (16, 17). Pull down assay for Rab27A-GTP was performed using purified GST-JCF1 as described in the manuscript (materials and method section). As shown in Figure R16, TMEM135 depletion did not affect the activation state of endogenous Rab27A suggesting that the effect of TMEM135 depletion on Rab8 activation seems to be very specific.

Figure R16: TMEM135 depletion does not affect Activation of Rab27A. Cells were transfected as indicated and the cell lysates were incubated with purified GST-JCF1 (RBD). The amount of GTP-Rab27A bound to GFT-JCF1(RBD) was analyzed by western blot with Rab27A antibody. Bar graphs represents the Rab27A-GTP. Data represents mean \pm SD (n=3 experiments). *P < 0.05, Student's t-test. NS, not significant.

5. The Rab8 compartment fails to coalesce perinuclearly. ESSENTIAL: Do the Golgi and lysosomes also show a different morphology? Previous work reported changes in overall mitochondrial morphology. This could be due to a general microtubule or motor defect and would be essential to address.

Our Reply:

TMEM135 depletion does not affect Golgi morphology and Golgi still coalesces in the perinuclear region (Figure R17). Condensation of Golgi are seen in some cells in during TMEM135 depletion.

Figure R17: TMEM135 depletion does not alter Golgi morphology and organization. Representative fluorescent images showing GM130 (green) and Nucleus/DAPI (blue). Scale bar, 10 μ m.

As discussed early, TMEM135 depletion did not change the average lysosome puncta per cell (included in Fig EV1 and Fig EV1C of the revised manuscript). Since lysosomes form punctate structures, analysis of its morphology might be difficult. Only morphological changes that could be analyzed with fluorescent images would have been the abnormal size or dispersion of lysosome marker in the cytosol rather than forming distinct punctate structure and these of morphological anomalies were not observed. Although TMEM135 depletion results in lysosomal cholesterol accumulation, staining was slightly enhanced in TMEM135 depleted cells. Taken together, our data suggest that TMEM135 does not result in any observable lysosomal morphological abnormalities.

The previous report by Lee *et al.*, have suggested that TMEM135 regulates mitochondrial dynamics, specifically, TMEM135 overexpression results in mitochondrial fission and TMEM135 resulting in severe mitochondrial fragmentation. depletion enhances mitochondrial fusion resulting in elongated mitochondria (18). As shown in Appendix Figure S1B, lower panel, TMEM135-myc overexpressed cell did not show fragmented mitochondria rather mitochondria are elongated in structure. As shown in Figure R18, TMEM135 depletion did not result in observable changes in mitochondrial morphology. Therefore, our data suggest that TMEM135 depletion does not alter mitochondrial morphology.

Figure R17: TMEM135 depletion does not alter mitochondrial morphology. Representative fluorescent images showing Tomm20 (red) and Nucleus/DAPI (blue). Scale bar, 10 μ m.

In order to investigate if TMEM135 depletion causes defects in microtubule, we first analyzed the bulk microtubule distribution. As shown in Figure R18-A, TMEM135 depletion results in curls, and buckle with some breakage in the microtubule thread at the edges of the cell, suggestive of depolymerized microtubule (19). The amount of alpha-tubulin level was not affected by TMEM135 depletion Figure R18-B. Next, we investigated that if depolymerized microtubule observed in

TMEM135 depleted cells affects the distribution of the motor protein. Surprisingly, we did not find any changes in the distribution of Dynein heavy chain in TMEM135 depleted cells even though, microtubule shows extensive curls and buckle at the edges of the cell (Figure R18-C). Thus, we suggest that TMEM135 depletion might affect microtubule re-polymerization or causes enhanced depolymerization. Next, we speculated that TMEM135 depletion might affect microtubule nucleation and anchoring (this was one of the concerns of Reviewer #3 and the Figure R19 have been included in the comment #1 of Reviewer#3). Surprisingly, TMEM135 depletion did not affect microtubule nucleation (1 min regrowth after nocodazole wash out) and anchoring (10 min regrowth after nocodazole washout) in a microtubule regrowth assay after nocodazole washout (Figure R19). Therefore, extraction and quantification of tubulin fraction would be required for the analysis of polymeric and monomeric tubulin in order to understand the true status of microtubule dynamics and defects associated with depolymerization or re-polymerization. Further studies are needed to establish a relationship between Rab8 trafficking and microtubule dynamics and how these processes are influenced by cellular cholesterol distribution.

Figure R18: TMEM135 depletion affects microtubule dynamics but does not alter dynein distribution. (A) Representative fluorescent images showing alpha-tubulin (red) and Nucleus/DAPI (blue). Scale bar, 10 μ m. (B) alpha-tubulin was analyzed by immunoblotting in (C) Representative fluorescent images showing alpha-tubulin (magenta), dynein heavy chain (green) and Nucleus/DAPI (blue). Scale bar, 10 μ m.

Figure R19: TMEM135 depletion affects microtubule nucleation and anchoring. Cells grown in coverslips were transfected by siRNA for 48 hours followed by serum starvation for additional 24 hour. Nocodazole (10 μ M) was treated for treatment for 1 hour at 37°C in DMEM for the depolymerization of microtubule, washed with DMEM, and incubated at 37°C for a specified time period to allow microtubule regrowth, followed by fixation with cold methanol for 5 min at -20° and immunostaining with α -tubulin and γ -tubulin antibodies. Representative fluorescent images showing alpha-tubulin (red), γ -tubulin (green), and Nucleus/DAPI (blue). Scale bar, 10 μ m.

In summary, understanding the role of TMEM135 is interesting and important. This reviewer would prefer to see some more precise hint as to why Rabin 8 is not functioning under the experimental conditions employed. However it is likely a more general cellular defect, and more work is needed to understand the phenotypes observed before presentation in EMBO Reports can be recommended.

Our reply

In the revised manuscript, we have included new data showing that TMEM135 depletion prevents ciliary vesicle (CV) extension using EHD1 as a CV marker (the issue was raised by Reviewer #3 that inclusion of another marker is required to show that TMEM135 depletion does not prevent ciliary vesicle formation). EHD1 is essential protein required for CV formation and has been used as CV marker along with other CV membrane proteins like Rab8, Smo, ARL13B (20, 21). In TMEM135 depleted cells, single EHD1 vesicle are found to localize at the distal end of the basal body while EHD1 is localized in the cilium in the control cells suggesting that TMEM135 depletion did not prevent CV formation at the basal body but prevents extension of CV (Fig EV4C, D). Since, our data indicate that constitutively active (CA) GFP-Rab8 could be activated and rescues cilia formation while wild type (WT) GFP-Rab8 could not be activated and rescue ciliogenesis in TMEM135 depleted cells, we assumed that impaired ciliary vesicle extension observed in TMEM135 depleted cells could be associated with Rab8 and not its upstream like Rab11 and Rabin8. It has been reported that Rab11 and Rabin8 depletion also prevents CV extension but does not prevent CV formation (21). We also confirmed that Rab11 and Rabin8 depletion did not prevent CV formation but CV extension similar to Rab8 dysfunction and TMEM135 depletion (Appendix Figure S4 A-D).

To exclude the possibility of involvement of Rab8 upstream regulators, we performed TMEM135 and Rabin8 double knockdown and carried out the GST-JCF1 pull down for analyzing the activation state of CA GFP-Rab8. It is well known that Rabin8 is a guanine nucleotide exchange factor (GFE) for Rab8 which is required GDP release and GTP loading on Rab8 (15, 22, 23). As shown in Appendix Figure S4E, activation of CA GFP-Rab8 was prevented in Rabin8 and TMEM135 double knockdown cells. This data indicate that Rabin8 is required for GTP loading on CA GFP-Rab8 and the function of Rabin 8 is intact in TMEM135 depleted cells which allow the activation of CA GFP-Rab8 (Fig 4D and Appendix Figure S4E).

Furthermore, trafficking of Rabin8 to the centrioles was not affected by TMEM135 depletion (Fig EV3C), suggesting that Rabin8 trafficking was not altered by uneven intracellular cholesterol distribution. Thus, TMEM135 depletion seems to specifically affect Rab8 trafficking but not the other ciliogenesis regulators analyzed in this study like Rabin8, EHD1, Rab11, IFT88 which encompasses almost all the early step of ciliogenesis.

Cell viability assay was carried out as previously reported (24).

Additional materials used for the experiments presented in this review

Rab27A (17817-1-AP, proteintech), GM130 (610822, BD Biosciences), anti-mouse alpha-tubulin (T5168, Sigma), anti-rabbit alpha-tubulin (ab52866, abcam), Dynein heavy chain (12345-1-AP, proteintech) antibodies were used. In Situ cell death detection kit (11684795910, Roche), ATP Colorimetric/fluorometric assay kit (K354, biovision), Oil-Red O staining (Sigma), Nocodazole (M1404, Sigma) were used.

The forward and reverse human primer sequences used for qPCR are as follows:

PPAR-alpha (5'-CTATCATTT GCTGTGGAGATCG-3' and 5'- AAGATATCGTCCGGGTG GTT-3'), PGC1-alpha (5'-GCAACATGCTCAAGCCAAAC-3' and 5'-TGCAGTTCAGAG GTTCCA-3'), CPT1-alpha (5'-GCTGGAGGTGGCTTTGGT-3' and 5'-GCTTGGCGGATG GGTTC-3'), MCAD (5'-TTCCAGAGAAGTGTGGAGGTCTT-3' and 5'-TCAATAGCAGTC TGAACCCCTGT-3'), AOX1 (5'-CCTCTGGATCTTCACTTGG-3' and 5'-TGGGTTTCAG GGTTCATACG-3'), HSD17B4 (5'-CCTGGTCTCTCAAGCAGGAT-3' and 5'-CTAACGCTC CTCTTTCTGCAA-3'), SREBP1c (5'-GGAGCCATGGATTGCACATT-3' and 5'-GGCCCC GGAAGTCACTGT-3'), FAS (5'-GCGATGAAGAGCATGGTTTAG-3' and 5'-GGCTCAAG GGTTCATGTT-3')

References related to Referee #2

1. Simpson JC, Joggerst B, Laketa V, Verissimo F, Centin C, Erfle H, Bexiga VR, Heriche, JK, Neumann B, Mateos A, Blake J, Bechtel S, Benes V, Weimann S, Ellenberg J, Pepperkok R (2012) Genome-wide RNAi screening identifies human proteins with a regulatory function in the early secretory pathway. *Nat Cell Biol* **14**: 764-774
2. Islinger M, Lüers GH, Li KW, Loos M, Völkl A (2007) Rat liver peroxisomes after fibrate treatment. A survey using quantitative mass spectrometry. *J Biol Chem* **282**: 23055–23069
3. Wiese S, Gronemeyer T, Ofman R, Kunze M, Grou CP, Almeida JA, Eisenacher M, Stephan C, Hayen H, Schollenberger L, Korosec T, Waterham H.R, Schliebs W, Erdmann R, Berger J, Meyer HE, Just W, Azevedo JE, Wanders RJ, Warscheid B. (2007) Proteomics characterization of mouse kidney peroxisomes by tandem mass spectrometry and protein correlation profiling. *Mol Cell Proteome* **6**: 2045–2057
4. Benjamin JR, Thushara WM, Jon DH, *et al.*, (2018) TEM135 is an LXR-inducible regulator of Peroxisomal metabolism. *bioRxiv* doi: <https://doi.org/10.1101/334979>
5. Chu BB, Liao YC, Qi W, Xie C, Du X, Wang J, Yang H, Miao HH, Li LB, Song BL (2015) Cholesterol transport through lysosome peroxisome membrane contact. *Cell* **161**: 291–306
6. Radhakrishnan A, Ikeda Y, Kwon HJ, Brown MS, Goldstein JL (2007) Sterol-regulated transport of SREBPs from endoplasmic reticulum to Golgi: oxysterols block transport by binding to Insig. *PNAS* **104**: 6511-8
7. Huber LA, Pamplikar S, Parton RG, Virta H, Zerial M, Simons K (1993) Rab8, a small GTPase involved in vesicular traffic between the TGN and the basolateral plasma membrane. *J Cell Biol* **123**:35-45
8. Huber LA, Dupree P, Dotti CG (1995) A deficiency of the small GTPase rab8 inhibits membrane traffic in developing neurons. *Mol Biol Cell* **15**: 918-24
9. Henry L, Sheff DR (2008) Rab8 regulates basolateral secretory, but not recycling, traffic at the recycling endosome. *Mol Bio Cell* **19**:2059-68
10. Exil VJ, Sliva Avila D, Benedetto A, Exil EA, Adams MR, Au C, Aschner M (2010) Stress-induced TMEM135 protein is part of a conserved genetic network involved in fat storage and longevity regulation in *Caenorhabditis elegans*. *PLoS One* **5**: e14228
11. Gosney JA, Ceresa BP (2017) Using percoll gradient fractionation to study the endocytic trafficking to the EGFR. *Methods Mol Biol* **1652**:145-158
12. Gosney JA, Wilkey DW, merchant ML, Ceresa BP (2018) Proteomics reveals novel protein associations with early endosomes in an epidermal growth factor-dependent manner. *J Biol Chem* **293**: 5895-5908

13. Rah SY, Kwak JY, Chung YJ, Kim UH (2015) ADP-ribose/TRPM2-mediated Ca²⁺ signaling is essential for cytolytic degranulation and antitumor activity of natural killer cells. *Sci Rep* **5**:9482
14. Reverter M, Rentro C, Garcia-Melero A, Hoque M *et al.*, Cholesterol regulates Syntaxin 6 trafficking at trans-Golgi network endosomal boundaries. *Cell Rep* **7**:883-97
15. Wang J, Ren J, Wu B, Feng S, Cai G, Tuluc F, Peränen J, Guo W (2015) Activation of Rab8 guanine nucleotide exchange factor Rabin8 by ERK1/2 in response to EGF signaling. *PNAS*. **112**: 148–153
16. Strom M, Hume AN, Tarafder AK, Barkagianni E, Seabra MC (2002) A family of Rab27A-binding proteins. Melanophilin links Rab27a and myosin Va function in melanosome transport. *J Biol Chem* **277**:25423-30
17. Johnson JL, Ellis BA, Noack D, Seabra MC, Catz SD (2005) The Rab27a-binding protein, JCF1, regulates androgen-dependent secretion of prostate-specific antigen and prostatic-specific acid phosphatase. *Biochem J* **391**:699-710
18. Lee WH, Higuchi H, Ikeda S, Macke EL, Takimoto T, Pattnaik BR, Liu C, Chu LF, Siepka SM, Krentz KJ, Rubinstein CD, Kalejta RE, Thomson JA, Mullins RE, Takahashi JS, Pinto LH, Ikeda A (2016) Mouse Tmem135 mutation reveals a mechanism involving mitochondrial dynamics that leads to age-dependent retinal pathologies. *eLife* **5**: e19264
19. Hu JY, Chu ZG, Han J, Dang YM, Yan H, Zhang Q, Liang GP, Huang YS (2010) The p38/MAPK pathway regulates microtubule polymerization through phosphorylation of Map4 and Op18 in hypoxic cells. *Cell Mol Life Sci* **67**: 321-33
20. Liu Q, Insinna C, Ott C, Stauffer J, Pintado PA, Rahajeng J, Baxa U, Walia V, Cuenca A, Hwang YS, Daar IO, Lopes S, Lippincott-Schwartz J, Jackson PK, Caplan S, Westlake CJ (2015) Early steps in primary cilium assembly require EHD1/EHD3-dependent, ciliary vesicle formation. *Nat Cell Biol* **17**: 228–240
21. Wu CT, Chen HY, Tang TK (2018) Myosin-Va is required for preciliary vesicle transportation to the mother centriole during ciliogenesis. *Nat Cell Biol* **20**:175-185
22. Homma Y, Fukuda M (2016) Rabin8 regulates neurite outgrowth in both GEF activity-dependent and independent manners. *Mol Biol Cell* **27**: 2107-18
23. Nachury MV, Loktev AV, Zhang Q, Westlake CJ, peranen J, Merdes A, Slusarski DC, Scheller RH, Bazan JF, Sheffield VC, Jackson Pk (2007) A core complex of BBS proteins cooperate with GTPase Rab8 to promote ciliary membrane biogenesis.
24. Lee JN, park J, Kim SG, Kim MS, Lim JY, Choe SK (2017) 3-Aminotriazole protects against cobalt (II) chloride-induced cytotoxicity by inhibiting reactive oxygen species formation and preventing mitochondrial damage in HepG2 cells. *Mol Cell Toxicol* **13**:125-132

Referee #3:

In this manuscript, Maharjan and colleagues elucidated the function of TMEM135 during cilium-related cellular processes and cholesterol transport in mammalian cells. TMEM135 was previously identified in an shRNA screen as a potential protein that regulates cholesterol transport and was published in biorxiv as an LXR-inducible regulator of peroxisomal metabolism. In agreement with the published biorXiv work, the authors first showed that TMEM135 localizes to peroxisomes and regulates cholesterol transport. Given that they also identified ciliogenesis defects in TMEM135-depleted cells, they investigated the molecular mechanism of this defects and showed that TMEM135 depletion results in defective Rab8 trafficking to the basal body and its activation, as well as IFT20 recruitment to the basal body. Cholesterol supplementation to TMEM135-depleted cells rescued Rab8 trafficking and activation defects, suggesting a link between intracellular cholesterol distribution and ciliogenesis.

The molecular mechanism of ciliogenesis and its regulation is general interest to the field and the novelty of this paper is that it identifies a protein at the intersection of cholesterol biology and ciliary vesicle trafficking during ciliogenesis. Although more needs to be investigated to determine how these pathways are linked, the data included in this paper is sufficient to support this link and opens new avenues for the field. However, for some of their conclusions, the authors did not include the required controls and data, which are listed as major/minor points below. Once these points are addressed by the authors, I recommend publication of this paper in EMBO Reports

Our Reply:

We thank the reviewer for the positive evaluation of our manuscript. In response to Reviewer's comment, we have incorporated the results of number of additional experiments. We believe that these new results effectively address the Reviewer's concerns.

1- Intracellular cholesterol transportation depends on microtubules. If TMEM135 has role on intracellular cholesterol transportation via affecting microtubule organization, it would also affect Rab8 and Ift20 localization around centrosome. They should perform noc washout experiments to determine the affects of TMEM135 on microtubule nucleation and anchoring.

Our Reply

We agree that impaired Rab8 and IFT20 trafficking observed in TMEM135 depletion might be associated with microtubule organization. We performed the microtubule regrowth assay after nocodazole washout. We found that TMEM135 does not affect microtubule nucleation or anchoring as shown in Figure R19'. Similar concerns were raised by Reviewer #2 (also see our response included in Figure R19 in the comment #5 of Reviewer #2). TMEM135 depletion might affect the microtubule depolymerization or re-polymerization as microtubules were found to form curls and buckle with some breakage at the edges of the cell, suggestive of the depolymerized microtubule. Therefore, further studies are required for the analysis of the abundance of polymeric and monomeric tubulin in order to understand the true status of microtubule dynamics in TMEM135 depleted cells.

Figure R19': TMEM135 depletion affects microtubule nucleation and anchoring. Cells grown in coverslips were transfected by siRNA for 48 hours followed by serum starvation for additional 24 hour. Nocodazole (10 μ M) was treated for treatment for 1 hour at 37°C in DMEM for the depolymerization of microtubule, washed with DMEM, and incubated at 37°C for a specified time period to allow microtubule regrowth, followed by fixation with cold methanol for 5 min at -20° and immunostaining with α -tubulin and γ -tubulin antibodies. Representative fluorescent images showing alpha-tubulin (red), γ -tubulin (green), and Nucleus/DAPI (blue). Scale bar, 10 μ m.

2- Based on Smo-GFP localization to basal bodies, they suggest that there are no defects in ciliary vesicle formation. Given that this is one of the major conclusions of the paper, they should show this phenotype by other markers or experiments (i.e. TEM).

Our Reply:

We thank the Reviewer for raising this important issue that the effect of TMEM135 depletion on ciliary vesicle (CV) formation/extension should be sorted by other methods or markers for ciliary vesicle other than Smo. Due to the technical limitation, we preferred to choose immunofluorescence method to TEM for analyzing CV.

EHD1 is essential protein required for fusion of preciliary vesicles into single CV and it has been used as a CV marker along with other CV membrane proteins like Rab8, Smo, ARL13B (1, 2). It has been reported that single EHD1 vesicle localizes with Smo at the distal end of the basal body in Rab8 depleted cells (1). Moreover, Wu *et al.*, have successfully shown that EHD1 is localized in the CV in the early hour of serum starvation while localization of EHD1 as a CV is disrupted when upstream regulator of CV formation, Myo-Va was depleted. (2). Therefore, the presence of a single EHD1 vesicle at the distal end of the basal body in any given condition could suggest, CV formation is intact. In TMEM135 depleted cells, single EHD1 vesicle is found to localize at the distal end of the basal body while EHD1 is localized in the cilium in the control cells suggesting that TMEM135

depletion did not prevent CV formation at the basal body but prevents extension of CV. This result has been incorporated in the revised version of the manuscript as Fig EV4C-D.

3- They must show data to confirm efficient depletion of TMEM135 after siRNA treatment, ideally both by immunofluorescence and immunoblotting. This is important for a) confirming the specificity of the TMEM135 antibody 2) validating the knockdown of TMEM135. They used three different siRNAs in some experiments, and two or one in other. Which one is more efficient in TMEM135 knockdown?

We thank the Reviewer for asking an important question related to the TMEM135 siRNA efficiency and specificity of TMEM135 antibody. Similar issues were raised by Reviewer #1 (also see our response included in Figure R5 related with comment #2 of Reviewer #1). All three TMEM135 siRNAs used in this study significantly decreased the TMEM135 signal from the peroxisomes (Figure R5'), and in the Western blot (Appendix Figure S2A and S2E) suggesting that TMEM135 siRNA and TMEM135 antibody employed in this study is very specific. We agree that most of the experiments were carried out by using TMEM135 siRNA #2 but only some of the important parameters were analyzed by using all three siRNAs (ciliogenesis assay as shown in Fig 2A, B in the revised manuscript) and two siRNAs (lysosomal cholesterol accumulation as shown in Fig EV1E in the revised manuscript). Since all three siRNAs could significantly decrease the TMEM135 signal from peroxisomes (Figure R5'), we believe that all three siRNAs work in terms of the phenotype associated with TMEM135 depletion from peroxisomes including defective Rab8 trafficking (data not shown) and we believe that TMEM135 siRNA sequences used in this study are interchangeable.

Figure R5': TMEM135 siRNA efficiently reduces TMEM135 signals from peroxisomes. Representative image showing that all three siRNAs efficiently reduce TMEM135 protein from peroxisomes stained with PMP70. Scale bar, 10 μ m.

4- How did they conclude for Fig. EV1 that lysosomal accumulation is due to the physical dissociation between peroxisomes and lysosomes? This figure only demonstrates increased cholesterol accumulation upon TMEM135 depletion?

Our reply:

The search for critical determinants involved in intracellular cholesterol trafficking through genome-wide small hairpin RNA screening in combination with amphotericin B selection revealed the peroxisomal proteins, ABCD1, PEX1, PEX3, PEX10, PEX26, ACOT8, BAAT, and TMEM135, as potential candidates regulating intracellular cholesterol transport through lysosome peroxisome membrane contact (LPMC) (3). Deficiency in any of these proteins decrease the possibility of LPMC due to reduced expression of PI(4,5)P2 in the peroxisome membrane which is required for interaction with synaptotagmin 7 present on lysosomal membrane, thereby, triggering robust cholesterol accumulation in lysosomes that mimic the NPC1 phenotype (3). Therefore, we have concluded that lysosomal cholesterol accumulation in TMEM135 depleted cells is due to reduced lysosome-peroxisome overlap as shown in Figure 1 A and B. The Fig EV1 showing lysosomal cholesterol accumulation in old version of manuscript has been changed to Fig EV1E in the new manuscript.

5- What marker did they use to quantify cholesterol levels in sucrose gradients in Fig. 1C as all markers included in this figure are markers of other organelles.

Organelles were separated by subcellular fractionation using a discontinuous sucrose gradient. Each of the fraction of sucrose gradient was subjected to Western blotting to analyze the separation pattern of organelles like lysosomes, endosomes, peroxisomes, Golgi, and plasma membrane. Each of the fraction of sucrose gradient were then subjected towards lipid extraction and total cholesterol measurement using cholesterol assay kit as described in the materials and method section of the manuscript.

6- For Fig. 2A and 2B, they should perform rescue experiments with ectopic siRNA-resistant TMEM135 to confirm the specificity of the ciliogenesis defects.

Our reply:

We thank the Reviewer for raising the concerns about the rescue of ciliogenesis by ectopic siRNA resistant TMEM135 to confirm the specificity of the ciliogenesis defect in TMEM135 depleted cells. Given that human and mouse TMEM135 sequence has considerably high mismatching of nucleotides, we speculated that mouse TMEM135 mRNA could be resistant to the human TMEM135 siRNA. We analyzed the human TMEM135 siRNA efficiency in human cell line RPE1 and mouse embryonic fibroblasts. As shown in Figure R20A-B, all three human TMEM135 siRNA efficiently decreased human TMEM135 mRNA expression level in RPE1 cell while mouse TMEM135 expression level was not changed by any of the human TMEM135 siRNA in MEF cell. To exclude the possibility that siRNA transfection efficiency was not the problem associated with the failure of human TMEM135 siRNA in knockdown of mouse TMEM135 mRNA in MEF cells, MEF cells were transfected by mouse ATG7 siRNA, which significantly decreased the mouse ATG7 mRNA (Figure R20-C). Thus, mouse TMEM135 mRNA is virtually resistant to human TMEM135 siRNA. Next, we use the TMEM135-Myc plasmid (TMEM135 sequence is of mouse origin as described in the materials and method section) to perform rescue experiment for ciliogenesis in TMEM135 depleted cells. As shown in Figure R20D-E, percentage of ciliated cells in TMEM135-myc transfected cells were comparable between control and TMEM135 depleted RPE1 cells while untransfected cells were devoid of cilia in TMEM135 depleted cells.

Figure R20: Transfection of human siRNA resistant TMEM135 rescues ciliogenesis in TMEM135 depleted RPE1 cells. (A, B, C) QPCR analysis of expression of human TMEM135, mouse TMEM135 and mouse ATG7 genes. Data represents mean \pm SD (n=3 experiments). *P < 0.05, Student's t-test. NS, not significant. (D) TMEM135-Myc was transfected in control and TMEM135 depleted cells followed by immunostaining myc and ARL13B antibodies. Representative images of TMEM135-Myc (green) and cilia (red) are shown. Scale bar, 10 μ m. (E) Percentage of ciliated cells in TMEM135-Myc transfected cells. Data represents mean \pm SD (n=3 experiments). 50 transfected cells were scored per condition per experiment. Student's t-test. NS, not significant.

7- They should include a reference for the data on how MbetaCD impacts intracellular cholesterol distribution (Fig. 2F/G-related text). Can they use an assay to show that TMEM135-depletion causes a defect in intercellular cholesterol depletion as this is important for their conclusions in Fig. 2F/G.

Our reply:

The references suggesting the impact of MCD/cholesterol complex on intracellular cholesterol levels have been included in the revised manuscript in the respective figure section as reference #19 (4). Another reference suggesting that MCD/cholesterol supplementation could rescue ciliogenesis during cholesterol depletion has also been included as reference #18 (5).

We had concluded that clearance of accumulated cholesterol from lysosomes of TMEM135 depleted cells did not rescue ciliogenesis. It is because cellular cholesterol itself is important for ciliogenesis as cholesterol depletion itself has negative consequences on ciliogenesis (5). Therefore, depletion of cholesterol from TMEM135 knockdown cells did not recover ciliogenesis as shown in Fig 2C-E. Moreover, we also observed similar phenotypes that cholesterol depletion in control cells itself shows defective ciliogenesis (data not shown) and more studies are needed to elucidate the mechanism.

Reviewer #1 raised a very interesting concern that how does the inhibition of cholesterol synthesis by statins affects TMEM135 depletion. We intend to reply to this Reviewer's concern by the results generated for Reviewer#1 (also see our response included in the Figure R7 in the comment #5 of Reviewer #1). We measured the total cellular cholesterol as shown in Figure R7'B. TMEM135 depletion does not alter cellular cholesterol in both serum-fed and serum-starved conditions. It is interesting to note that even the addition of compactin (statin, HMGCoA inhibitor) does not

decrease total cholesterol in the serum-fed condition in both Control siRNA and TMEM135 siRNA treated cells, suggesting that cells depend upon serum-cholesterol for the maintenance to total cellular cholesterol levels rather than cholesterol biosynthesis. Compactin treatment slightly decreased in total cholesterol in serum-starved cells suggesting that endogenously synthesized cholesterol does contribute to the maintenance of total cellular cholesterol during serum starvation. Therefore, the accumulated cholesterol observed during serum-starvation in TMEM135 depleted cells (Fig 2C) should have been derived from serum-cholesterol before subjecting cells to serum-starvation. Importantly, even the addition of compactin during serum-starvation still results in lysosomal cholesterol accumulation as shown in Figure R7'C, confirming that most of the accumulated cholesterol was obtained from serum-cholesterol rather than endogenously synthesized cholesterol in TMEM135 depleted cells.

TMEM135 depletion clearly impairs intracellular cholesterol transport as the addition of LDL-cholesterol still activated SREBP2 and SREBP2 target genes in TMEM135 depleted cells suggesting that the added LDL-cholesterol was not transported to endoplasmic reticulum to suppress the SREBP2 processing as compared to Control siRNA treated cells (Fig 1E-F). In addition, most of the cellular cholesterol was sequestered in the lysosomes (Fig 1C-D and Fig EV1E in the revised manuscript). As intracellular cholesterol transport is a very dynamic process required for a variety of cellular process, the sequestration of cholesterol in lysosomes disturbs the intracellular cholesterol distribution in TMEM135 depleted cells. Therefore, we performed the rescue experiment with MCD-Cholesterol complex (which do not follow the intracellular cholesterol transport route as LDL-cholesterol) to replenish the cellular cholesterol level, which successfully recovered ciliogenesis in TMEM135 depleted cells, probably through cholesterol redistribution in cells. Furthermore, it has been previously shown that the use of MCD-Cholesterol can reduce ciliogenesis in cholesterol depleted cells (5). Therefore, defective ciliogenesis observed in TMEM135 depleted cells were associated with impaired intracellular cholesterol transport rather than cholesterol depletion.

Figure R7': TMEM135 depletion associated cilia phenotypes is not associated with intracellular cholesterol depletion. (B) Statin decreases total cholesterol in serum-starvation but not in serum-fed condition. Total cholesterol was measured from TMEM135 depleted cells with or without compactin. Control siRNA treated cells in serum-fed condition was taken as control for making relative total cholesterol levels. Data represents mean \pm SD (n=3 experiments*P < 0.05, Student's t-test. (C) TMEM135 depletion mediated cholesterol accumulation is not affected by statin. Representative images showing filipin staining (blue). Scale Bar, 20 μ m.

8- In Fig. 4A, the ability of the wild-type Rab8a and its mutants to localize to cilia varies in control and TMEM135-depleted conditions. The authors only showed that for the affect of TMEM135 depletion on Rab8a localization around the basal body. They should also quantify phenotypes associated with Rab8a ciliary recruitment to reveal the full set of phenotypes. Since Rab8a DN mutant and wt does not localize to cilia in TMEM135-depleted cells (based on their representative images in Fig. 4A), it is also likely that the phenotype is also in part due to the inability of Rab8a to localize to cilia.

We appreciate the reviewer's concern that the defective ciliary phenotype in TMEM135 depleted cells could be partially associated with the inability of Rab8 to localize to cilia. It has been well established that both the endogenous and exogenous Rab8 localizes to the basal body and the cilium (6, 7, 8, 9, 10). We also observe that endogenous Rab8 trafficking to the centrioles was enhanced during serum starvation in the Control siRNA treated cells. We failed to observe the ciliary localization of endogenous Rab8 in most of the ciliated Control siRNA treated cells as mentioned in the text for Fig 3A-B. We did observe the ciliary localization of GFP-Rab8 (wild type and constitutively active mutant) in the cilium as shown in Fig 4A-B. The previous study has suggested that Rab8 is first recruited to the basal body upon serum-removal and its localization is gradually extended along the growing ciliary axoneme (7). Recently the role of Rab8 in the extension of the ciliary vesicle at the mother centriole has been elucidated (1). Moreover, the recruitment of Rab8 to the basal body is regulated by Ahi1 while the activation of Rab8 at the basal body is controlled by Rabin8 (11,6). Therefore, the targeting of Rab8 at the basal body is critical for its function at the primary cilium as an active form of Rab8 modulate the entry of protein cargo to the growing cilium (11). In the present study, TMEM135 clearly disrupts the centriolar trafficking of endogenous Rab8 suggesting that the downstream process of Rab8 function like ciliary vesicle and axoneme extension along with the ciliary localization of Rab8 would be negatively affected. Since Rab8 activity was also compromised in TMEM135 depleted cells, the possibility that Rab8 could accumulate at the cilium is very low as only activated Rab8 localizes to the cilium (12). Moreover, TMEM135 depletion prevents ciliogenesis at the level of ciliary vesicle extension and without intact cilium, it is highly improbable that defective ciliogenesis observed in TMEM135 depleted cells was due to the inability of Rab8 to localize to the cilium.

Expression of Dominant-negative (DN) mutant of Rab8 is well known to suppress ciliogenesis (12). Therefore, the DN mutant Rab8 did not localize to cilia in both Control siRNA and TMEM135 treated cells due to defective ciliogenesis, possibly due to the failure of ciliary vesicle extension (12, 1). Expression of constitutively active (CA) form of Rab8 efficiently rescues ciliogenesis in TMEM135 depleted cells showing similar localization of CA Rab8 in the cilium of both control and TMEM135 depleted cells Fig 4A-B. Wild-type Rab8 was not efficient enough in recovering ciliogenesis in TMEM135 depleted cells resulting in lesser ciliated cells with ciliary localization of WT Rab8. Rescue experiment using WT-Rab8 and mutant Rab8 suggests that TMEM135 depletion not only prevents Rab8 trafficking to the centrioles but also impairs Rab8 activation. Therefore, differences between ciliary localization pattern of WT and mutant Rab8 in control and TMEM135 depleted cells were associated with activation state of WT and mutant Rab8 rather than the inability of Rab8 to enter the cilium.

9- For Fig. 4D, GST must be included as a control in parallel to GST-JCF1 pull-downs. Our reply:

We appreciate the Reviewer's concern about the necessity to include the GST pull-down control. In the revised manuscript we have now included the GST pull-down control in Figure 4D, which suggests that JCF1 (RBD) is responsible for pull down of GTP bound GFP-Rab8 and it is not associated with GST. In the revised manuscript we have now included the GST and GST-JCF1 (RBD) pull-down of constitutively active GFP-Rab8 in the same blot in the Appendix Figure S4E. This experiment was performed to show that function of Rabin8 was not affected by TMEM135 depletion and therefore allows the CA GFP-Rab8 to get activated

10- For Fig. 5F, they must quantify the percentage of IFT20-positive cilia to conclude on ciliary IFT20 recruitment defects, because the decrease in IFT20-positive cilia can just be due to the decrease in ciliogenesis upon TMEM135 depletion. Our reply:

Failure of IFT20 trafficking to mother centriole has been linked to impaired ciliary vesicle formation (1, 13). However, TMEM135 depletion also impairs enhanced IFT20 localization at centrioles during serum-starvation, constitutively active Rab8 efficiently recover ciliogenesis suggesting that the defective ciliogenesis is due to Rab8 function (ciliary vesicle extension) but not IFT20 (ciliary vesicle formation). The previous study has shown that transfected IFT20 localizes to cilium itself (14). In the present study, transfected IFT20 was only present in the cilium of control cells but not in TMEM135 depleted cells possibly due to defective ciliogenesis. This also suggests that IFT20 overexpression could not rescue ciliogenesis in TMEM135 depleted cells. Although IFT20 is abundant, its localization to cilium depends upon Rab8 function. Possibly, IFT20 enter the cilium simultaneously with Rab8 during ciliary vesicle extension along the ciliary axoneme.

11- In the manuscript, for all phenotypes they report a significant change (i.e. ciliogenesis ..), they must include numbers of averages, SEM, p value, how many experiments they performed and how many samples they counted.

Our Reply

We thank the Reviewer for raising the comments as it is necessary to include proper statistics information for clarity of the represented data. In the revised manuscript, we have included number of experiments being performed, the number of cells counted in each experiment, standard deviation, p values, and the test used to generate the p value.

12- For some of the supplementary data (i.e. S4 ..), they only include conclusions on these data in discussion, which makes manuscript very hard to follow and does not address questions until one reads discussion. All the data in discussion must be moved to associated figures in the results section.

We believe that the Reviewer's concern is about Appendix Figure S6 (in the revised manuscript), which shows that TMEM135 depletion does not affect IFT88 trafficking to the centrioles. However, IFT88 deficiency causes massive decrease in ciliogenesis and the role of IFT88 has been speculated to be involved in ciliogenesis in so many ways including, cilium assembly, cilium maintenance, transport of cilia proteins, regulation of cell cycle and microtubules, its involvement in the early ciliogenesis is still obscure. Interestingly, the role of IFT88 seems to be unimportant for up to the early step of ciliary vesicle formation (13). We show that IFT88 at the centriolar region was not affected by TMEM135 depletion suggesting that IFT88 mediated transport of COPII vesicles from ER to centrioles for cilium assembly seem to be unaffected (15). Since the phenotype of IFT88 observed in TMEM135 depletion could not be directly related to the defect associated with the early step involved in ciliogenesis, we choose to include the IFT88 associated parameters in the discussion section.

13- The figures demonstrating peroxisomal localization of myc-TMEM135 and endogenous TMEM135 (referred as Appendix FigS1) is not available as part of the manuscript files. This localization was previously published in a biorxiv paper, which must be cited. <https://www.biorxiv.org/content/10.1101/334979v3>

Our reply

We sincerely apologize for the inconvenience created due to the unavailability of key data. We have included the Appendix Figure S1 in this comment section and we will the request editor to make all the data available to the Reviewers. We acknowledge that the recent report in biorxiv have suggested that TMEM135 is a peroxisome localized protein, but this is not the recent group to show that TMEM135 is a peroxisome protein. Two independent studies of rat and liver peroxisome proteomics had identified TMEM135 (originally referred to as peroxisome membrane protein 52, PMP 52) as a peroxisome localized protein (16, 17) and have been cited in old and revised manuscript.

Appendix Figure S1. TMEM135 localizes to peroxisomes. (A) Huh7 cells were transfected with TMEM135-myc plasmid (Upper and lower panel) and co-transfected with TMEM135-Myc and RFP-SKL plasmid (middle panel), followed by immunostaining with antibodies against PMP70 (red) and Myc (green) in upper panel, Myc (green) in middle panel, Myc (red) and Tomm20 (green) in lower panel. Scale bar, 10 μ m. (B) RPE1 cells were co-transfected with TMEM135-Myc plasmids (Upper and lower panel) and immunostained with antibodies against PMP70 (red) and Myc (green) (Upper panel), and Tomm20 (Green) and Myc (red) (lower panel). Endogenous TMEM135 and PMP70 was stained by TMEM135 antibody (green) and PMP70 antibody (red) middle panel. Scale bar, 10 μ m.

14- The figures demonstrating consequences of TMEM135 depletion on peroxisomal protein abundance is not available (referred as Appendix FigS2).

Our reply

We sincerely apologize for the inconvenience created due to the unavailability of data. We have included Appendix Figure S2 in this comment section and we will request the editor to make all the data available to the Reviewers.

Appendix Figure S2. TMEM135 depletion is not associated with impaired peroxisomal matrix protein import or accumulation of VLCFA. (A) Huh7 cells were transfected with siRNAs as indicated followed by Western blot analysis for TMEM135 knock down efficiency by using TMEM135 antibody. (B) Huh cells were transfected with siRNAs as indicated followed by Western blot analysis of different peroxisome protein (PMP70, catalase, Pex5, Pex14). (C) Huh cells were transfected with siRNAs as indicated and immunostained with antibodies against catalase (red) and PMP70 (green). Scale bar, 10 μ m. (D) Huh7 cells were transfected with siRNAs as indicated followed by GC-MS analysis of VLCFA. Data represent mean \pm SD (n =3). (E) RPE1 cells were transfected with siRNAs as indicated followed by Western blot analysis for TMEM135

antibody. **(F)** RPE1 cells were transfected with siRNAs as indicated and immunostained with antibodies against catalase (red) and PMP70 (green). Scale bar, 10 μ m.

15- They should include in "materials and methods" details on how they quantified the percentage of the lysosomal-peroxisomal overlap for Fig. 1A/B.

Our reply:

The method described for the measurement of lysosome-peroxisome overlap is a very simple method and was adopted from the previous report (3), which requires Colocalization plugins for Image J Software. We believe that the method described in the materials and method section encompasses the whole procedure for calculating the percentage of lysosome-peroxisome overlap. The confocal images (co-immunostained by lysosome and peroxisome markers) are opened in Image J Software followed by measurement of Mander's overlap coefficient (OC), which gives the maximum output value of 1. The OC of lysosome fluorescent signal to the peroxisomes was quantified as percentage taking OC equivalent to 1 as 100%.

16- For Figure 4B, the quantification bars for GFP-Rab8T22N condition is missing from the graph.

Our reply:

The bar graph has zero values for cells expressing dominant-negative (DN, T22N) form of GFP Rab8 since DN GFP-Rab8 inhibits ciliogenesis (12).

17- For Fig. 5B-E, they summarize their conclusions in one sentence, which is not sufficient to support their conclusions and makes it very hard to follow the manuscript.

Our reply:

We appreciate the Reviewer's concern regarding the short description of the experimental results presented in Fig 5B-D in the revised manuscript (Fig 5B-E in the previous version). TMEM135 depletion presented two different phenotypes associated with ciliogenesis, one is Rab8 trafficking defect and another is the inability to perform enhance IFT20 localization at centrioles during serum-starvation. It has been suggested that IFT20 localization at centrioles is required for ciliary vesicle formation (1, 13) while Rab8 is not required for ciliary vesicle formation (1). However, the data in Figure 4 suggests that defective ciliogenesis in TMEM135 depletion is associated with impaired Rab8 function, it was necessary to analyze if Rab8 trafficking defect is associated with the IFT20 function or not. We tried to dissect the mechanism if Rab8 is controlling the IFT20 trafficking and vice versa. The data presented in Fig 5B-D only suggests that although IFT20 depletion severely damage ciliogenesis (probably by suppressing ciliary vesicle formation), Rab8 trafficking to centrioles was not affected. This means that Rab8 is always trafficked to the centrioles irrespective of the presence or absence of ciliary vesicles at the basal body and even though it is trafficked to the basal body without ciliary vesicles, Rab8 cannot function in ciliogenesis. Taken together, Fig5B-D suggests that Rab8 trafficking to the centrioles is independent of the IFT20 function. Furthermore, data presented in Fig 6 suggests that Rab8 activation controls the enhanced IFT20 localization at centrioles. Therefore, we believe that our short summary is sufficient to describe the results shown in Fig 5B-D and the only intention of this experiment was to check in IFT20 depletion affect the Rab8 trafficking to the centrioles or not.

18- They should comment on why they might have observed a contradictory result from a previously published paper for the effects of Rab8a depletion on centriolar IFT20 recruitment (Fig. 6).

Our reply:

We agree with the Reviewer that data shown in Fig 6F-G (in the revised manuscript) is contradictory to those suggested by Liu *et al.*, that Rab8 depletion does not affect the IFT20 recruitment to the basal body (1). It could be because of the timing of the experimental strategies employed. In our method, siRNA was treated for 48 hours followed by serum-starvation for additional 24 hours, followed by fixation, immunostaining, and analysis of IFT20 intensity at the centrioles. Liu *et al.*, have performed Rab8 depletion for 72 hours with an additional 24 hours of incubation in serum-starvation medium followed by fixation, immunostaining and analysis of the IFT20 intensity at the centrioles. Since Rab8 knockdown efficiency data is unavailable, we cannot make sure about the efficiency of siRNA for longer duration which might be the reason that IFT20 localization to centrioles was not reduced significantly by Rab8 siRNA. Furthermore, we have provided evidence that enhanced IFT20 recruitment at centrioles might be associated with Rab8 activation (Fig 6).

- In Figure 1.E, CREB is used as nucleus marker and UBXD8 is used as membrane marker however they are not indicated in text and figure legend.

Our reply:

We appreciate the Reviewer's concerns about minute details. Nuclear marker CREB and membrane marker UBXD8 has been indicated in the Figure legend of Fig 1E in the revised manuscript.

- Cilium are not visible in Figure 5C. They should show better images, maybe with another ciliary marker like Arl13b.

Our reply:

Representative images showing ciliated cells have been replaced with new images, stained with ARL13B as shown in Fig 5C (in the revised manuscript). Although the bar graph showing quantitative data have changed, the meaning of data is same as in old version. In the figure 5C we have included both the representative and quantitative data as same figure.

- Representative images are needed for Figure 5E.

Our reply:

The representative images showing Rab8 trafficking to centrioles in each condition has been included in Figure 5E in the revised manuscript.

- Figure numbers are missing.

Our reply:

All the figure numbers have been carefully checked and assigned in the revised manuscript.

Additional siRNA sequence and qPCR primers used:

Mouse ATG7 siRNA sequence 5'-CUGUUCACCCAAAGUUCUU-3'

Mouse ATG7 forward primers 5'- AAAGGCTTTCACCAAAACAGAT-3'

Mouse ATG7 reverse primers 5'- CTCGACACAGATCATATAGGC-3'

Mouse TMEM135 forward primers 5'- TCATGGACTCCTGGCTTTG-3'

Mouse TMEM135 reverse primers 5'- TGGCCATATAAATTGTGAGCA-3'

References related to Referee #3

1. Liu Q, Insinna C, Ott C, Stauffer J, Pintado PA, Rahajeng J, Baxa U, Walia V, Cuenca A, Hwang YS, Daar IO, Lopes S, Lippincott-Schwartz J, Jackson PK, Caplan S, Westlake CJ (2015) Early steps in primary cilium assembly require EHD1/EHD3-dependent, ciliary vesicle formation. *Nat Cell Biol* **17**: 228–240
2. Wu CT, Chen HY, Tang TK (2018) Myosin-Va is required for preciliary vesicle transportation to the mother centriole during ciliogenesis. *Nat Cell Biol* **20**:175-185
3. Chu BB, Liao YC, Qi W, Xie C, Du X, Wang J, Yang H, Miao HH, Li LB, Song BL (2015) Cholesterol transport through lysosome peroxisome membrane contact. *Cell* **161**: 291–306
4. Zidovetzki R, Levitan I (2007) Use of cyclodextrins to manipulate plasma membrane cholesterol content: evidence, misconceptions and control strategies. *Biochim Biophys Acta* **1768**:1311-24
5. Maerz LD, Burkhalter MD, Schilpp C, Wittekindt OH, Frick M, Philipp M (2019) Pharmacological cholesterol depletion disrupts ciliogenesis and ciliary function in developing zebrafish. *Commun Biol* **2**:31
6. Hsiao YC, Tong ZJ, Westfall JE, Ault JG, Page-McCaw PS, Ferland RJ (2009) Ahi1, whose human ortholog is mutated in Joubert syndrome, is required for Rab8 localization, ciliogenesis and vesicle trafficking. *Hum Mol Genet* **18**:3926-41
7. Westlake CJ, Baye LM, Nachury MV, Wright KJ, Ervin KE, Phu L, Chalouni C, Beck JS, Kirkpatrick DS, Slusarski DC, Sheffield VC, Scheller RH, Jackson PK (2011) Primary cilia membrane assembly is initiated by Rab11 and transport protein particle II (TRAPII) complex-dependent trafficking of Rabin8 to centrosome. *PNAS* **108**:2759-64
8. Feng S, Knodler A, Ren J, Zhang J, Zhang X, Hong Y, Huang S, Peranen J, Guo W (2012) A Rab8 guanine nucleotide exchange factor-effector interaction network regulates primary ciliogenesis. *J Biol Chem* **287**:15602-9
9. Kuhns S, Schmidt KN, Reymann J, Gilbert DF, Neuner A, Hub B, Carvalho R, Wiedemann P, Zentgraf H, Erfle H, Klingmuller U, Boutrous M, Pereira G (2013) The

- microtubule affinity regulating kinase MARK4 promotes axoneme extension during early ciliogenesis
10. Wang L, Lee K, Malonis R, Sanchez I, Dynlacht BD (2016) Tethering of an E3 ligase by PCMI regulates the abundance of centrosomal KIAA0586/Talpid3 and promotes ciliogenesis. *Elife* **5**: e12950
 11. Hsiao YC, Tuz K, Ferland RJ (2012) Trafficking in and to the primary cilium. *Cilia*. **1**: 4
 12. Nachury MV, Loktev AV, Zhang Q, Westlake CJ, Peranen J, Merdes A, Slusarski DC, Scheller RH, Bazan JF, Sheffield VC, Jackson PK (2007) A core complex of BBS proteins cooperate with GTPase Rab8 to promote ciliary membrane biogenesis.
 13. Joo K, Kim CG, Lee MS, Moon HY, Lee SH, Kim MJ, Kweon HS, Park WY, Kim CH, Gleeson JG, Kim J (2013) CCD41 is required for ciliary vesicle docking to the mother centriole. *PNAS* **110**: 5987–5992
 14. Follit JA, Tuft RA, Fogarty KE, Pazour GJ (2006) The intraflagellar transport protein IFT20 is associated with the Golgi complex and is required for cilia assembly. *Mol Biol Cell* **17**: 3781–3792
 15. Ding J, Shao L, Yao Y, Tong X, Liu H, Yue S, Xie L, Cheng SY (2017) DGK δ triggers endoplasmic reticulum release of IFT88-containing vesicles destined for the assembly of primary cilia. *Sci Rep* **7**: 5296
 16. Islinger M, Lüers GH, Li KW, Loos M, Völkl A (2007) Rat liver peroxisomes after fibrate treatment. A survey using quantitative mass spectrometry. *J Biol Chem* **282**: 23055–23069
 17. Wiese S, Gronemeyer T, Ofman R, Kunze M, Grou CP, Almeida JA, Eisenacher M, Stephan C, Hayen H, Schollenberger L, Korosec T, Waterham H.R, Schliebs W, Erdmann R, Berger J, Meyer HE, Just W, Azevedo JE, Wanders RJ, Warscheid B. (2007) Proteomics characterization of mouse kidney peroxisomes by tandem mass spectrometry and protein correlation profiling. *Mol Cell Proteome* **6**: 2045–2057

2nd Editorial Decision

22 January 2020

Thank you for submitting the revised version of your manuscript. It has now been seen by two of the original referees. My apologies for this unusual delay in getting back to you. It took longer than anticipated to receive the referee reports due to the recent holiday season.

As you can see, the referees find that the study is significantly improved during revision and recommend publication here. Before I can accept the manuscript, I need you to address some minor points below:

- Referee #2 finds that Rab8 immunofluorescence in Figure 3A might not be representative as they show differences in Rab8 levels which are not consistent with the western blots in Figure 4. Please address this concern by providing more representative immunofluorescence images.
- Referee #1 finds that it would be better if you include some of the data from the point-by-point response into the manuscript as Expanded View Figures. I agree that this would make the data more accessible.

REFeree REPORTS

Referee #1:

In this revised manuscript and the related response letter the authors addressed all of my concerns. Some of the information and data provided within the response to the reviewers might be added to the main manuscript.

Referee #2:

Although a lot of work is presented here, we still do not really learn the precise role of TMEM135. The conclusions of the paper include, "TMEM 135 depletion blunted Rab8 trafficking to the centrioles." Key data are shown in Fig. 3A. Why is the Rab8 staining so much weaker in TMEM

siRNA condition with and especially without serum? The blots in 4 indicate no change in total protein so better images with equal image level intensity need to be included with better analysis. This is not proper unless there is really a decrease in protein level and this kind of inconsistency indicates that the authors need to evaluate their findings with greater care. Not sure we really know what happens to Rab8 although differences in JCF1 binding are seen. This reviewer still is not confident that the fig. 1 gradient is correct--Golgi complexes float with endosomes-- but has to trust the authors response.

2nd Revision - authors' response

31 January 2020

Reviewer #1:

In this revised manuscript and the related response letter the authors addressed all of my concerns. Some of the information and data provided within the response to the reviewers might be added to the main manuscript.

We thank this Reviewer for the constructive criticisms which significantly contributed to our understanding about the role of cholesterol on ciliogenesis beyond the scope of our study. We deeply appreciate this Reviewer's point in accommodating data and informations provided in the response letter to the revised manuscript. After having several discussion series, we reached a conclusion that even we add more data from response letter to the manuscript, it will not contribute much to the conclusions that have been drawn from this study and manuscript tends to become unnecessarily lengthy. Especially, the data generated in response to the comments raised by the Reviewer#2 and Reviewer#3 have already been added in the previous version of revised manuscript which contributed significantly to the boost the conclusion of this study. We acknowledge the comment that if TMEM135 siRNA could actually deplete TMEM135 from peroxisomes as very critical to define basic aspects associated with TMEM135 depletion and ciliogenesis. Therefore, we have decided to include this data in the revised manuscript and has been incorporated as Appendix Figure 3.

Referee #2:

Although a lot of work is presented here, we still do not really learn the precise role of TMEM135. The conclusions of the paper include, "TMEM 135 depletion blunted Rab8 trafficking to the centrioles." Key data are shown in Fig. 3A. Why is the Rab8 staining so much weaker in TMEM siRNA condition with and especially without serum? The blots in 4 indicate no change in total protein so better images with equal image level intensity need to be included with better analysis. This is not proper unless there is really a decrease in protein level and this kind of inconsistency indicates that the authors need to evaluate their findings with greater care. Not sure we really know what happens to Rab8 although differences in JCF1 binding are seen. This reviewer still is not confident that the fig. 1 gradient is correct--Golgi complexes float with endosomes-- but has to trust the authors response.

We absolutely agree with this reviewer that we do not know the precise role of TMEM135 until now. We have mentioned in the previous point-by-point referee report that all the existing study on TMEM135 tends to contradict with each other in one way or another and elucidation of TMEM135 function might require some kind of special condition. The ground breaking finding by Chu *et. al.*, that involvement of peroxisome in the intracellular cholesterol transport, suggested that beyond the inherent function of certain peroxisomal proteins, they also contribute in maintaining lysosome peroxisome membrane contact for efficient balance in intracellular cholesterol homeostasis. Although we do not know the inherent function TMEM135, its role in intracellular cholesterol homeostasis could not be ignored as it is linked with ciliogenesis.

We have always observed that Rab8 staining tends to be slightly weaker in serum fed condition. During serum starvation, Rab8 staining was enhanced and tends to localize around the centriolar region and almost all cells have Rab8 localized to the centrioles. Total Rab8 protein level was always similar in both serum-fed and serum starvation by Western blot. However, we did observe the higher amount of activated Rab8 in serum-starved condition suggesting the correlation between Rab8 activation and enhanced trafficking of Rab8 at the centrioles and centriolar area (Fig 3A and 4C). We would like to emphasize that TMEM135 depletion did not alter the total amount of Rab8 in either serum-fed or serum-starved condition as observed by both the immunofluorescence and Western blot. Because TMEM135 depletion results in morphologically aberrant bigger size Rab8 vesicles in the immunofluorescence it seems that TMEM135 depletion decreases Rab8 in immunofluorescence which contradicts Western blotting results of Rab8. We agree with this

reviewer that proper images with equal image level intensity should be provided to match the Western blot results. Therefore, the representative image showing Rab8 distribution in serum-starved TMEM135 knockdown condition has been changed and intensity of all the images shown in Fig 3A has been equalized.

Reference

Chu BB, Liao YC, Qi W, Xie C, Du X, Wang J, Yang H, Miao HH, Li LB, Song BL (2015) Cholesterol transport through lysosome peroxisome membrane contact. *Cell* **161**: 291–306

Accepted

14 February 2020

Thank you for submitting your revised manuscript. I have now looked at everything and all looks fine. Therefore I am very pleased to accept your manuscript for publication in EMBO Reports.

Corresponding Author Name: Raakil Park

Manuscript Number: EMBOR-2019-48901V2